# Characterizing the complexity of microseismic signals at slow-moving clay-rich debris slides: The Super-Sauze (Southeastern France) and Pechgraben (Upper Austria) case studies

Naomi Vouillamoz[1], Sabrina Rothmund[1], Manfred Joswig[1]

[1]Insitute for Geophysics, University of Stuttgart, 70174 Stuttgart, Germany

*Correspondence to*: Naomi Vouillamoz (naomi.vouillamoz@geophys.uni-stuttgart.de)

**Abstract.** Soil and debris slides are prone to rapid and dramatic reactivation. Deformation within the instability is accommodated by sliding, whereby weak seismic energies are released through material deformation. Thus, passive microseismic monitoring provides information that relate to the slope dynamics. In this study, passive microseismic data acquired at Super-Sauze (Southeastern France) and Pechgraben (Upper Austria) slow-moving clay-rich debris slides ("clayey landslides") are investigated. Observations are benchmarked to previous similar case studies to provide a comprehensive and homogenized typology of microseismic signals at clayey landslides. A well knowledge of the various microseismic signals ~~potentially triggered~~ generated by the slope deformation is crucial for the future development of automatic detection systems to be implemented in early-warning systems. Detected signals~~seismic events~~ range from short duration (< 2 s) quake-like signals to a wide variety of longer duration tremor-like radiations (> 2 s – several min). Complex seismic velocity structures, low quantity and low-quality of available signal onsets and non-optimal seismic network geometry severely impedes the source location procedure, thus rendering source processes characterization challenging. Therefore, we constrain sources location using the prominent waveform attenuation pattern characteristic of near-source area (< ~about 50 m) landslide-induced microseismic events. A local magnitude scale ~~($M_L$)~~ for clayey landslides ($M_{L-LS}$) is empirically calibrated using calibration shots and hammer blows data. The derived $M_{L-LS}$ ~~scale~~ returns landslide-induced microseismicity rates that correlate ~~in general~~positively with higher displacement rates. However, high temporal and spatial resolution analyses of the landslide dynamics and hydrology are required to better decipher the potential relations linking landslide-induced microseismic signals to landslide deformation.

## 1 Introduction

Slow-moving soil and debris slides (Hungr et al., 2014) developed in tectonised marl formations are characterized by seasonal dynamics as well as by sudden (generally rainfall triggered) reactivation and liquefaction phases (e.g. Malet et al., 2005). The slow deformation of soil and debris slides is expected to generate elastic accumulation and rupture whereby seismic energy will be released within the landslide body. Therefore, passive seismic monitoring is a good approach to monitor and mitigate slope instabilities since it provides high temporal resolution data (sample rates up to 1,000 Hz) in near

real-time that relate to the dynamics of the landslide. The transition from steady-state sliding to a rapid transformation of the landslide in a debris flow may be detected and slope failure anticipated.

Seismic investigations of natural and artificial slope instabilities started in the 1960's with acoustic emission (AE: 10-1,000 kHz) (e.g. Beard, 1961; Cadman and Goodman, 1967; Jurich and Miller Russell J., 1987) and have been complemented

during the last decades by an increasing number of passive microseismic monitoring studies (1-1,000 Hz), carried out in various geological context. The shear boundaries of the Slumgullion earthflow in Colorado were first investigated by Gomberg et al. (1995) as a strike-slip fault zone analog. The study confirmed the existence of detectable brittle deformation processes associated to the slide deformation. In Europe, investigated clayey landslides include the Heumoes slope in the Austrian Vorarlberg Alps (Walter and Joswig, 2008; Walter et al., 2011), the Super-Sauze landslide in the French

Southwestern Alps (Walter and Joswig, 2009; Walter et al., 2012; Tonnellier et al., 2013; Provost et al., 2017) and the Valoria landslide in the Northern Apennines in Italy (Tonnellier et al., 2013). Case studies carried out at rockslides include for example the Randa rockslide in the Swiss Alps (Eberhardt et al., 2004; Spillmann et al., 2007); the Åknes rockslide in Norway (Roth et al., 2005; Fischer et al., 2014); the Séchilienne rockslide in the Southeastern French Alps (Helmstetter and Garambois, 2010; Lacroix and Helmstetter, 2011); and the Gradenbach, Hochmais-Atemskopf and Niedergallmigg-

Matekopf deep-seated rock slope deformations in the Eastern Austrian Alps (Brückl and Mertl, 2006; Mertl and Brückl, 2007; Brückl et al., 2013).

Observed near (receiver-source distances < 0.5-1 km) microseismic signals comprise microearth-quake-like events, for which Gomberg et al. (1995) introduce the term 'slidequake'. Such events have been reported both at rock and debris slides and are inferred to be associated to fracture processes in the host rock, at the sliding surface, or within the landslide body.

Rockfalls and rock-avalanches signals were also characterized at steep debris slides and at rockslides (Helmstetter and Garambois, 2010; Walter et al., 2012; Tonnellier et al., 2013; Provost et al., 2017). In addition, a wide variety of tremor signals have been reported marginally (Gomberg et al., 1995; Brückl and Mertl, 2006; Mertl and Brückl, 2007; Spillmann et al., 2007; Gomberg et al., 2011; Walter et al., 2012; Tonnellier et al., 2013; Provost et al., 2017). No common typology has yet been suggested for these signals and the signal source interpretation remains speculative.

This study aims at proposing a classification of microseismic signal types observed at slow-moving clay-rich debris slides ("clayey landslides") based on simple waveform and spectral attributes of the signals and using microseismic observations reported by similar case studies as a benchmark. Standard seismological approaches to source location derive minimum uncertainties of ± 50 m for near-source area microseismic events at clayey landslides (e.g. Tonnellier et al., 2013). Therefore, Wwaveform attenuation patterns of natural events were then used to constrain the receiver-source distance of near-source

area landslide-induced microseismic events. The study stresses the difficulties held in signal sources characterization: (1) Seismic velocities show drastic variations in space (highly heterogenous material) and time (change in water content and deformation of the landslide body). (2) Waveforms are strongly scattered and attenuated due to the landslide material heterogeneity and high saturation; phases onsets are difficult to identified. (3) The network geometry is in most cases non-optimal relative to the signal source. These complexities severely impact seismic event location and uncertainties of at best

20-50 m are obtained for calibration shots carried out within the recording network. The shallow installation of seismic stations in the landslide body result in high level of noise contamination of the data and the distinction between landslide-induced seismic events and other environmental (or anthropological) sources is difficult. Since the uncertainties held in the source location impact on a power law basis the source magnitude estimation, the analysis of seismicity rate is challenging.

ThereforeWith the aim, to reduce bias and errors in the estimation of landslide-induced microseismicity rates, the distance attenuation function of the $M_L$ local magnitude scale was then calibrated for clayey landslides using calibration shots and hammer blow datasets. Waveform attenuation patterns of natural events were then used to constrain the receiver-source distance. Microseismic observations were gathered in a comprehensive catalog. The final catalog of landslide-induced microseismic signals provides an initial microseismic signals library to train automatic detection and classification systems as well as an important basis for a multidisciplinary comparative analysis with other landslides observations such as displacement, cracks and fissures development, or hydrometeorological data to gain knowledge about landslide dynamics.Using this approach, first results show an increase of seismicity rates with higher average displacement rate.

## 2 Data

Seismic measurements were acquired at two well-instrumented slopes: The Super-Sauze (Southwestern French Alps) and Pechgraben (Upper Austria) landslides (Fig. 1a-b). Both instabilities are characterized by a clay-rich matrix transporting rigid boulders of marls and limestones (including leftovers and remains of vegetation at Pechgraben) with moving rates ranging between a few mm $d^{-1}$ up to several tenths of cm $d^{-1}$per day in the investigated areas and periods (Fig. 1a1c-d). In the monitored areas, the thickness of the instability reaches more than 10 m at Super-Sauze, but do not exceed a few meters (2-4 m) at Pechgraben. More details about the two landslides can be found in Malet (2003); Travelletti (2011); Tonnellier et al. (2013) for Super-Sauze and Lindner et al. (2014); Lindner et al. (2016) for Pechgraben.

Continuous data of three seismic campaigns are were investigated (Fig. 1):

- **Super-Sauze 2010 (SZ10)**: May 28–July 24, 2010; 58 days; 18 sensors in 2 ha; average displacement of 0.4 mm cm $d^{-1}$, obtained by daily dGNSS (differential global navigation satellite system) measurements.

- **Pechgraben 2015 (PG15)**: October 7-15, 2015; 9 days; 12 sensors in 6 ha; average displacement of 2 cm $d^{-1}$, obtained by weekly dGNSS measurements.

- **Pechgraben 2016 (PG16)**: November 8-12, 2016; 5 days; 12 sensors in 1 ha; average displacement of >more than 20 cm $d^{-1}$, estimated by triangulation, using grids of fixed nails both on the stable and on the active part of the slide and daily photo-monitoring.

Tripartite seismic arrays were deployed with station spacing of 5-50 m (Fig. 1c-d). Each seismic array consists of a central three-component (3-C) short-period seismometer (Lennartz 3Dlite) which is surrounded by three to six vertical short-period seismometers (Lennartz 1Dlite). The seismometers are buried about 30 cm deep in the landslide material. Data are were collected by battery powered SUMMIT M Hydra data loggers. At Super-Sauze, the array S3 consists of Noemax Agécodagis

velocimeters (one 3-C and six verticals) with associated bandpass of 0.1-80 Hz, connected to a Képhren Agécodagis acquisition system powered by solar panels. This array is part of a permanent monitoring installation (National French Landslide Observatory Facility and RESIF Datacenter, 2006). The seismometers feature therefore a robust installation and are housed in plastic drums on top of a concrete slab. A comparison of the data collected by the different installation systems

proved consistent: identical waveforms featuring similar amplitudes are observed for microseismic events recorded at the co-located stations S1.5, S2.6 and S3.6. ~~At SZ10 and PG15, the seismic campaigns targeted areas of various dynamics (Fig. 1c-d). The objective was to detect potential variations in seismicity rates and test whether signals are less scattered when recorded by stations operated in stable ground. Whereas more landslide-induced signals were observed at the higher dynamic location, n~~No significant difference in terms of waveform scattering was found for signals recorded by stations installed in

the more stable areas. Due to the relatively large aperture (30-50 m) of the PG15 seismic arrays, many near-source area microseismic events were recorded by less than three sensors. Consequently, a denser seismic network configuration was designed for the PG16 campaign. Inherent difficulties of operating systems continuously on landslides resulted in partially incomplete datasets (Fig. 1e). This aspect must be considered ~~in the~~when evaluati~~ng~~~~on of~~ the completeness of landslide-induced microseismic catalogs ~~completeness~~.

**3 Method**

Data were analyzed following the "*Nanoseismic Monitoring*" methodology using the NanoseismicSuite software package developed at the Institute for Geophysics of the University of Stuttgart (Wust-Bloch and Joswig, 2006; Joswig, 2008; Sick et al., 2012; Vouillamoz et al., 2016). The method is supported by a realtime, analyst-guided interactive multi-parameter visualization approach. First, signals are identified by visual screening of continuous sonogram, where sonograms are

logarithmically scaled spectrograms featuring a dynamic frequency-dependent noise adaptation. The enhanced visualization of sonograms has unmatched power to ~~that~~ facilitate the detection and recognition of various type of weak signal energies in low-SNR (signal-to-noise ratio) conditions without a-priori knowledge (Joswig, 1990; Sick et al., 2012; Vouillamoz et al., 2016). The SonoView module of the NanoseismicSuite software provides a dynamic layout, where single-trace sonograms or multi-trace (array-stacked) super-sonograms are visualized on a common time line, with up to several hours in one laptop

screen. Different resampling can be applied to the data, facilitating the focus on various event types (short/long duration, low/high frequency). Detected events are tagged and synchronized in the linked HypoLine module of the software suite for further evaluation. There, waveforms are analyzed interactively to provide an optimized graphical hypocentral solution~~.~~, Data can be simultaneously ~~processing~~ processed~~data~~ in network and array mode, taking advantage of the tripartite configuration of the seismic mini-arrays (see Joswig (2008) and Vouillamoz et al. (2016) for a comprehensive description of

the HypoLine software). The strength of the method is its ability to easily detect and successfully evaluate any kind of signals without a-priori knowledge in noisy environment. The drawback is that the process is not automated. It is therefore time-consuming and not well-suited for large datasets (years). Results may also not be reproducible to 100 %.

Much attention was paid to design a comprehensive database gathering all microseismic signals observed by passive microseismic monitoring on active debris slides. Continuous sonograms of the three seismic datasets (SZ10, PG15, PG16) were visually screened in SonoView. To avoid false noise detection, special attention was paid when screening day-time measurements contaminated by with a focus on night time measurements to minimize false noise detections (anthropogenic noise caused by geophysicists or geotechnical work during the day)carried out on the slope. Only signals recorded coherently by three sensors at least were considered as a detection. Each detection was first picked for evaluationed individually and interactively in HypoLine, where phases information were picked, and back azimuth and apparent wave-front velocities calculated. Local, regional and distant seismicity was catalogued as well. Various external sources of noise were gathered with the aim to initiate a template library of seismic signatures produced by noise events. Potential landslide induced signals were finally classified using simple spectral and waveform features of the observed seismic signals.

### 3.1 Catalog design

Much attention was paid to design a consistent seismic catalog. First, all sonogram based detection recorded consistently by at least three sensors were catalogued. Second, for each catalogued detection,Then, waveform and spectral features of the all signals were evaluated qualitatively in HypoLine and by analyzed quantitatively using MATAB® routines: (1) For each event, Aall vertical trace seismograms of the seismic network were visualized on a common time-line with normalized and non-normalized amplitudes, using a set of pre-defined time windows (5, 10, 30, 60 and 120 s). The event duration, the signals coherency and the waveform attenuation pattern across the seismic network were checked. (2) Traces on which the signal of interest is contaminated by noise and traces that did not record the event were tagged and discarded from further analysis. (23) For each trace that recorded the event, the non-logarithmic spectrogram, the unfiltered waveform and a series of selected band-passed waveforms were plotted and evaluated. (34) The amplitude spectrum (FFT, fast Fourier transform) was calculated to estimate the dominant frequency content of the signals. Last, for those events featuring trackable wave packets, back azimuth and apparent velocities were calculated for individual wave packets using the interactive array processing module of the HypoLine software.Potential landslide-induced microseismic events were finally classified We considering the following waveform and spectral features for the classification:

- **Signal duration in seconds**. Signals are classified in three duration classes: short duration (< 2 s); medium duration (2-20 s); and long duration signals (minutes).

- **Waveform attenuation pattern**. The signals of landslide-induced microseismic sources are expected to be severely attenuated, because of their source proximity and their propagation through heterogenous clay-rich soils of various water saturation (e.g. Aki and Richards, 2002; Koerner et al., 1981). Only those events featuring prominent and consistent attenuation of the signal maximum amplitudes across the seismic network are considered as potentially landslide-induced by the landslide dynamics.

- **Signal onset**. Rather impulsive, broadband onsets are distinguished from emergent onsets.

- **Dominant frequency**. The waveforms and the distribution of the dominant energies at individual station records ~~is~~ are evaluated in five frequency bands: 1-5; 5-20; 20-50; 50-100 and 100-200 Hz. The existence of signal content in higher bandpassed waveforms provides an additional clue about landslide-induced microseismic events source proximity.

- **Apparent velocity of trackable wave packets**. Well-constrained apparent velocities (computed by array-processing for wave packets showing at least four traces with correlation thresholds > 70 %) range from less than 0.2 km s$^{-1}$ to more than 5.0 km s$^{-1}$. We distinguish three classes of apparent velocities: < 0.5 km s$^{-1}$ (top most volume of the landslide body); 0.5-2.0 km s$^{-1}$ (landslide body); > 2.0 km s$^{-1}$ (sedimentary bedrock), in agreement with published velocity profiles at clayey landslides (Williams and Pratt, 1996; Tonnellier et al., 2013).

Based on these features and using previous studies (Gomberg et al., 1995; Walter and Joswig, 2008, 2009; Gomberg et al., 2011; Walter et al., 2012; Tonnellier et al., 2013; Provost et al., 2017) as a benchmark, microseismic events detected at clayey landslides are gathered in ~~four~~ three main groups that we describe and discuss in the next section:

1. Earthquakes (local, regional and teleseismic).
2. Quakes (receiver-source distance < 50-500 m).
3. Tremors (Landslide-induced tremor ~~like~~ signals and external sources of tremor-like radiations).
4. ~~External sources of tremor-like radiations.~~

## 4 Unified microseismic signals typology at clayey landslides

To help the reader in the comparison of the different microseismic signals, we apply the layout of Figure 2 for all representative events of the classification (where only vertical traces are presented):

a. Displays the signal sonogram (Joswig, 1990) up to the Nyquist frequency with a logarithmic ordinate, which corresponds to 1.95-250 Hz for Pechgraben data and to 3.91-500 Hz for Super-Sauze data. Darker colors indicate higher relative energies.

b. Shows the non-logarithmic spectrogram of the signal, with an ordinate up to 250 Hz. The time-window is taken as the signal length divided by 30 and an overlap of 90 % was applied. Red colors indicate higher energies. Both the MATLAB® spectrogram code and colormap were provided by Clément Hibert, of the EOST (Ecole et Observatoire des Sciences de la Terre), University of Strasbourg, France.

c. Displays the unfiltered seismogram with maximum absolute 0-to-peak amplitude indicated above the trace in nm s$^{-1}$.

d. Provides a selection of bandpassed waveforms in nm s$^{-1}$, filtered from bottom to top between 1-5, 5-20, 20-50, 50-100 and 100-200 Hz using a second order Butterworth filter. Maximum absolute 0-to-peak amplitudes are indicated in nm s$^{-1}$ above each respective trace.

e. Provides the amplitude spectrum in nm Hz$^{-1}$, computed by FFT for the time window indicated by the red bar in (d).

## 4.1 Earthquakes (local, regional and teleseismic)

Local, regional and teleseismic earthquakes are detected daily by seismic networks. Because earthquakes are potential trigger of landslides, it is important to ~~include~~ catalogue these events ~~in landslides seismicity catalogs~~. Seismic features of earthquakes are well known from routine seismogram analysis. At clayey landslides, earthquakes produce medium to long duration signals that are recorded with similar amplitudes across the complete seismic network. The duration and strength of an earthquake signal as well as its frequency content vary as a function of source distance and magnitude. Sharp and broadband distribution of initial frequency content is typically followed by a decrease in frequency content of the signal energy with successive phase onsets, resulting in a typical triangular-shaped sonogram pattern for earthquakes. Onsets of high-SNR events are impulsive. Individual phases with moderate scattering can be identified and return apparent velocities above 2.0 km s$^{-1}$ (Table 1, Fig. 2).

## 4.2 Quakes

### 4.2.1 Previous observations

Quake signals have been observed in previous studies carried out at clayey landslides. Gomberg et al. (1995) and Gomberg et al. (2011) report short-duration earthquake-like signals, with clearly discernable, trackable wave packets that they refer as slidequakes. Dominant frequencies of slidequakes are not stated, but can be evaluated visually between 10 and 100 Hz based on the waveforms displayed in Figure 5 and Figure 6 of Gomberg et al. (2011). Walter et al. (2012) describe earthquake-like events with duration of up to 5 seconds and associated frequency content of 10-80 Hz, which they refer as slidequakes after Gomberg et al. (1995). Tonnellier et al. (2013) and Provost et al. (2017) report quake-like signals with duration of about one second, dominant frequencies around 10 Hz, emergent first arrivals and undistinguishable P- and S-waves.

### 4.2.2 Updated classification of quake signals

Based on ~~duration,~~ waveform attenuation pattern, ~~and~~ dominant frequency content and duration of the signals, we propose four types of quake events which represent a continuum between very near-source area quake events recorded only at a few nearby stations to local micro-quake events recorded consistently across the complete seismic network (Table 1~~Table 1~~; Fig. 3 and Fig. 4).

- **Type I – Near high-frequency quakes.** Signals show durations of less than 1 s and are recorded only at a few nearby stations, suggesting ~~receiver source distances < ~50 m~~ a nearby source (Fig. 3a). The range of signal amplitudes reaches several orders of amplitude units (Fig. 3e). Maximum absolute amplitudes of about 10,000 nm s$^{-1}$ were observed. High-SNR signals feature impulsive onsets. Dominant frequencies of the highest amplitude traces

are in the 20-100 Hz range (spectrogram, bandpassed waveforms and amplitude spectrum in Fig. 4a). P- and S-phases cannot be clearly distinguished; however, successive phases may be identified based on the apparent velocity of trackable wave packets that scale within 0.2-1.8 km s⁻¹ ~~(later phases are slower)~~.

- **Type II – Near low-frequency quakes**. Signals have duration of 1-2 s and are recorded by the complete seismic network with amplitudes also ranging within a few orders of amplitude units and suggesting a nearby source ~~(< ~50 m)~~ (Fig. 3b and 3e). Maximum amplitudes of a few 10,000 nm s⁻¹ were observed. Dominant frequencies of the highest amplitude signals stay typically in the 5-50 Hz range (spectrogram, bandpassed waveforms and amplitude spectrum in Fig. 4b and 4e, lower panel). The signals consist of prominent and scattered surface waves that can be tracked over the seismic network. P- and S-phases cannot be clearly distinguished, but successive phases can eventually be discriminated based on the apparent velocity of trackable wave packets that range within 0.2-1.8 km s⁻¹.

- **Type III – Moderate distance quakes**. Signals last 1.5-2.5 s and are recorded by the complete seismic network with consistent amplitudes across the complete seismic network suggesting ~~receiver-source distances of up to a few 100 m~~ a source outside of the seismic network (Fig. 3c and 3e). Most events feature low amplitudes and are recorded just above the noise threshold (100-500 nm s⁻¹). Dominant frequencies are in the 5-50 Hz range, but weak signal energies are typically found within 50-100 Hz at the onset of the events (spectrogram, bandpassed waveforms and amplitude spectrum in Fig. 4c). Apparent velocities of scattered wave packets are typically higher than 1.5 km s⁻¹. P- and S-phases are difficult to identify.

- **Type IV - Local ~~microearth~~micro-quakes**. Signals have duration of 2-10 s and are recorded by the complete seismic network with similar amplitudes (Fig. 3d-e). Successive phases can be tracked consistently over the seismic network with apparent velocity ~~above 2.0~~ranging within 2.0-5.0 km s⁻¹. Dominant frequencies are in the 5-50 Hz ~~(Fig. 4d)~~ but signal onsets generally display energies in the 50-100 Hz (spectrogram, bandpassed waveforms and amplitude spectrum in Fig. 4d). P- and S-phases ~~are difficult~~can be ~~to~~ identif~~y~~ied.

~~Only type I and type II quakes are interpreted as 'slidequake' events, with the meaning they are triggered within or at the edge of the landslide body. Indeed, the short duration of the signals (< 2 s) and the prominent attenuation of the waveforms are evidence of a proximal source. The slow apparent velocities (< 2 km s⁻¹) of trackable wave packets are consistent with velocities estimated for clay rich landslide material (Williams and Pratt, 1996; Tonnellier et al., 2013) and corroborate a source originated within the landslide body. Higher frequency content, shorter duration and few station records of type I events likely reflects a closer and maybe smaller source. Low frequency content and longer duration of type II events may account for slower rupture velocity and maybe larger rupture area.~~

~~The high apparent velocities of trackable wave packets of type IV events as well as the moderate scattering of well distinguishable successive phases is evidence for a source origin outside of the landslide body. Whereas type IV events with duration < 3 s are likely originated very close to the landslide in the host rock, longer duration events must correspond to~~

local (< 10 km) microearthquakes and are therefore not necessarily associated to the landslide deformation. Type III events represent likely a low SNR boundary between type II and type IV signals.

## 4.3 Tremor signals

### 4.3.1 Previous observations

Various tremor-like signals were observed at clay-rich instabilities. Gomberg et al. (1995) and Gomberg et al. (2011) report episodes of tremor-like radiation and sinusoidal waveforms lasting tens of minutes and coherent across the seismic network, which they infer as ETS (episodic tremor and slip) analog of strike-slip faults. A deeper analysis showed that many of these signals feature gliding spectral lines above 50-100 Hz in the spectrogram. Although gliding frequency tremors are known under 20 Hz at volcanoes and inferred to image change in the source properties (e.g. Hotovec et al., 2013; Unglert and

Jellinek, 2015; Eibl et al., 2015 and references therein), gliding harmonics are also characteristic of environmental noise signals produced by moving vehicles such as airplanes or helicopters (e.g. Biescas et al., 2003; van Herwijnen and Schweizer, 2011; Eibl et al., 2015; Eibl et al., 2017). There, the gliding harmonics correspond to the Doppler shift produced by a moving source passing a stationary receiver. At Slumgullion landslide, Gomberg et al. (2011) interpret gliding frequency tremors in the 50-100 Hz range as ~~triggered~~ generated by the action of moving vehicles along a distant (several

15  km) road. ~~and speculate that the saturated state of the landslide may facilitate such high Q wave propagation at remote distances.~~ However, a slide-generated source (slow rupture of faults or materials entrained within the faults like trees or boulders, or slow basal slip) is not excluded for tremor-like radiation devoid of gliding frequency and featuring the highest amplitudes at the seismic network most remote location from the road. These events last several minutes and show dominant energies distributed broadly above 30-50 Hz and diminishing toward the Nyquist at 125 Hz (Gomberg et al., 2011).

At Super-Sauze and Valoria landslides, tremor-like signals lacking clear onsets and with undistinguishable phases were observed with duration of a few seconds to tens of seconds (Walter et al., 2012; Tonnellier et al., 2013; Provost et al., 2017). Spiky, cascading signals are interpreted as rockfalls. Such events feature repeated jolts in the 10-30 Hz that correspond to the rockfall impacts, as well as a 'noise band' in the 30-130 Hz range, likely generated by fine-grain material flows. These events are normally well recorded across the complete seismic network, with moderate waveform attenuation and maximum

amplitudes reaching 1,000-10,000 nm s$^{-1}$. High-frequency tremor-like signals with duration of less than 20 s and maximum amplitudes under 10,000 nm s$^{-1}$, featuring drastic waveform attenuation and thus recorded only partially across the seismic network were also observed (Walter et al., 2012; Tonnellier et al., 2013). Walter et al. (2012) showed that the occurrence rate of these signals correlates well with the measurements of an extensometer installed about a fissure and co-located with a 1-C seismometer at Super-Sauze, July 2009. They concluded that such signals must be triggered by fissure formations at the

surface of the landslide, but also consider scratching and grinding of landslide material against (emerging) hard rock crests as potential source.

### 4.3.~~1~~ 2 Updated classification of tremor signals

As in previous studies, a wide range of tremor-like signals were recorded at SZ10, PG15 and PG16. Medium duration (< 20 s) events are distinguished from long duration, minute-long lasting sequences of tremor-like radiations. While medium duration events feature trackable wave packets consisting of spikes or jolts, minute-long lasting sequences are characterized by sinusoidal waveforms and gentle rumbles, that are difficult to track coherently across the seismic network. Due to the general waveform intricacy and the wide range of observed dominant frequency, finding an unequivocal classification for tremor events is difficult. Based on the literature and searching for consistent observations at SZ10, PG15 and PG16 we propose the following typology of tremor events, where landslide-induced tremor-like signals are distinguished from external sources of tremor-like radiations. Among the landslide-induced events, signals potentially generated by deformation and stick-slip within the landslide body are separated, when possible, from tremor-like signals originating from exogenous landslide dynamics such as rockfalls or small debris flows. Anthropogenic noises can share similarities in waveform amplitudes and in spectral content with landslide-induced tremor signals. It is therefore important to gain knowledge about the characteristics of such events for the manual and automatic detection of landslide-induced tremor signals.

- **ETS-like signals ~~(episodic tremor and slip)~~.** Microseismic signals showing similarities to ETS ~~like~~ signals at strike-slip faults were observed. ETS-like signals at debris slides are emergent and cigar-shaped, last a few seconds and are strongly attenuated across the seismic network (Fig. 5b and 5d, top panel). Dominant frequency of the highest amplitude signals range within 5-50 Hz (spectrogram, bandpassed waveforms and amplitude spectrum in ~~Fig. 5 and~~ Fig. 6a and 6d, top panel). Maximum observed absolute amplitudes reach some 10,000 nm s$^{-1}$; however, most events show amplitudes no higher than a few 100-1,000 nm s$^{-1}$. Phases cannot be identified, instead, the waveforms feature repeating and intricated spikes or jolts with prominent scattering. Individual wave packets which can be tracked return apparent velocity below 2.0 km s$^{-1}$.

- **Confirmed rockfall events.** Signals generated by rockfalls resemble ETS-like signals (compare Fig. 5b and 5d with ~~and~~ Fig. 6b and 6d, top panel). The impacts of falling blocks produce spikes or jolts in the waveforms; loose material saltation and flow combined to the moving character of the source increase waveform intricacy. Signal duration and dominant frequency, as well as waveform attenuation pattern vary significantly depending on the size of the rockfall event and its distance to the recording seismic network. Apparent velocities derived for individual impact signals remain below 2.0 km s$^{-1}$. Because rockfalls are exogenic, potential source areas are known from field observations. In addition, the signal source can eventually be caught by field observations or remote sensing. At SZ10, one ~~ETS like~~landslide-induced tremor signal ~~event~~ could be matched with a single-marl block failure event caught in a high-repetition rate UAV imagery (unmanned aerial vehicle) and optical ground-based images (Rothmund et al., 2017). ~~Since potential source areas of rockfall are known (field observable), near ETS-like signals returning back azimuth towards these areas can be classified as rockfall signals with good certainty. At Pechgraben, the low topography of the monitored area and the extremely wet and muddy conditions of the slope material during~~

the PG15 and PG16 field campaigns preclude the occurrence of large rockfall events. However, small rockfalls consisting of landslide material failure in opening fissures (up to 20-30 cm width, 0,5-1,0 m depth) might occur and produce weak signals recorded at the nearest stations. Nevertheless, many ETS-like signals were detected across the complete network at Pechgraben, stressing the difficulty of characterizing ETS-like signal sources in the absence of additional constraints.

-   **Harmonic tremors.** Signals lasting a few seconds and consisting of harmonic peaks were observed at SZ10, PG15 and PG16 (Fig. 5a, 5c and Fig. 6c). The main harmonic is generally found around 8-10 Hz, followed by several multiples of lower energies (Fig. 6c, amplitude spectrum). Maximum absolute amplitudes do not exceed a few 100-1,000 nm s$^{-1}$, and most signals lie barely above the noise threshold. At SZ10, harmonic tremors were observed only at single sensors. At Pechgraben, such eventsharmonic tremors were detected with various waveform attenuation pattern across the seismic network, suggesting a non-unique source location origin for these signals. Because of the harmonics, apparent velocities are difficult to calculate. For high-SNR signals, apparent velocities calculated with the first arrivals derived velocities of less than 0.7 km s$^{-1}$. At PG15, hHarmonic tremors occur typically in minute-long lasting sequences, alternating with ETS-like signals (Fig. 4a-c). Models to explain harmonic tremors include resonance of fluid/gas driven cracks (e.g. Chouet, 1988; Schlindwein et al., 1995) as well as stick slip (i.e. swarms of small repeating earthquakes) (e.g. Helmstetter et al., 2015; Lipovsky and Dunham, 2016). Both models could reasonably explain harmonic signals observed at clayey landslides.

-   **Dispersive tremors.** Several instances of long duration (few minutes) dispersive tremor-like signals were detected at SZ10, PG15 and PG16. Due to the dispersive character of the signals, the waveforms and spectrograms feature important variations from one station to another, rendering the events difficult to detect (Fig. 7). At the nearest stations, the signals feature high amplitude initial onset (several 10,000 nm s$^{-1}$). The Figure 7a shows an example of a dispersive tremor well recorded across the seismic network at SZ10. The high amplitudes (> 20,000 nm s$^{-1}$) and dominant frequency content above 50 Hz (see spectrogram, bandpassed waveforms and amplitude spectrum in Fig. 7a, top panel) at station S3.7 suggest a source origin close to that station. Then, with increasing distance to the most probable source area (see receiver-source distances indications above the sonograms in Fig. 7a), the signals show prominent dispersion and waveform attenuation with increasing distance to the source. Apparent velocities calculated at the signal onset range within 0.3-0.5 km s$^{-1}$, close to the velocity of sound in the air or velocities in the top most layer of the landslide (e.g. Tonnellier et al., 2013). The Ttemporal evolution of the dominant frequency content of the signals, well observed in the spectrograms of Figure 7 is comparable to signals produced by snow avalanches (e.g. Biescas et al., 2003) or by persons walking about the seismic network (spectrogram in Fig. 8a) and therefore suggest a moving source. Animals can be excluded with good certainty since signals triggered by animals likely show spikier patterns, comparable to human footsteps (Fig. 8a; Fig. 9b). The inferred source area of these events is difficult to access at Super-Sauze and extremely marshy at Pechgraben. No animals or animal traces could be observed there in day time. Debris flows were observed neither in the field nor in daily ground-based and UAV

imagery in the affected areas. ~~At SZ10, a secondary rotational slide was originated near the source area and crown cracks opening was observed during the detection period of the signals. Thus, we postulate rotational sliding initiation and/or opening of crown crack(s) as potential source trigger for these signals. Such a source mechanism would be compatible with field observations made in the potential source area of dispersive events at Pechgraben.~~

— **External sources of microseismic noise and tremor-like radiations.** Shallow installations of the seismometers in clayey materials result in important noise contamination of the seismograms, especially in the high frequency range (> 50 Hz). The variety of events produced by external source of noise is large. Signals range from short to long durations, onsets are usually emergent but sharp onsets can be found for nearby sources. In common to all signals is the absence of identifiable successive phases. Individual wave packets are difficult if not impossible to track. Thus, apparent velocities cannot be calculated. Maximum waveform amplitudes can be high (several 10,000-100,000 nm s$^{-1}$) and waveform attenuation patterns are ambiguous.

- ~~A selection of~~The most common microseismic signals produced by external source of noise ~~is~~ are presented in Figures 8 ~~and Figure~~ 9. ~~Local~~ Nearby (< 50-100 m) moving source such as geophysicists walking about the stations produce long duration spiky tremor radiations (Fig. 9b). Typical of such local moving source is the change towards higher frequency of the dominant energies of the signal as the source (the person walking) is approaching the recording station and the change towards lower frequency content of the dominant energies of the signal as the source is getting further away (sonogram and spectrogram in Fig 8a~~, sonogram and spectrogram panels~~). Distant moving sources such as airplanes and vehicles passing on nearby roads, produce long duration cigar-shaped seismograms (Fig. 7b, ~~Fig.~~ 8b and ~~Fig.~~ 9c) and spectrograms with typical gliding harmonics in the 50-200 Hz range. ~~At PG15 and PG16, frequent timberwork and construction engines passing on the road located at the bottom of the landslide, as well as a local site effect amplifying the acoustic noise produced by airplanes in the valley resulted in important noise contamination of the data in the high frequencies.~~ Beside anthropological noises, many environmental sources of noise were recorded but could not necessarily be distinguished in the absence of additional data at SZ10, PG15, and PG16. Wind bursts, rainfall and storms as well as water streams and bedload transports all produce long duration tremor-like radiations. These events illuminate either several frequencies or only specific ones in the spectrograms (see also Provost et al., 2017). However, the spectrograms are clearly devoid of gliding harmonics (Fig. 8c-d). Maximum amplitudes can reach several 1,000 nm s$^{-1}$ and waveform attenuation pattern across the seismic network is ~~usually~~ typically incoherent (Fig. 9d).

## 5 MicroSseismic source characterization

### 5.1 Source location

Seismic velocities and source location quality can be estimated and verified by calibration shots or hammer blows. Calibration shots and hammer blows were carried out at SZ10 and PG16 and could be located with average accuracies of

about ± 50 m, when using all available first arrivals and back azimuth information with a half-space velocity model. Our results concur with previous results by Tonnellier et al. (2013) at Super-Sauze landslide, where uncertainties of 40-60 m where estimated for calibration shots carried out within the seismic network. It is worth mentioning that this corresponds to the size of the seismic network and scales with the landslide itself. Thus, even if the seismic network is dense, ~~L~~locating

landslide-induced microseismic sources in clayey landslides and discriminating between a source originated within or outside the landslide body is challenging: (1) The velocity structures show ~~important~~ drastic variations in short distances (complex material mélange, topography)~~,~~. and also evolves with time (slope deformation, hydrological state). Velocity models are thus only approximated by tomographic analysis for a specific time (Fig. 10a-b). (2) Scattering and attenuation of the waveforms result in low-SNR onsets ~~and complex signal coda~~ where phases are difficult (if not impossible) to identify.

(3) The seismic network geometry relative to the source is in most ~~(natural)~~ cases not optimal. (4) With an average station spacing of 5-50 m, as it is the case in our study, most landslide-induced microseismic events show no more than four unambiguous phase information.

We use~~d~~ HypoLine (see Section 3) to simulate and analyze graphically the contribution of these parameters on the epicentral location solutions of calibration shots (SISSY, Seismic Source Impulse System, http://www.liag-hannover.de/s/s1/a1/sissy,

last accessed September 13, 2017) at SZ10 (Fig. 10). Three layered $v_P$ velocity models simplified from Tonnellier et al. (2013) featuring both higher and lower velocity contrasts between the landslide material and the sedimentary host rock are tested (Fig. 10a-b; Table 2). For each pair of first arrivals, the time-reversal hyperboles (hypolines) are computed at depth zero. To image the weight of phase uncertainties on the epicenter solutions, all hypolines are also computed for two shifted values of the first arrival by ± five samples (Fig. 10c). An epicenter solution is found at the highest concentration of

hyperboles intersections (see Joswig, 2008 and Vouillamoz et al., 2016 for details). The exercise is carried out for the three velocity models and the resulting epicenter solutions are analyzed for different station combinations. The Figure 10d shows the results obtained when using first arrivals of the three seismic arrays individually. The outcomes of this analysis can be summarized as follow:

- The applied velocity model has low impact on the epicentral solution (few meters) within the considered station

network or in small distances. However, outside of the seismic network, solutions diverge significantly.

- Five samples (±) uncertainties at 1,000 Hz correspond to a high-quality phase onset pick in routine earthquake catalogs (e.g. Diehl et al., 2009). Such high-quality phase onsets derive consistent solutions within the considered station network but the solutions also diverge significantly outside of the considered seismic network.

- First arrivals of natural sources are of lower quality than those of calibration shots (Fig. 10c). Lower quality onsets

have an important impact on the epicentral solutions. At ± 20 samples (± 0.02 s), no more mathematical existent solution is found.~~The domain of existence of the hypolines was tested: it shrinks for higher uncertainties and finally becomes inexistent around ± 20 samples (± 0.02 s)~~.

- The seismic network geometry relative to the source has the most significant influence on the location solution. Whereas the epicenter is resolved with uncertainties of about 20 m when using a set of stations surrounding the

calibration shot (Fig. 10d, central panel), the potential location solutions are biased by 50 m and more when using a station network that do not surround the source (Fig. 10d, left and right panels).

- First arrivals at stations in tripartite configurations derive three zones of high-density hyperboles intersections that cannot be discriminated without additional constraints such as back azimuth information (beam-processing).

- ~~Sources originated within the seismic network return incoherent array-processing and back azimuth data.~~

- Complex velocity structures and resulting waveforms scattering impedes array-processing and back azimuth information can be significantly biased. The calibration datasets at Super-Sauze and Pechgraben derive uncertainties in the order of one quadrant (± 45°) for well constrained beams (using high correlation values of four and more coherent waveform spikes), for source located at 50-100 m outside of the seismic mini-array.

- Sources originated within the seismic network return incoherent array-processing and back azimuth data.

Thus, it can be concluded that approximation in the velocity model, low-quality first arrivals and non-optimal seismic network geometry at clayey landslides result in natural source location uncertainties ranging from tens of meters for sources originated within the seismic network to hundreds of meters for sources originated outside of the seismic network. Consequently, the risk of including biased data in maps of landslide-induced microseismicity and landslide-induced microseismicity rates map is high.

## 5.2 Waveform attenuation pattern to estimate source proximity

Because of the high uncertainties returned by standard seismological approaches to event location, Tthe drastic attenuation of waveforms observed within the landslide body is was used to evaluate constrain the source proximity of near-source area landslide-induced microseismic events to be used in the calculation of events local magnitude. Distance attenuation data of SISSY calibration shots and hammer blows at Super-Sauze and Pechgraben show that signals are strongly attenuated within the first 50 m. The water content of the landslide material influences the waveform attenuation: signals are less attenuated when dryer conditions prevail (Fig. 11a). This observation is consistent with laboratory experiments (e.g. Koerner et al., 1981). ~~The normalized difference (%) between the maximum amplitude of the signal and the median value of all maximum amplitudes for the considered event also show drastic attenuation with distance (Fig. 11b). We refer this percentage value as the scatter about the median amplitude and use it to evaluate the source distance of natural events. Thresholds values of 200 %, 1,000 % and 2,000 % are inferred to image receiver-source distances of 50 m, 20 m and 10 m from the recording station respectively. The source proximity of natural events for which the scatter about the median amplitude values remain below 200 % is considered as uncertain.~~ To quantify the waveform attenuation pattern of an event, we use the scatter about the median amplitude, S, which we compute for each trace that recorded the signal (Eq. (1)):

$$S = \frac{A_{sta} - Med(A_{sta})}{Med(A_{sta})} \times 100\ \% \tag{1}$$

where $A_{sta}$ is the station maximum absolute vertical trace amplitude of the signal in nm s$^{-1}$ and Med($A_{sta}$) is the median value of all $A_{sta}$ where the signal was recorded. S values computed for the calibration dataset of Figure 11a show a drastic

diminution with increasing receiver-source distances (Fig. 11b). Based on these observation, we use maximum S values of landslide-induced microseismic events to approximate receiver-source distances. We infer S values higher than 200 % to correspond to receiver-source distance of less than about 50 m. This is consistent with the observation that local and regional earthquake never return S values above 200 %. At smaller distances, we selected thresholds (in an arbitrary, but very conservative way) of 1,000 % and 2,000 % to correspond respectively to receiver-source distances of about 20 m and 10 m from the recording station. The source distance of natural events for which S values remain below 200 % is considered as uncertain. Among the inferred landslide-induced microseismic events (quakes and tremors), 28 % of events at SZ10, 42 % at PG15 and 39 % at PG16 feature at least one station with a scatter about the median amplitude value above 200 %. With estimated source-receiver distance $\leftarrow$ of less than about 50 m, these events can be reasonably assumed as originated within the landslide body or at its edges and are therefore used in the analysis of landslide-induced microseismicity rates (see Section 6.3) analysis.

### 5.3 Calibrating the local magnitude ($M_L$) scale at clayey landslides

Richter (1958) defines the earthquake local magnitude scale $M_L$ as Eq. (12):

$$M_L = log_{10}(A_{WA}) - log_{10}(A_0) \tag{12}$$

where $A_{WA}$ is originally the half of the maximum peak-to-peak amplitude in microns recorded on a Wood-Anderson (WA) seismograph and $log_{10}(A_0)$ is the distance attenuation function; i.e. a correction applied for the attenuation of the waveforms with distance. The scale is defined so that a $M_L$ 3 earthquake writes a record of 1 mm peak amplitude on a WA seismograph at a receiver-source distance of 100 km. The distance attenuation function of the $M_L$ scale has been calibrated empirically for earthquakes in many regions around the world (e.g. Bakun and Joyner, 1984; Hutton and Boore, 1987; Stange, 2006; Edwards et al., 2015); however, standard calibrated receiver-source distances range within 10-1,000 km (Fig. 11a12a). Therefore, these distance attenuation functions are unappropriated for near-source area microseismic events recordings at landslides. Wust-Bloch and Joswig (2006) calibrated a distance attenuation function within 30-300 m for sinkhole events in the Dead Sea valley. Its slope is very similar to extrapolated distance attenuation function at distances $\leftarrow$ of less than 1 km (Fig. 12b).

Therefore, wWe calibrated $M_L$ in clayey landslides ($M_{L-LS}$) by defining the slope and the intercept of the simplest form of the distance attenuation function (Eq. (23)):

$$log_{10}(A_{0-LS}) = slope \times log_{10}(D) + intercept \tag{23}$$

where $log_{10}(A_{0-LS})$ is the distance attenuation function in landslides and D is the receiver-source distance in km. The slope is defined using the MATLAB® *logfit* function (© 2014, Jonathan C. Lansey), which returns regression in the form $Y = 10^{intercept}X^{slope}$ for the calibration datasets presented in Figure 11a. An average slope value of -1.75 is found for the different regression curves and taken for $log_{10}(A_{0-LS})$ (Fig. 12b).

The intercept of $log_{10}(A_{0-LS})$ is then calculated as follow:

1. The theoretical moment magnitude $M_w$ of a SISSY calibration shot is estimated following the Gutenberg-Richter magnitude energy relation, where $log_{10}(E) = 1.5M_w + 11.8$, E being the radiated seismic energy in ergs. Using E = 240 K~~ilo~~joule (SISSY working principle, http://www.liag-hannover.de/en/s/s1/a1/sissy/project-presentation.html, last visited September 21, 2017), we find $M_{w\text{-SISSY}} = 0.39$.

2. Following Deichmann (2017), we derive $M_L$ of a SISSY shot as $M_{L\text{-SISSY}} = 1.5M_{w\text{-SISSY}} = 0.58$.

3. The intercept of $log_{10}(A_{0\text{-LS}})$ is found using $M_{L\text{-SISSY}} = 0.58$ with the mean slope of the regression curves (-1.75) and the average maximum absolute vertical trace 0-to-peak amplitude of the calibration shots in 1 m distance ($A_{LS} = 5e10^6$ nm s$^{-1}$).

The calibrated local magnitude scale $M_{L\text{-LS}}$ in clayey landslides finally writes as Eq. (3~~4~~):

$$M_{L-LS} = log_{10}(AA_{LS}) + 1.75log_{10}(D) - 0.87$$

(3~~4~~)

where $A_{LS}$ is the maximum absolute vertical trace ~~maximum (vertical trace)~~ 0-to-peak amplitude of the signal in nm s$^{-1}$ and D the receiver-source distance in km.

The calibrated distance attenuation curves are steeper than the average slope of regional earthquakes -$log_{10}(A_0)$ curves (Fig. 12b). However, since no simple relation exist between $A_{WA}$ in ~~µm~~ mm as used in the calculation of standard $M_L$ and $A_{LS}$ in nm s$^{-1}$ as read on a detection trace in landslides, the comparison of standard distance attenuation functions $log_{10}(A_0)$ with $log_{10}(A_{0\text{-LS}})$ is not straightforward. Well imaged in Figure 12b, is the strong influence of various water saturation of the landslide material prevailing during the different calibration measurements, which can result in bias of one order of magnitude or more at distances smaller than 100 m. The range of potential $M_L$ of landslide-induced microseismic events is evaluated in Figure 13. $M_{L\text{-LS}}$ is plotted as a function of the amplitude read in nm s$^{-1}$ using $log(A_{0\text{-LS}})$ for three receiver-source distances (1, 10 and 100 m). Since the estimation of the magnitude has a logarithmic dependence to the receiver-source distance, high uncertainties (> 50-100 m) held in the source location can affect the magnitude calculation by several orders of magnitude units. Considering the range of observed signal amplitudes, the graphic shows that landslide-induced microseismicity must scale within about -3.0 < $M_{L\text{-LS}}$ < 1.0. This agrees with the potential magnitude range that can be inferred from field observations and assumptions, where active seismogenetic structures are expected to fall in the decimeter-meter range.

~~**5.4 Seismicity catalogs at clayey landslides**~~

~~Detected events were gathered into the four main groups described in Section 4 (earthquakes, near quakes, tremors, external source of tremor-like radiations). Since many events feature hybrid characteristics, the distinction between sub-classes of events was often not straightforward. For instance, we observed frequent near quakes (type I and II) featuring short duration harmonics in their coda at Pechgraben. As well, several instances of near quakes doublets, similar to short duration ETS-like signals were observed at both landslides. A systematic template matching analysis as performed in Gomberg et al. (2011) was not carried out. Instead, we cross-correlated the waveforms of individual event classes using a 1-30 Hz bandpass filter.~~

No events sharing high waveform similarity such as the repeating events of Gomberg et al. (2011) were found by this analysis. Because of the high uncertainties held in the source location (Section 5.1), no seismicity maps were produced. Events showing a scatter about the median amplitude above 200 % are considered to be originated in the vicinity (10, 20 or 50 m, see Section 5.2) of the highest amplitude station. Only these events are used in the magnitude catalog to avoid bias in seismicity rate estimations (Fig. 14).

### 5.4.1 Seismicity rates

The Figure 14 shows the temporal $M_{L-Ls}$ distribution of near (< 50 m) landslide induced seismic events relative to displacement and precipitation data. Average displacement rates at Super-Sauze 2010 were much lower than annual average values observed since 1991 (see Figure 5.25 in Travelletti, (2011)). During the SZ10 field campaign, displacement was measured daily at the reference dGNSS station (differential global navigation satellite system) located next to station S2.3. The eight week average displacement in the study area is about 0.4 cm d$^{-1}$. Two main acceleration phases that unevenly follow rainfall events around June 16 and June 26, 2010 reach a maximal displacement of 1.5 and 1.9 cm d$^{-1}$. Near landslide-induced events show temporal clustering which do not necessarily correlate to acceleration peaks. No link is found between the energy radiated by local and regional earthquakes and landslide induced seismicity (Fig 14b). At PG15, an average displacement of about 2 cm d$^{-1}$ was derived on the most active part (array S1) from dGNSS data collected once a week. At PG16, the mean displacement during the campaign was estimated above 20 cm d$^{-1}$ by triangulation, using grids of fixed nails both on the stable and active part of the slide. At both PG15 and PG16, the short durations of the field campaigns combined to the low resolution of the kinematics data prevent us to analyze precisely the potential links between landslide induced seismicity and landslide dynamics (Fig. 14c). Landslide-induced seismic events also cluster in time at Pechgraben: events typically occur in sequences lasting a couple of minutes to hours. Seismicity rates ($M_L > -1$) for the three campaigns show a clear increase with increasing average displacement rates (Fig. 14d).

## 6 Discussion of microseismicity catalogs at clayey landslides

### 6.1 Landslide-induced microseismic events detection and classification

Automatic detection algorithms work fine for well-known routine seismic signatures but fail for unknown and unexpected low-SNR microseismic events. In order to gain knowledge about the existing types of landslide-induced microseismic event signatures, we therefore used an enhanced visualization alternative, where continuous seismic data were screened for visual pattern recognition in the form of sonograms (Joswig, 2008; Sick et al., 2012; Vouillamoz et al., 2016). Using a minimal number of seismic features, detected events could be gathered into three main groups (Section 4): earthquakes, quakes and tremors signals. The shallow installation of seismic stations in the landslide body resulted in high level of noise contamination of the data and the distinction between landslide-induced microseismic events and other environmental (or anthropological) sources was not straightforward. Due to the near-source area of the targeted microseismicity, individual

source signal seismic signature can show tremendous variations between records from different stations, depending on the respective receiver-source distance (e.g. Figs. 3a-b, 5a-c and 7). Despite many landslide-induced microseismic events were observed to occur in sequences, thus suggesting a potential common source process, a cross-correlation analysis performed in the time domain within 1-30 Hz returned no evidence of similar events among the considered sequences. This stresses out the complexity of the signals radiated by near-source area microseismic processes and hence the variability of the related seismic feature space. Individual microseismic sources can also occur simultaneously on a complex debris-slide, thereby leading to time-overlapping tremor signals with hybrid characteristics, where individual source radiations cannot be unambiguously separated. Several instances of quakes doublets (type II and III), similar to short-duration ETS-like signals were observed at both landslides. At Pechgraben, frequent near quakes (type I and II) featuring short duration harmonics were observed. Thus, it can be concluded that an unequivocal manual or automated classification of landslide-induced microseismic signals is possible for well-defined landslide-induced single microseismic events. This is supported by previous results of Provost et al. (2017) at Super-Sauze landslide, where quake and rockfall microseismic events could be successfully detected and classified using a Random Forest supervised algorithm trained with 71 seismic features on a large training set. However, inputs from the analyst are still a requisite in the analysis of complex, hybrid tremor signals recorded at active clayey landslides in order to obtain comprehensive and robust landslide-induced microseismic events libraries for the training of automated detection systems and classifiers.

### 6.2 Landslide-induced microseismic event location and interpretation

Due to the major difficulties encountered in the standard location procedure of landslide-induced microseismic events (Section 5.1), the proximity of near-source area landslide-induced microseismic events was constrained qualitatively and no maps of landslide-induced microseismicity was produced. Events featuring S values above 200 % were inferred to be recorded in receiver-source distance of less than about 50 m, according to calibration tests performed at both landslides (Section 5.2). For these near-source area microseismic events, observations of high-SNR signal spectral content above 50 Hz in the bandpassed waveforms or in the amplitude spectrum corroborated a nearby source.

Quake events are inferred to be generated by a single rupture process. Type I and type II quakes always feature S values above 200 % and signal duration of less than 2 s. Thus, we consider that they are generated in less than 50 m distance. The slow apparent velocities (< 2.0 km s$^{-1}$) derived for trackable wave packets are consistent with velocities estimated for clay-rich landslide material (Williams and Pratt, 1996; Tonnellier et al., 2013) and corroborate a source originated within or at the edge of the landslide body. However, one cannot discriminate between both, mainly because a depth estimation is not possible. S values above 1000 %, higher frequency content, shorter signal duration and few station records of type I events (Fig. 4a and 4e) most probably reflects a small and very nearby source (< 20 m). Low-frequency content and longer duration of type II events may account for slower rupture velocity and maybe larger rupture area (Fig. 4b). Type III events feature S values which are generally below 200% whereas type IV events feature S values which are systematically below 200%. Therefore, type III and type IV quake events likely represent a continuous transition to quake events recorded at larger

receiver-source distances. The higher apparent velocities of trackable wave packets of type IV events and the consistent signal amplitudes of well distinguishable successive phases across the seismic network may suggest a source origin outside of the landslide body in the host rock.

The complexity and frequent hybrid characteristics of observed tremor-like signals makes their interpretation challenging. ETS-like signals were inferred to be generated by stick-slip (near-repeating quakes) at shear boundaries of the landslide or through fissure development or clogging at the landslide surface (e.g. Gomberg et al., 2011; Walter et al., 2012; Tonnellier et al., 2013). The multiple impacts of a rockfall event produces tremor signals, also consisting of spikes and jolts, in some instance very similar to ETS-like tremors. Since potential source areas of rockfall can be observed in the field, multiple-spike microseismic signals returning back azimuth towards such areas can be classified as rockfall signals with good certainty. However, in the absence of additional constraints, an unambiguous classification of rockfall and ETS-like signal can be difficult, in particular when the signals are of low-quality. At Pechgraben, the smooth topography and water-saturated conditions prevailing during the two field campaigns preclude the occurrence of large rockfalls. Small rockfalls consisting of landslide material failure in opening fissures (up to 20-30 cm width, 0,5-1,0 m depth) might happen and produce microseismic signals recorded at the nearest stations. ETS-like signals were mainly recorded across the complete seismic network at Pechgraben. These ETS-like signals occurred in sequences, simultaneously with harmonic tremors. Models to explain harmonic tremors include resonance of fluid/gas driven cracks (e.g. Chouet, 1988; Schlindwein et al., 1995) as well as stick-slip (i.e. swarms of small repeating earthquakes) (e.g. Helmstetter et al., 2015; Lipovsky and Dunham, 2016). Thus, we consider ETS-like and harmonic tremor signal sequences at Pechgraben as being generated by stick-slip episodes. The dispersive tremors seem to be originated near a secondary rotational slide in SZ10, where crown cracks opening was observed during the detection period of the signals. Thus, we postulate rotational sliding initiation and/or opening of crown crack(s) as a potential source trigger for these signals. Such a source mechanism would be compatible with field observations made in the potential source area of dispersive events at Pechgraben.

### 6.3 Landslide-induced microseismicity rates

Only near-source area quakes type I and II and tremors events (ETS-like, rockfall, harmonic and dispersive) with S > 200% were used in the final catalog of landslide-induced microseismic events. This catalog served as a basis to evaluate landslide-induced microseismicity daily rates, in accordance with the resolution of available displacement datasets (see Section 2). The Figure 14 shows the temporal $M_{L\text{-}LS}$ distribution of the near-source area landslide-induced microseismic events for SZ10 (a), PG15 (b) and PG16 (c) and the cumulated events curves with $M_{L\text{-}LS} > -1$ (d). The corresponding daily landslide-induced microseismicity rates, for $M_{L\text{-}LS} > -1$ and $M_{L\text{-}LS} > 0$, show a clear increase with increasing average daily displacement rates of the three passive microseismic field campaigns (e). At both landslides, no temporal links were found between the energy radiated by local and regional earthquakes (maximum vertical trace absolute amplitude) and the occurrence of landslide-induced microseismic events. At all campaigns, temporal clustering of near-source area landslide-induced microseismic events was observed. Sequence typically last a few minutes to a few hours and are followed by quiescent times.

## ~~6~~7 Conclusion and ~~o~~Outlook

We propose a unified typology of microseismic signals observed at slow-moving clay-rich debris slides by comparing passive microseismic recordings of three campaigns carried out at two landslides and using published similar case studies as a benchmark. The highly heterogenous and water-saturated state of the material within the slides result in strongly attenuated
and scattered waveforms. Signals generally consists of complex and intricated surface waves, where P- and S-phases cannot be clearly distinguished and successive phase (or wave packets) onsets are difficult, if not impossible to pick. Therefore, very simple waveform and spectral attributes of the signals are used for the classification (see list in Section 3.1). The principal discriminating parameters we find to differentiate landslide-induced microseismic signals from external unrelated sources are (1) the prominent and consistent waveform attenuation of near-source area events ~~(< ~50 m) sources~~ across the
recording seismic network (Section 5.2) and (2) the low apparent velocity (< 2 km s$^{-1}$) of trackable wave packets, that also applies for landslide-~~related~~ induced signals generated in 50-500 m (estimated) receiver-source distances. Despite the complexity of the waveforms, comparable landslide-induced microseismic signals were detected at both landslides, thereby suggesting that similar microseismic source processes are taking places at different landslides and that the method is therefore scalable and reproducible. Two main classes of landslide-induced signals ~~are~~ were found: (1) quake-like signals
('slidequakes') ~~and~~ (2) a variety of tremor-like signals (Sections 4.2-4.3). Because ~~of complex seismic velocity structures, low SNR signal onsets and non-optimal network geometry, source location uncertainties scale at best with the size of the used seismic network. Errors in estimated receiver-source distance also adversely affect event magnitude estimation~~ standard approaches to event location result in high location uncertainties at clayey landslides, ~~.~~ ~~Consequently, it is difficult (and speculative) to characterize source processes and evaluate seismicity rates. Using~~ waveform attenuation pattern were used to
better ~~estimate~~constrain receiver-source distances. Since S values of local and regional earthquake stay systematically below 200 %, we did not correct for potential site-effects and~~, we nevertheless computed~~ M$_{L\text{-LS}}$ were computed for near-source area events ~~signals~~ (< ~~~~about 50 m), applying a distance attenuation function calibrated for clayey landslides (Section 5.3). Results show an increase of landslide-induced microseismicity rates with higher average displacement rates. Although much attention was paid to derive unbiased magnitude catalogs, uncertainties are still high. In addition, the catalogs may be
incomplete in the lower magnitude range due to incomplete datasets (see Section 2). Consequently, we did not derive b-values.

Since passive seismic methods alone do not allow a detailed characterization of microseismic source processes taking place at clayey landslides, it is recommended to supplement seismic data with high spatial-temporal resolution remote sensing, geodetic, geotechnical, geophysical, meteorological and hydrological measurements. One inconvenient is that ground-based
measurements on the landslide during the day result in high anthropological noise level, corrupting a significant part of day-time seismic measurements, when other measurements are available. The seismic monitoring of ~~Super-Sauze 2010~~SZ10 and PG16 ~~was~~were part of ~~a~~ multi-disciplinary field experiment~~s~~ and ~~.~~ ~~Dynamics of the landslide could be derived from daily UAV orthomosaics, ground-based optical images and dGNSS measurements over a dense grid covering the seismic arrays~~

(Rothmund et al., 2017). In one promising instance, a single marl-block failure event could be matched both in the imagery and the seismic datasets. Other dynamics like displacement or fissure development (opening and clogging) could also be observed. Intrinsic differences in spatial and temporal resolution of the various datasets (lower sample rate but higher spatial resolution of remote sensing, kinematics and geotechnical measurements in comparison to the instantaneous nature but high spatial uncertainty of seismic detections) generally prevent to link with certainty seismic detections to remote sensing observations. We nevertheless consider that multi disciplinary approaches can bring key additional constraints to better understand landslide induced seismicity.

Ffuture directions of this study involve a detailed comparison of the various datasets at Super-Sauze. The aim will be to precisely evaluate the degree to which the main limitation of passive seismic monitoring (high spatial uncertainty of the detected microseismic events and hence speculative sources characterization) can effectively be compensated by remote sensing and other geodetic and geotechnical information. Similarly, at Pechgraben, data acquired by a GB InSAR (ground-based interferometry synthetic aperture radar) together with one seismic array during a highly active one-month period in November December 2016 by the EOST, University of Strasbourg, France, will be analyzed and benchmarked to seismic data with the aim to improve our understanding of the mechanisms driving brittle deformation in clayey landslides.The landslide-induced microseismic event catalog also provides an initial signal library to train future automatic detection systems and classifiers of complex and hybrid microseismic signals at clayey landslides. In addition to Random Forest supervised classifier already implemented by Provost et al. (2017) at Super-Sauze, unsupervised pattern recognition (e.g. Sick et al., 2015) or Hidden Markov Models (e.g. Hammer et al., 2012; Hammer et al., 2013) should be tested and success rate as well as method reproducibility and scalability benchmarked.

**Data and resources**

The Super-Sauze and Pechgraben microseismic datasets used in this study are stored at the Institute of Geophysics of the University of Stuttgart, Germany, in SEG-2 and MSEED data format. Request to these data can be addressed to the authors. Computations and plots were done with MATLAB® (www. mathworks.com/products/matlab, last accessed November 10, 2017) under a campus license of the University of Stuttgart. The MATLAB spectrogram code and colormap was provided by Clément Hibert of the EOST, University of Strasbourg, France.

**Acknowledgements**

This work was funded by an early postdoc mobility fellowship of the SNSF (Swiss National Science Foundation, grant P2FRP2_158749). Birgit Jochum, David Ottowitz and Robert Supper of the Geological Survey of Austria in Vienna are warmly acknowledged for sharing datasets, as well as providing insightful tips and help for the field work in Pechgraben. Jon Mosar of the Institute of Earth Sciences of the University of Fribourg, Switzerland is thanked for lending seismometers

and dataloggers for the Pechgraben seismic campaigns. The authors are very grateful to Clément Hibert, Jean-Philippe Malet and Floriane Provost of the EOST, Strasbourg University, France, for fruitful inputs in this project as well as for sharing datasets and codes. The authors thank Marco Walter, Ulrich Schwaderer and Patrick Blascheck of the IfG (Institute for Geophysics) of the University of Stuttgart for their support at Super-Sauze 2010 seismic monitoring campaign and their help

5     in the seismic data pre-process. Juan-Carlos Santoyo Campus (IfG, University of Stuttgart) is warmly thanked for his participation to the 2015 Pechgraben field campaign.

**Tables and Figures**

**Table 1. Seismic features of microseismic signal types detected at slow-moving clay-rich debris slides. Features are indicated for high-SNR high-energy signals.**

| | Signal duration | Signal onset | Attenuation pattern | Dominant frequency | Number of recording station | Max. amplitude (order in nm s$^{-1}$) | S [%] |
|---|---|---|---|---|---|---|---|
| **Earthquakes** | | | | | | | |
| Local/Regional | ~10-60 s | impulsive | none | 1-20 Hz | all | 10,000 | < 200 |
| Tele | ~~60      s~~ minutes | emergent | none | < 5 Hz | all | 100 | < 100 |
| **Quakes** | | | | | | | |
| Type I Near high frequency | <1 s | impulsive | clear | 20-100 Hz | < 5 | 1,000-10,000 | 200-10,000 |
| Type II Near low frequency | 1-2 s | impulsive | clear | 5-50 Hz | all | 1,000-10,000 | 200-10,000 |
| Type III Moderate distance | ~2 s | impulsive | ambiguous | 5-50 Hz | all | 1,000 | < 200 |
| Type IV Local micro~~earth~~quake | 2-10 s | impulsive | ambiguous | 5-50 Hz | all | 1,000 | < 200 |
| **Tremors** | | | | | | | |
| **Landslide-induced tremor-like signals** | | | | | | | |
| ETS-like | <20 s | emergent | clear | 5-50 Hz | <5-all | 1,000-10,000 | 200-10,000 |
| Confirmed ~~R~~rockfall | 5-10 s | emergent | clear | 5-100 Hz | <5-all | 100-10,000 | 200-10,000 |
| Harmonic | <5 s | emergent | clear | 5-20 Hz | <5 | 100-1,000 | 200-1'000 |
| Dispersive | 30-120 s | emergent | clear | 50-250 Hz | <5-all | 10,000 | 200-100'000 |
| **External source of tremor-like radiations** | | | | | | | |
| Footsteps | 5 s-minutes | emergent | clear | 5-100 Hz | < 5-all | 10,000 | > 200-10,000 |
| Gliding frequency | 20 s-minutes | emergent | none | 50-100 Hz | all | 1000 | 100-1,000 |
| ~~Meteorological~~Environmental | 20 s-minutes | emergent | ambiguous | 20-250 Hz | all | 10,000 | > 200-10,000 |

**Table 2. Three simplified layered $v_P$ velocity models at clayey landslides.**

| Layer thickness (m) | Model 1 | Model 2 | Model 3 |
|---|---|---|---|
| 10 | 0.4 km s-1 | 0.65 km s-1 | 0.8 km s-1 |
| Half-space | 2.3 km s-1 | 1.5 km s-1 | 2.3 km s-1 |

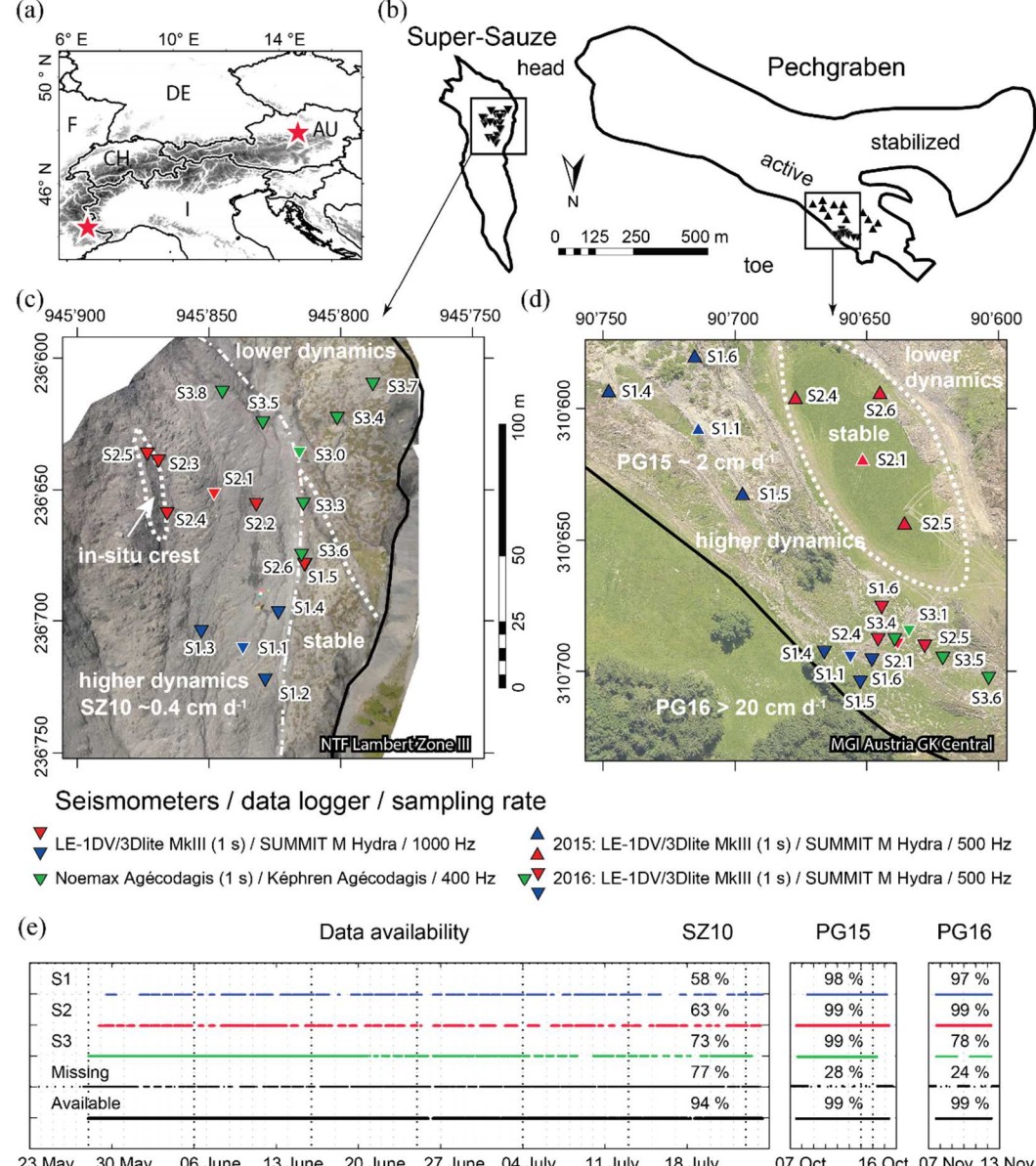

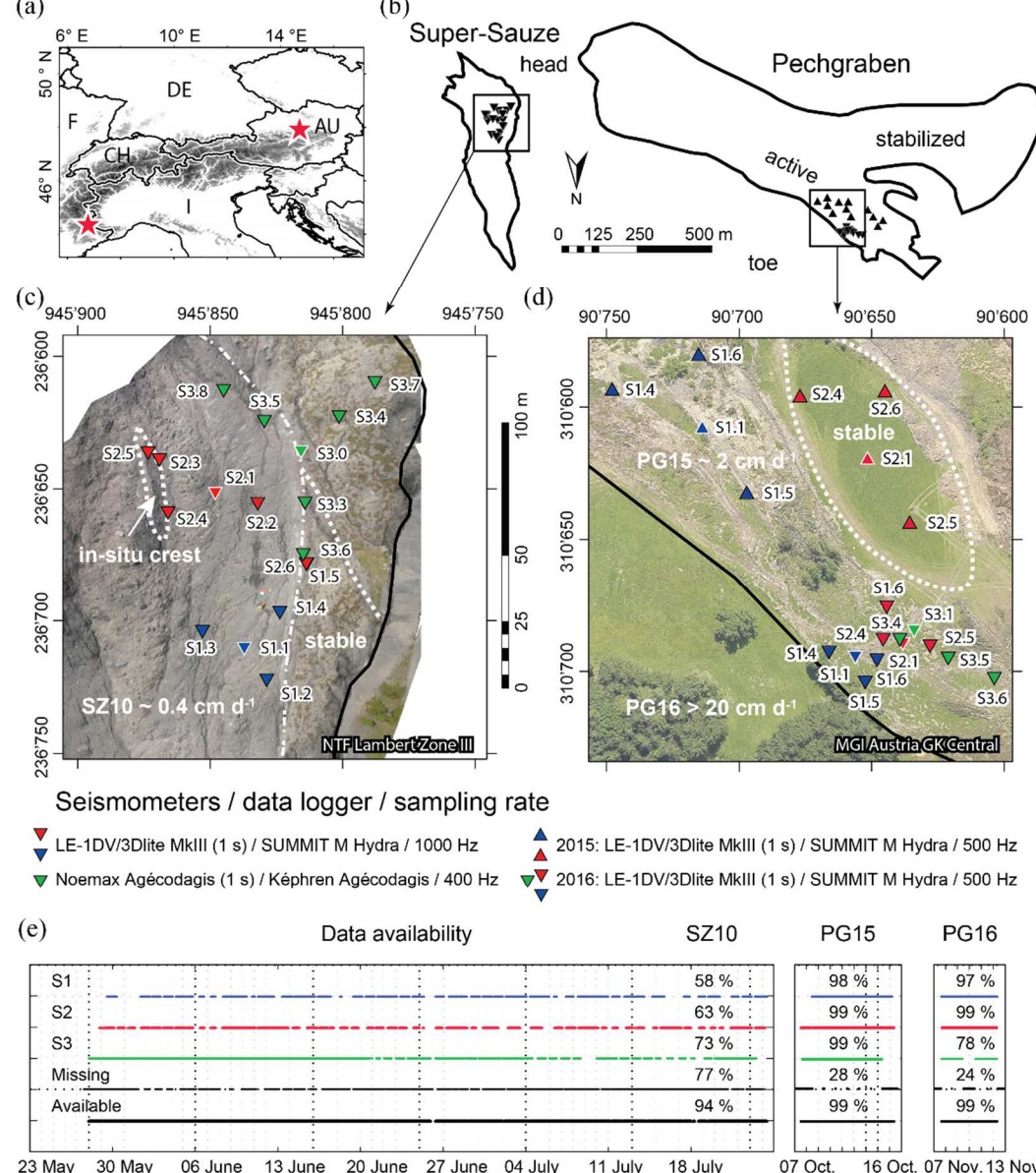

**Figure 1. Data overview. (a) Location of Super-Sauze (Southeastern France) and Pechgraben (Upper Austria) clayey landslides (stars). (b) Orthogonal projection of Super-Sauze and Pechgraben instabilities with situation of instrumented areas. (c-d) Zoom into Super-Sauze ~~seismic network in 2010 (SZ10). (d) Zoom into~~and Pechgraben seismic networks ~~deployed in 2015 (PG15) and 2016 (PG16)~~, where triangles indicate the seismic stations and colors refer to different tripartite arrays (S1, blue; S2, red; and S3, green). ~~In (c-d), zones of higher and lower dynamics and t~~The average underline{daily} displacement rates prevailing during individual field campaigns are indicated in white text; white dashed lines indicate main subparts of the landslide and black bold lines show the limits of the landslides~~by white dotted lines~~. 3-C seismometers (S1.1, S2.1, S3.0 and S3.1)~~triangle symbols~~ are highlighted by white outlines. Orthophotos credits: Super-Sauze, Rothmund et al. (2017); Pechgraben, Lindner et al. (2014); Lindner et al. (2016). (e) Data records availability for individual seismic arrays ~~(S1, S2 and S3)~~ based on 2 minutes data segments. The *missing* line indicates incomplete records (measurements from one or two arrays are missing); the *available* line shows where at least one array is recording.**

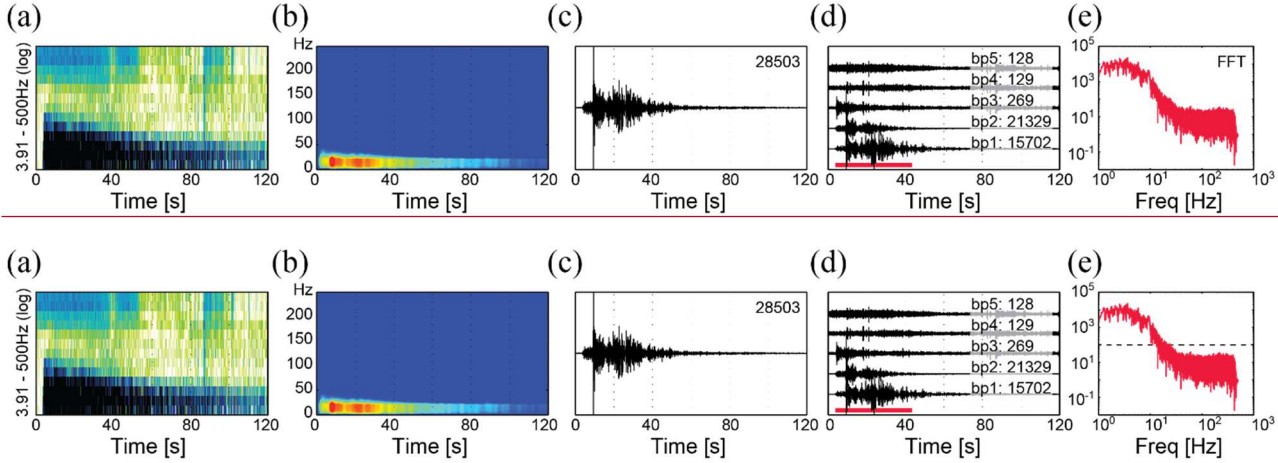

**Figure 2. Seismic features of an earthquake.** Regional event (in 110 km distance) of June 30, 2010 11:53 with $M_L$ 4.3 at Saint-Jean-de-Maurienne, France, recorded at SZ10, station S2.1 at $t_0$ 2010.06.30 11:54:00. (a) Sonogram ~~(log Hz): A typical sonogram pattern for an earthquake maximizes signal onset and enhances changes in distribution of signal energy as a function of time and frequency~~. (b) Spectrogram (0-250 Hz~~(window: signal length/30; overlap: 90 %; colormap and code provided by Clément Hibert, EOST (Ecole et Observatoire des Sciences de la Terre), University of Strasbourg, France~~). (c) Unfiltered seismogram (nm s$^{-1}$). (d) Bandpass filtered seismograms (nm s$^{-1}$, bp1: 1-5; bp2: 5-20; bp3: 20-50; bp4: 50-100; bp5: 100-200 Hz). (e) Amplitude spectrum ~~Fast Fourier Transform~~ (FFT, nm Hz$^{-1}$). This layout is applied to all figures presenting the microseismic signals classification. Time indication is always UTC. Waveforms absolute maximum 0-to-peak amplitudes are indicated in nm s$^{-1}$ above the seismograms in (c) and (d). The signal window for which the FFT is computed is indicated by the red horizontal line in (d).

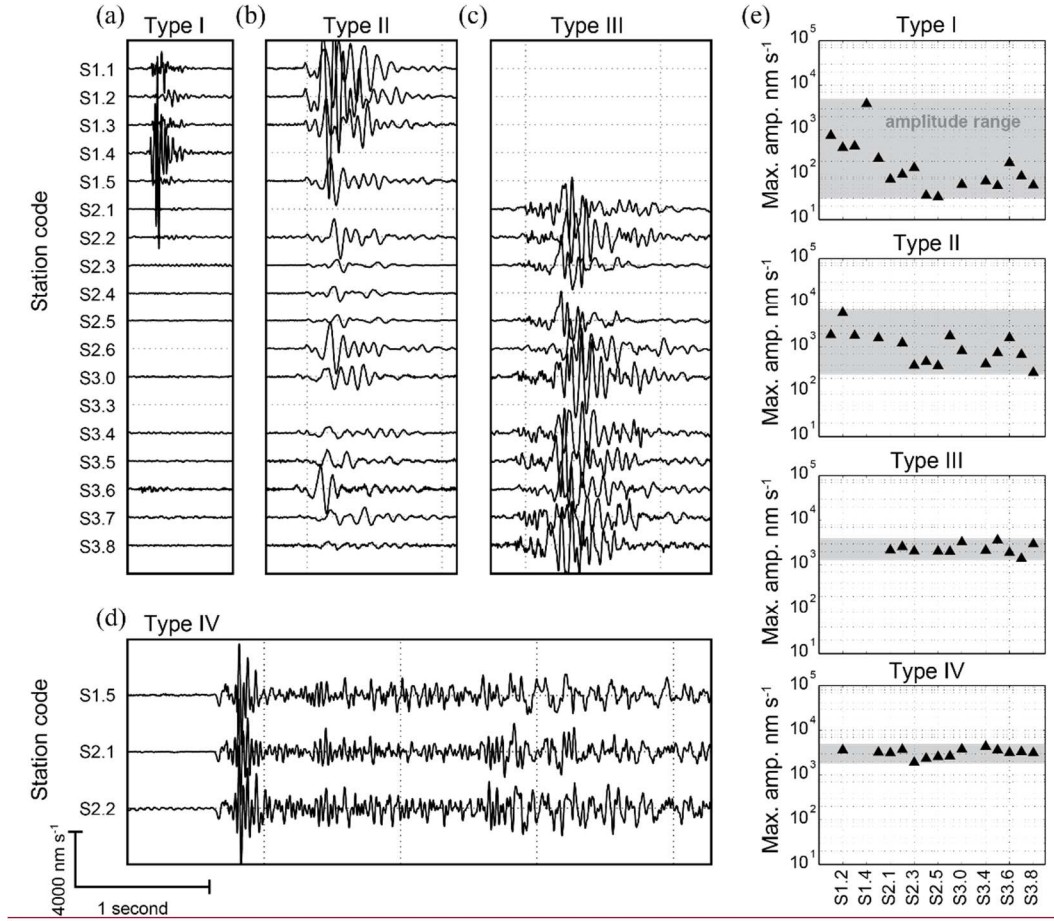

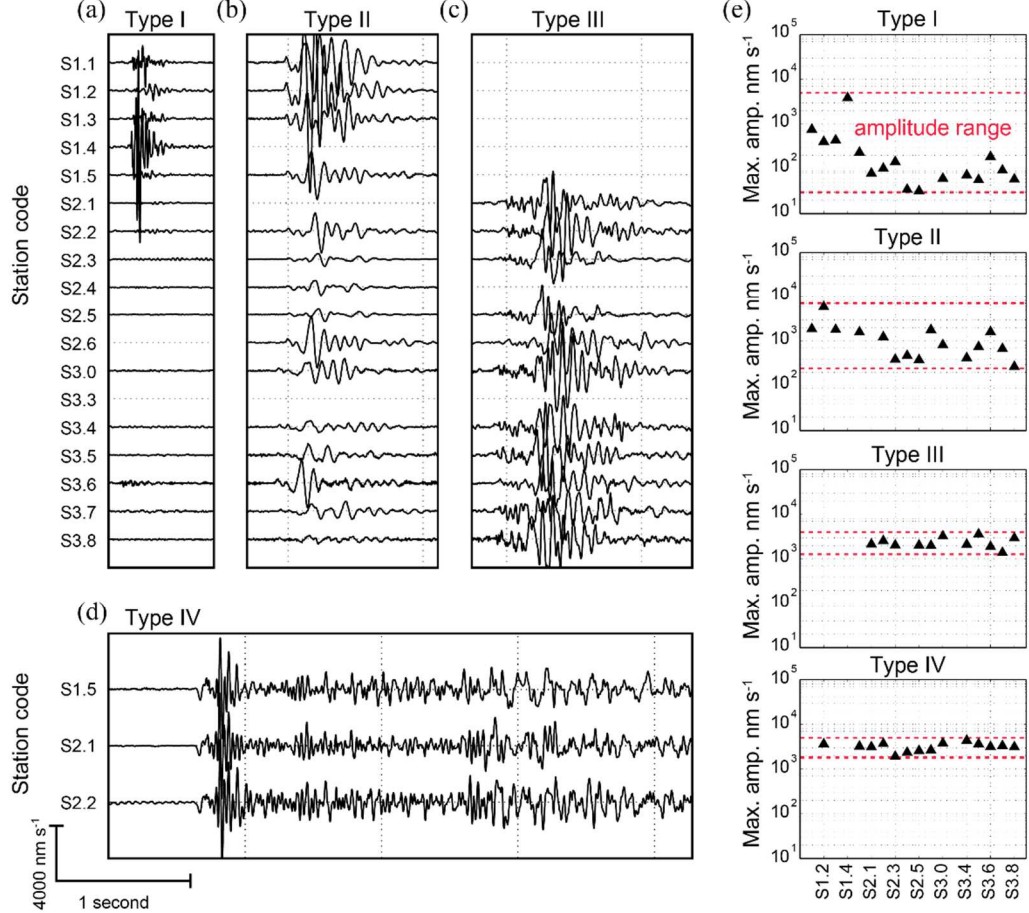

**Figure 3. Vertical trace seismograms of quake events recorded at SZ10 (see station location and nomenclature in Figure 1, empty traces correspond to missing or corrupted records). A fixed amplitude and time scale is applied to all waveforms (bottom left). (a) Near high frequency quake type I (May 29, 2010, 23:05:05). (b) Near low frequency quake type II (June 26, 2010, 18:44:55). (c) Moderate distance quake type III (June 17, 2010, 15:32:45). (d) Local micro~~earth~~ quake type IV (June 7, 2010 11:24:29). Note the highly coherent successive phases and moderate scattering. (e) Maximum amplitudes (log nm s$^{-1}$) recorded at individual stations for the four events displayed in a-d. Large amplitude range~~s~~ (i.e. important waveform attenuation, indicated by dashed red lines~~grey rectangles~~) enable to discriminate events types I and II from events type III and IV which typically feature narrow amplitude ranges.**

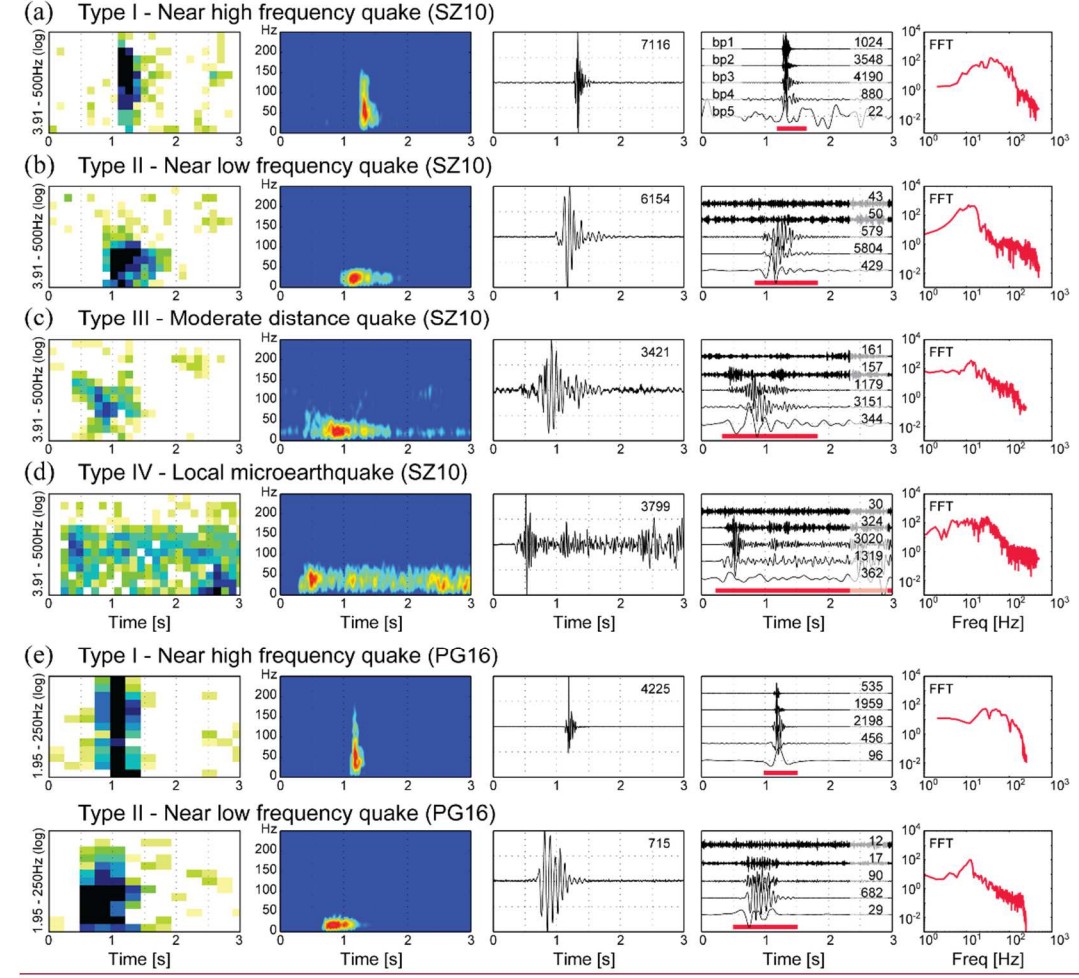

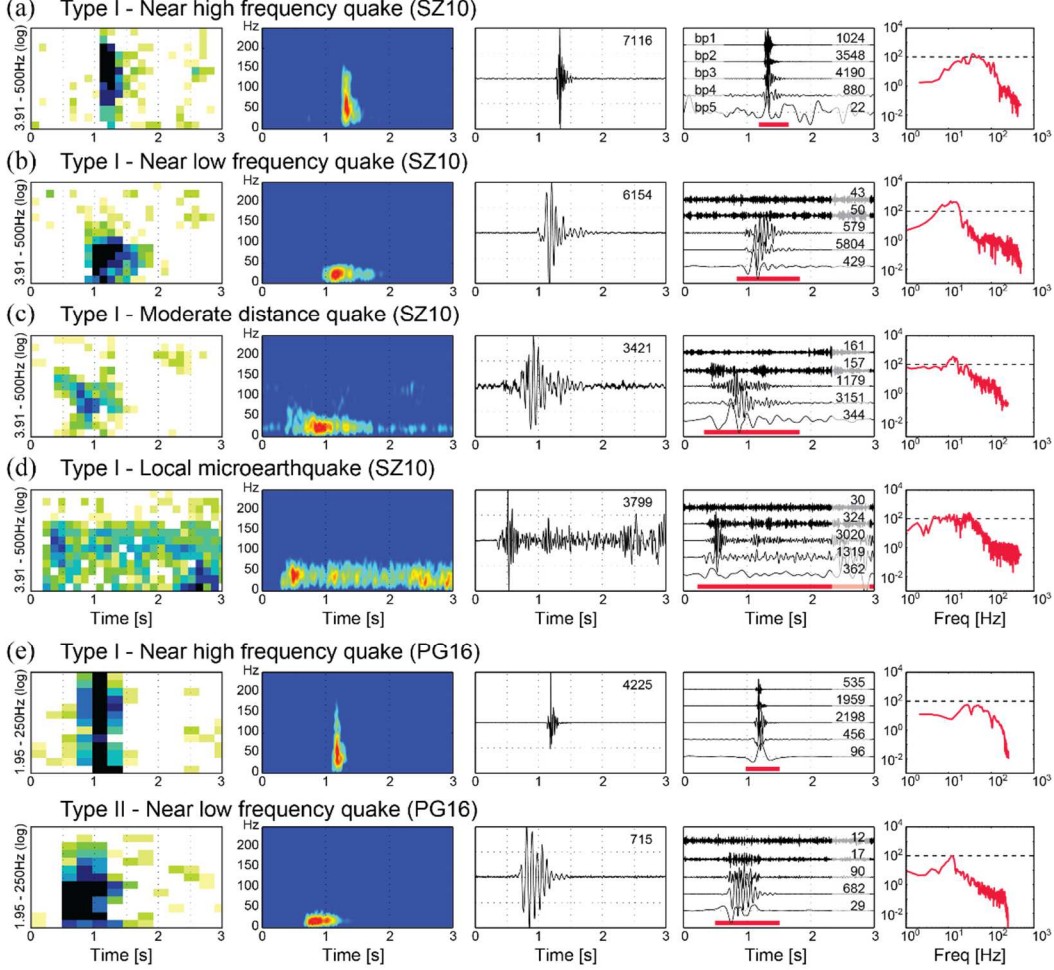

**Figure 4. (a-d) Seismic features of the highest SNR/amplitude trace of events presented in Figure 2 and corresponding to type I-type IV quakes. (a) Type I, SZ10, S1.4, $t_0$ 2010.05.29 23:05:04. (b) Type II, SZ10, S1.2, $t_0$ 2010.06.26 18:44:55. (c) Type III, SZ10, S3.0, $t_0$ 2010.06.17 15:32:45.500. (d) Type IV, SZ10, S2.1, $t_0$ 2010.06.~~17~~ 07 11:24:29.300. (e) Example of near quakes recorded at Pechgraben. Top: Type I, PG16, S2.6, $t_0$ 2016.11.07 22:43:05.500. Bottom: Type II, PG16, S1.4, $t_0$ 2016.11.09 01:50:13.**

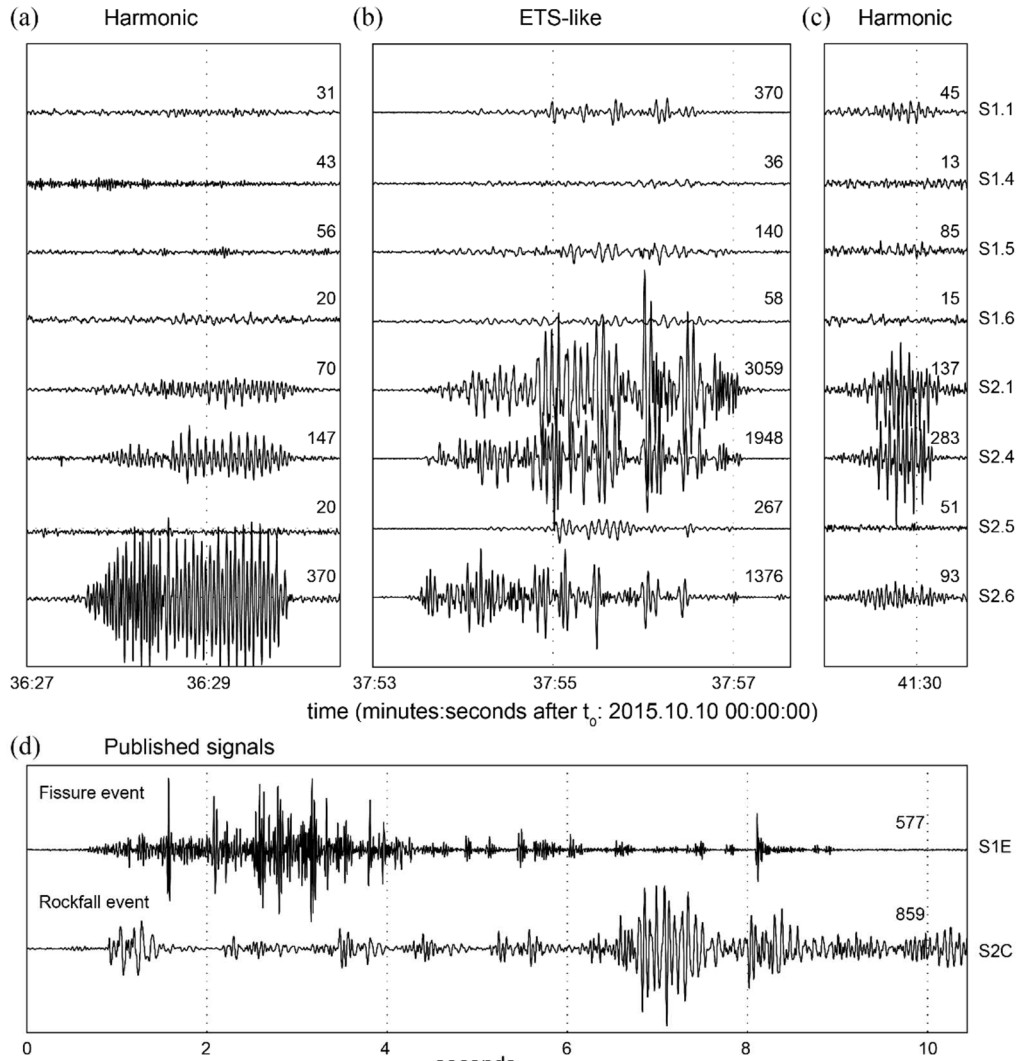

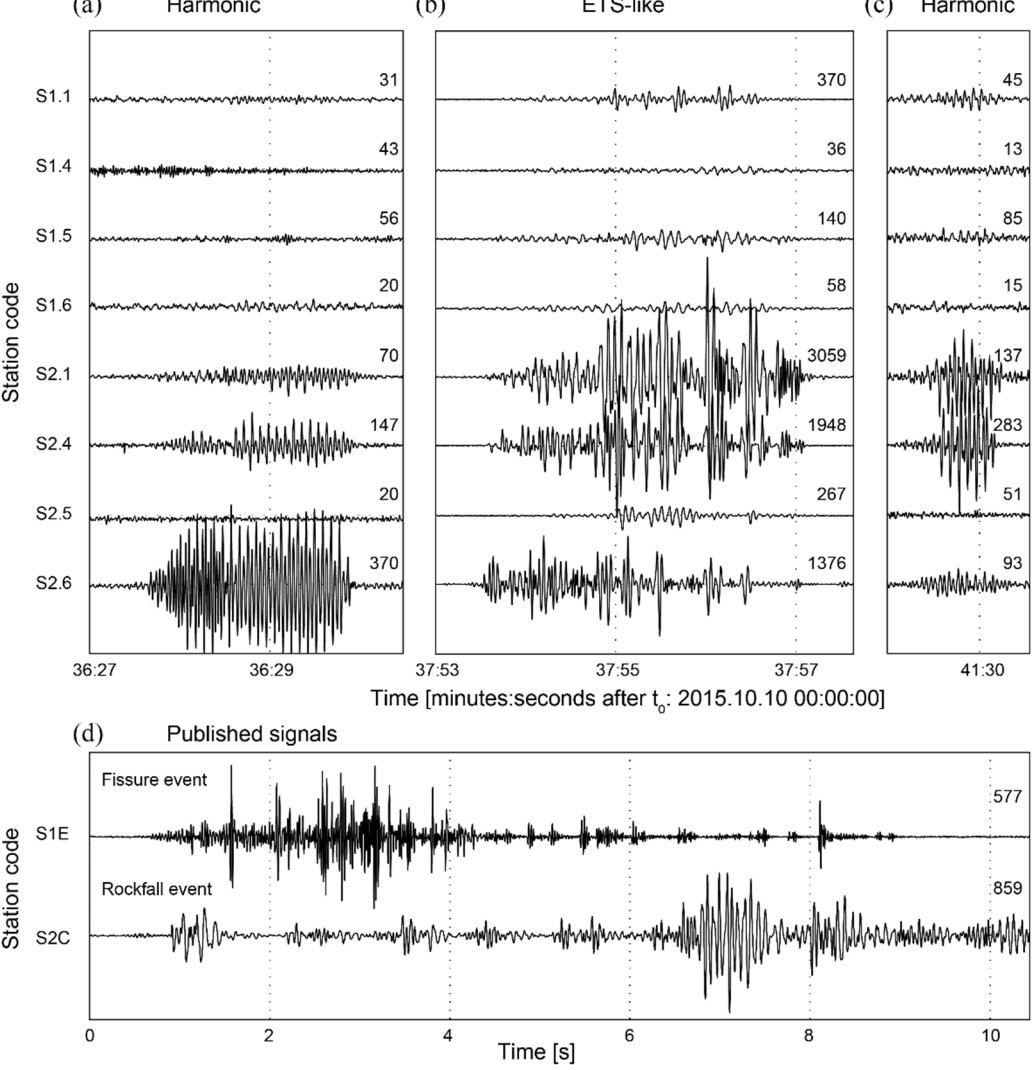

**Figure 5. (a-c)** Vertical trace seismograms featuring selected signals of a 40 minutes long tremor sequence recorded October 10, 2015 between 00:35 and 01:15 at PG15. See station nomenclature in Figure 1. Waveforms are normalized to the highest amplitude trace of individual events and maximum **absolute** 0-to-peak amplitudes are given in nm s$^{-1}$ on top of each seismogram. Event (a) and (c) are harmonic tremors, event (b) corresponds to an ETS-like event. Note the prominent attenuation of the waveforms and the relatively lower amplitudes of harmonic tremors. **(d)** Signals published in Walter et al. (2012) and interpreted as a fissure event (top, $t_0$ 2008.07.14 23:48:40) and a rockfall event (bottom, $t_0$ 2008.07.14 23:49:04). ~~For better comparison, w~~Waveforms are plotted using the same time scale as in a-c to facilitate the signal comparison.

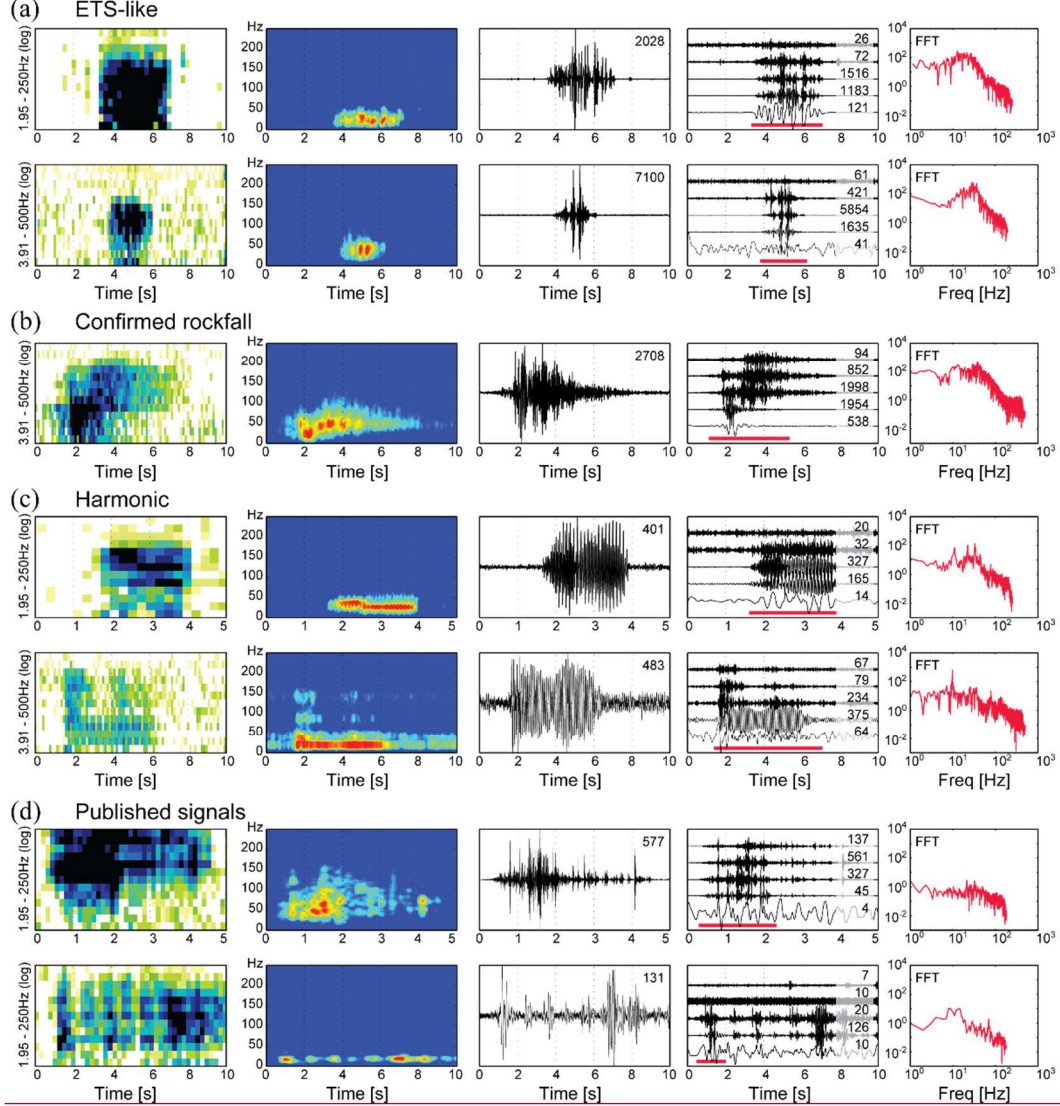

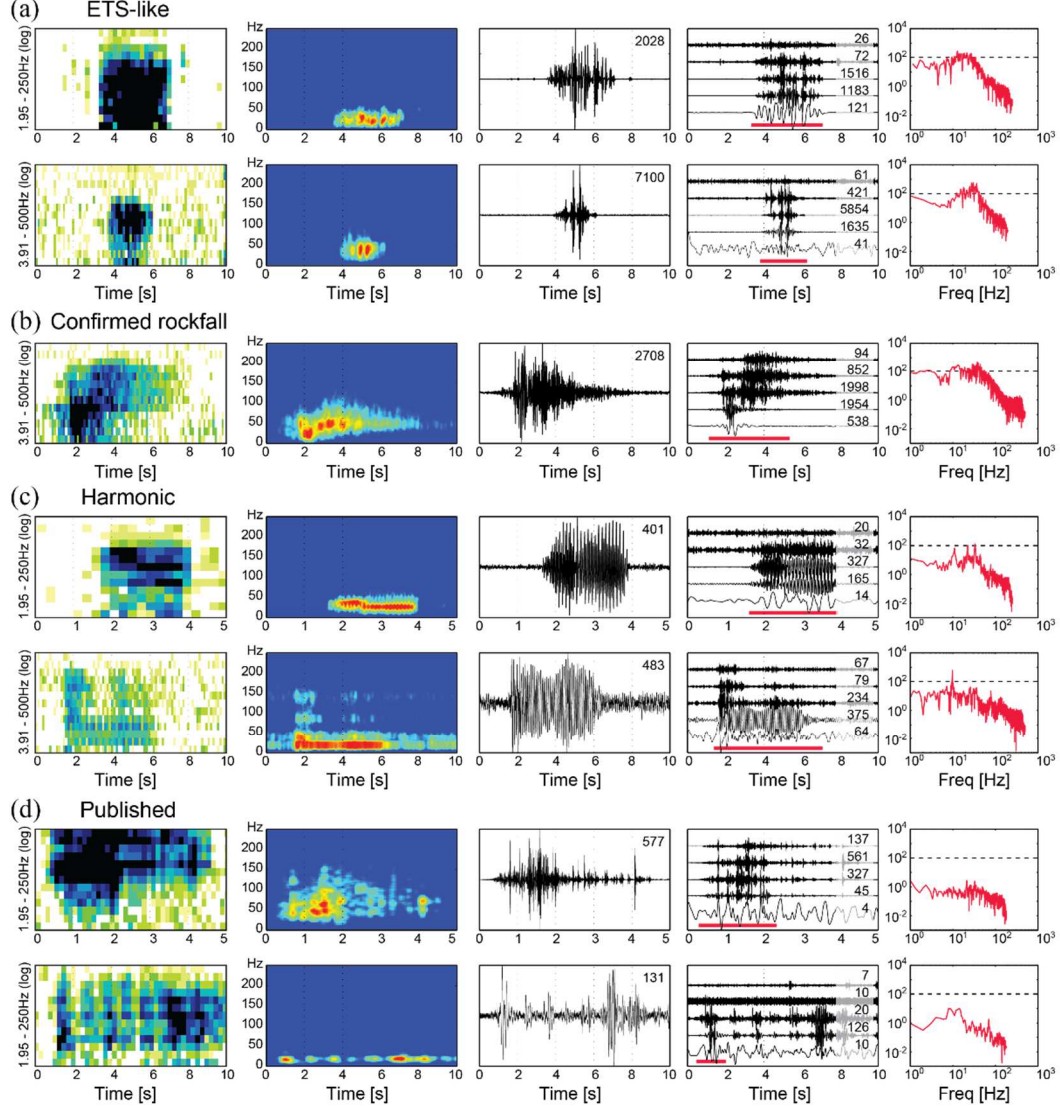

**Figure 6. Seismic features of moderate duration (< 20 s) tremor signals recorded at Super-Sauze and Pechgraben. (a) ETS-like events. Top: PG15, S2.4, t₀ 2015.10.10 00:37:50. Bottom: SZ10, S3.4, t₀ 2010.06.05 15:26:35. (b) Confirmed rockfall event at receiver-source distance of 29 m (Rothmund et al., 2017), SZ10, S2.5, t₀ 2010.06.04 06:45:20. (c) Harmonic tremors. Top: PG15, S2.6, t₀ 2015.10.10 00:36:26. Bottom: SZ10, S2.3, t₀ 2010.06.04 20:07:28. (d) Published tremor signals by Walter et al. (2012). Top: Fissure event, t₀ 2008.07.14 23:48:40. Bottom: Rockfall event, t₀ 2008.07.14 23:49:04.**

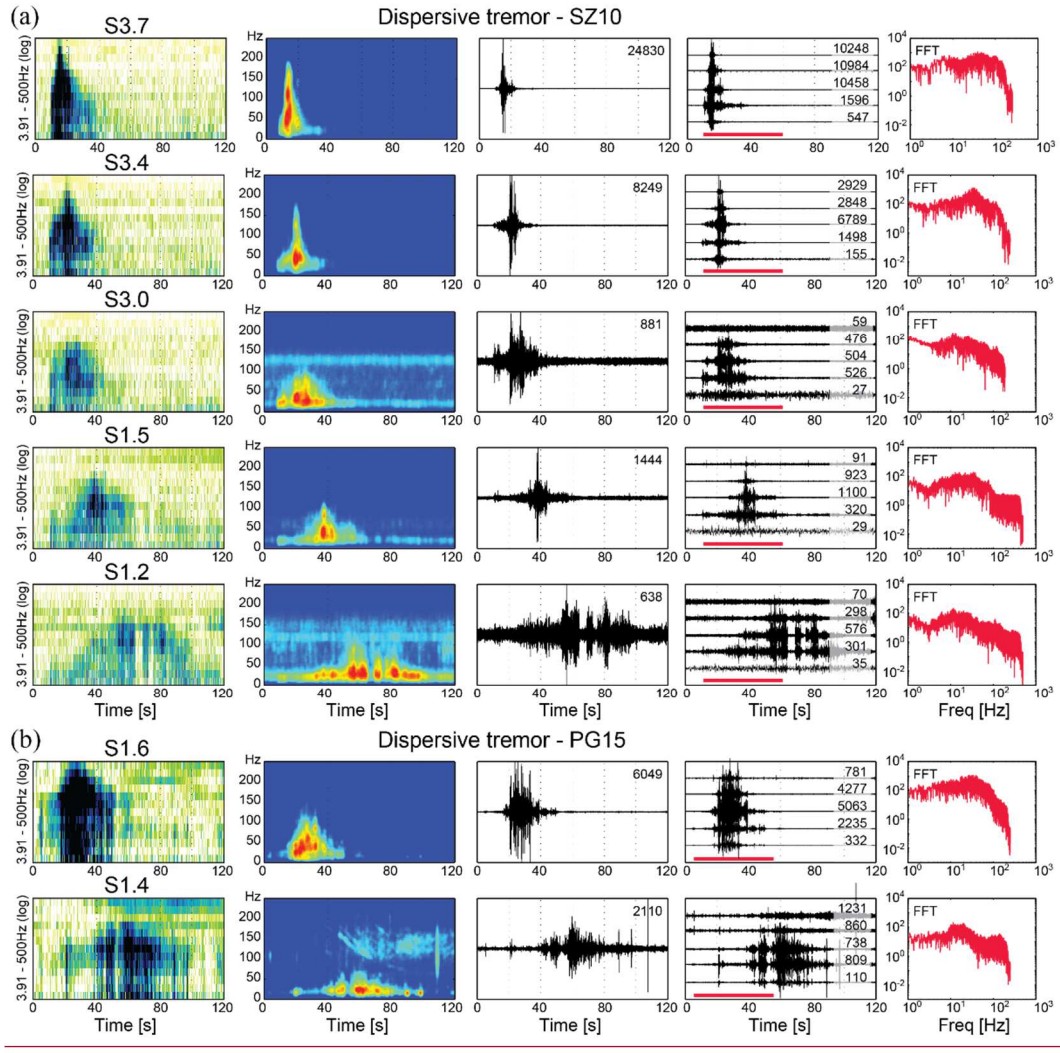

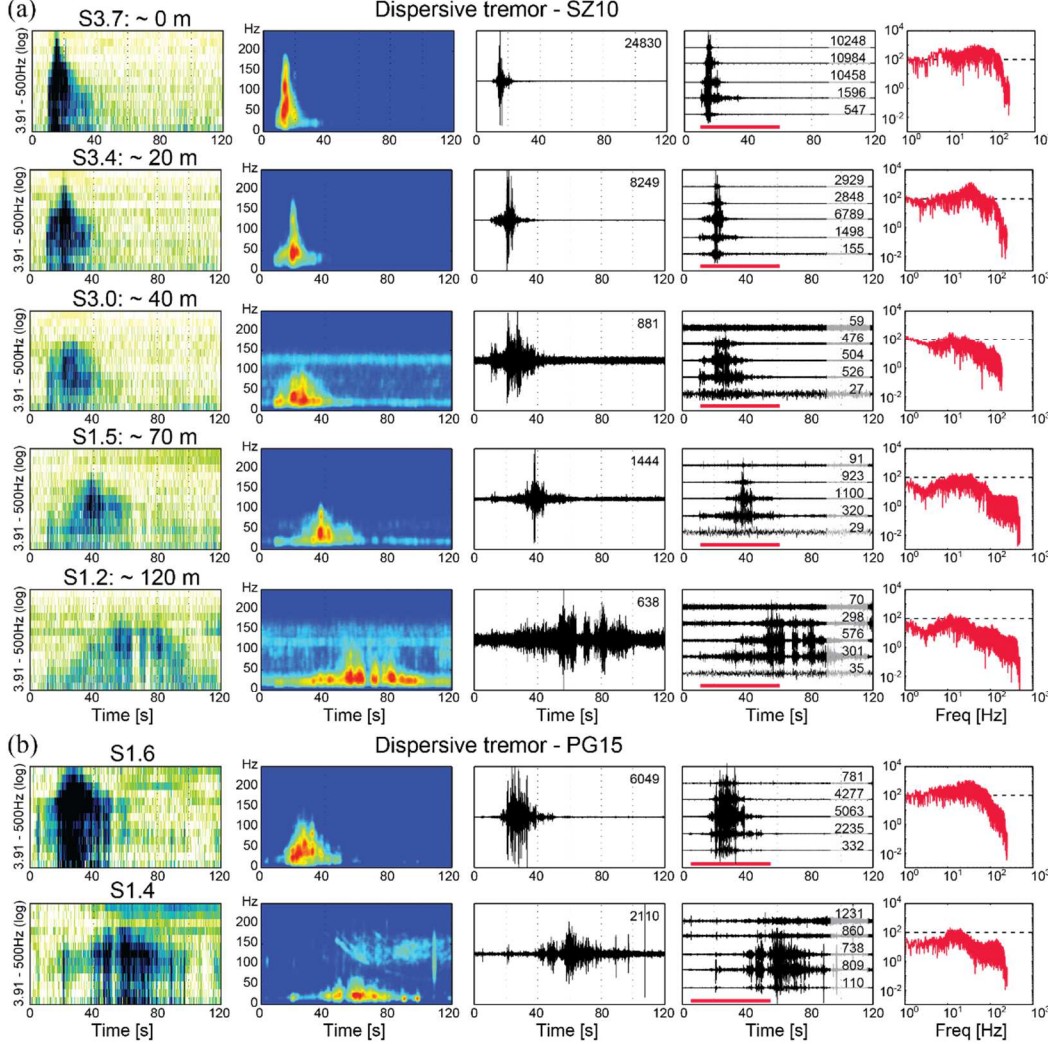

**Figure 7. Seismic features of two dispersive tremor events recorded at (a) SZ10 at t₀ 2010.07.04 00:45:20 and (b) PG15 at t₀ 2015.10.08 18:02:08. Stations are indicated on top of the sonogram panels and displayed in (a) and (b) from top to bottom with increasing inferred distance to the most probable source area (SZ10 stations S3.7 and S1.2 are about 120 m distant; PG15 stations S1.6 and S1.4 are about 50 m distant, the receiver-source distance could not be estimated). Note the noise contamination by an airplane (gliding harmonics in the spectrogram) well visible at PG15 station S1.4. The airplane signal was well recorded by the complete seismic network, whereas the dispersive event is only seen at array S1 stations.**

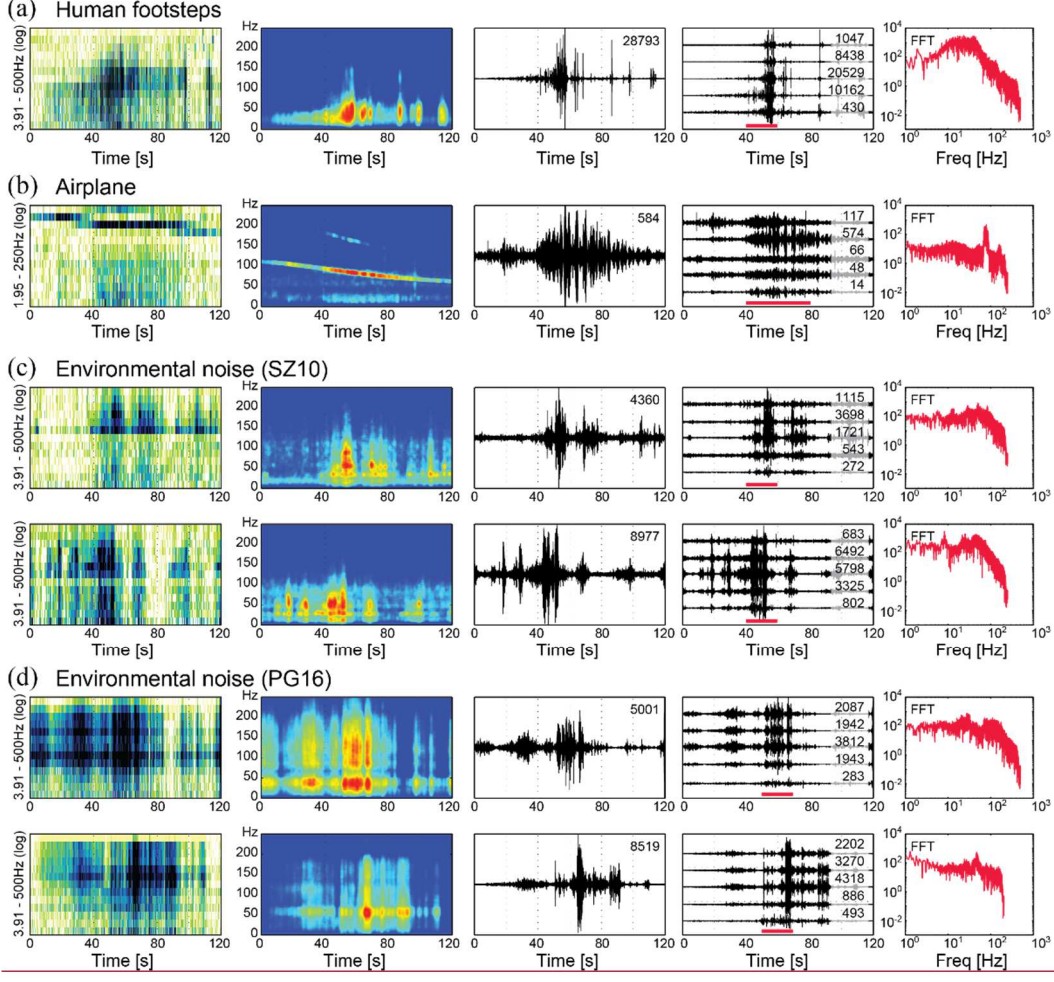

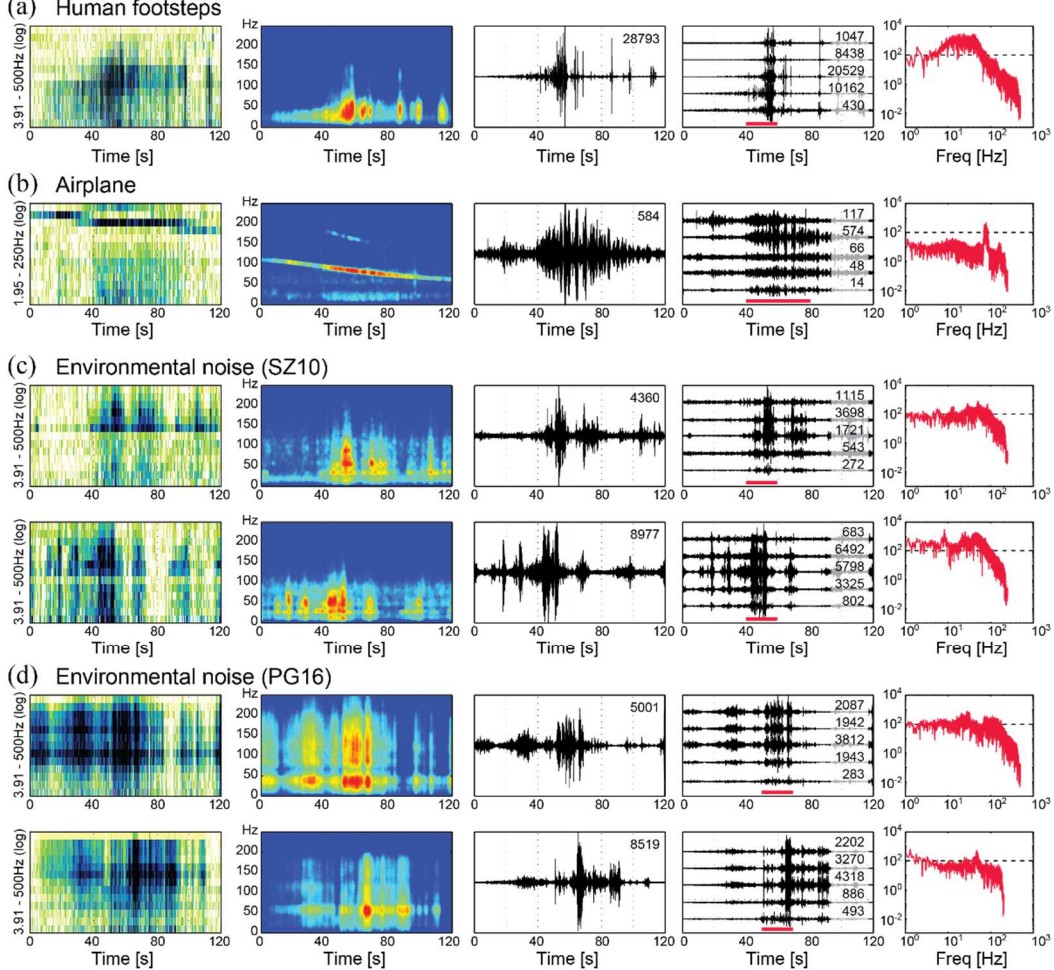

**Figure 8. Seismic features of ~~the most common~~ external sources ~~(non-exhaustive)~~ of tremor-like radiations. (a) Human footsteps at SZ10, S1.2, $t_0$ 2010.06.05 13:08:33. (b) Airplane at PG16, S2.1, $t_0$ 2016.11.08 04:56:00 with typical gliding harmonics in the spectrogram. (c) Environmental noise recorded at SZ10, stations S2.3 (top) and S3.8 (bottom) at $t_0$ 2010.06.09 22:54:10 and (d) at PG16, stations S2.1 (top) and S3.1 (bottom) at $t_0$ 2016.11.08 03:00:40.**

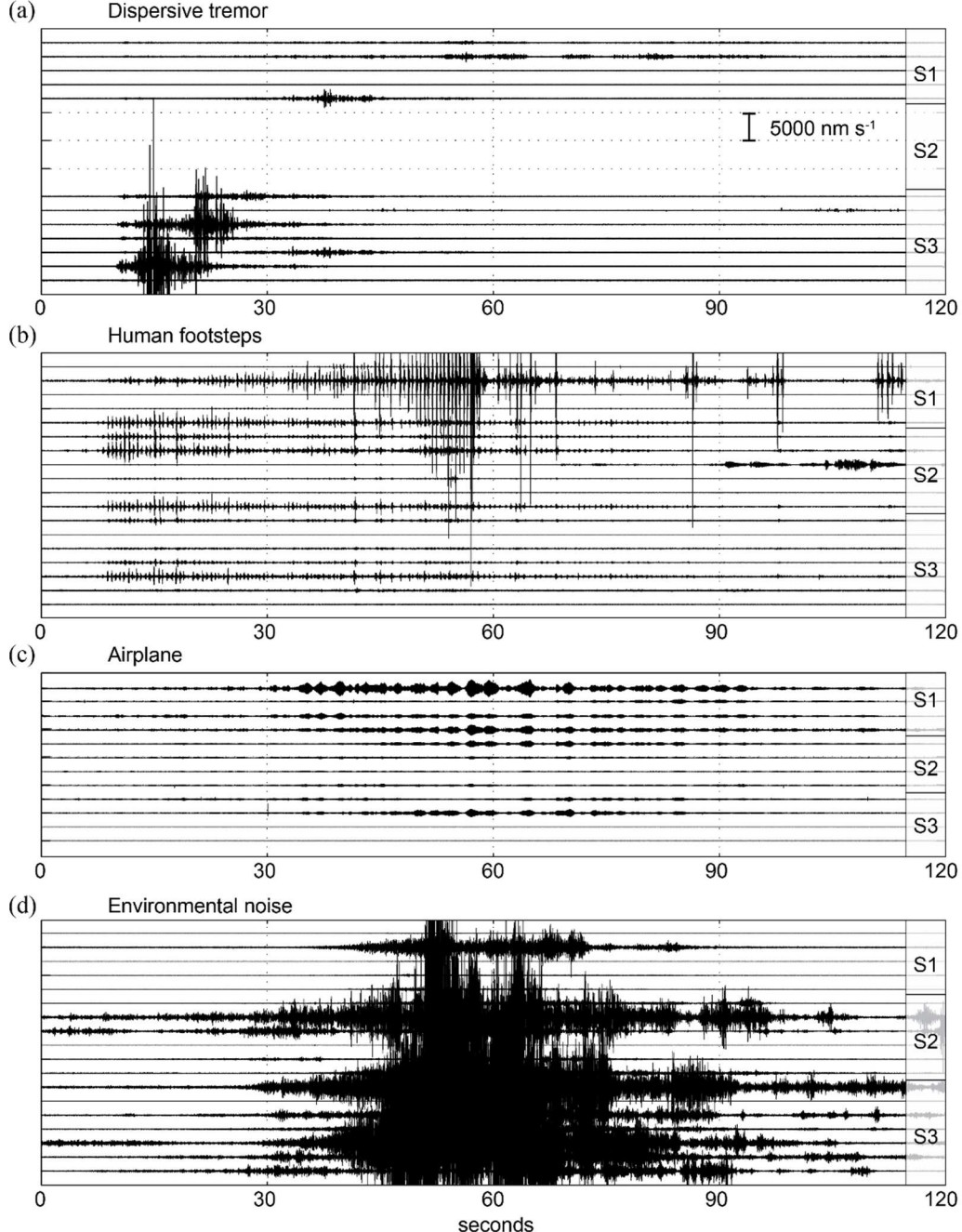

**Figure 9. Vertical trace seismograms of long duration tremor-like signals recorded at Super-Sauze and Pechgraben. A constant time and amplitude scale (indicated in (a)) is applied. (a) Dispersive tremor, SZ10, $t_0$ 2010.07.04 00:45:20. (b) Human footsteps, SZ10, $t_0$ 2010.06.05 13:08:33. (c) Airplane, PG16, $t_0$ 2016.11.08 04:56:00. (d) Environmental noise, SZ10, $t_0$ 2010.06.09 22:54:10.**

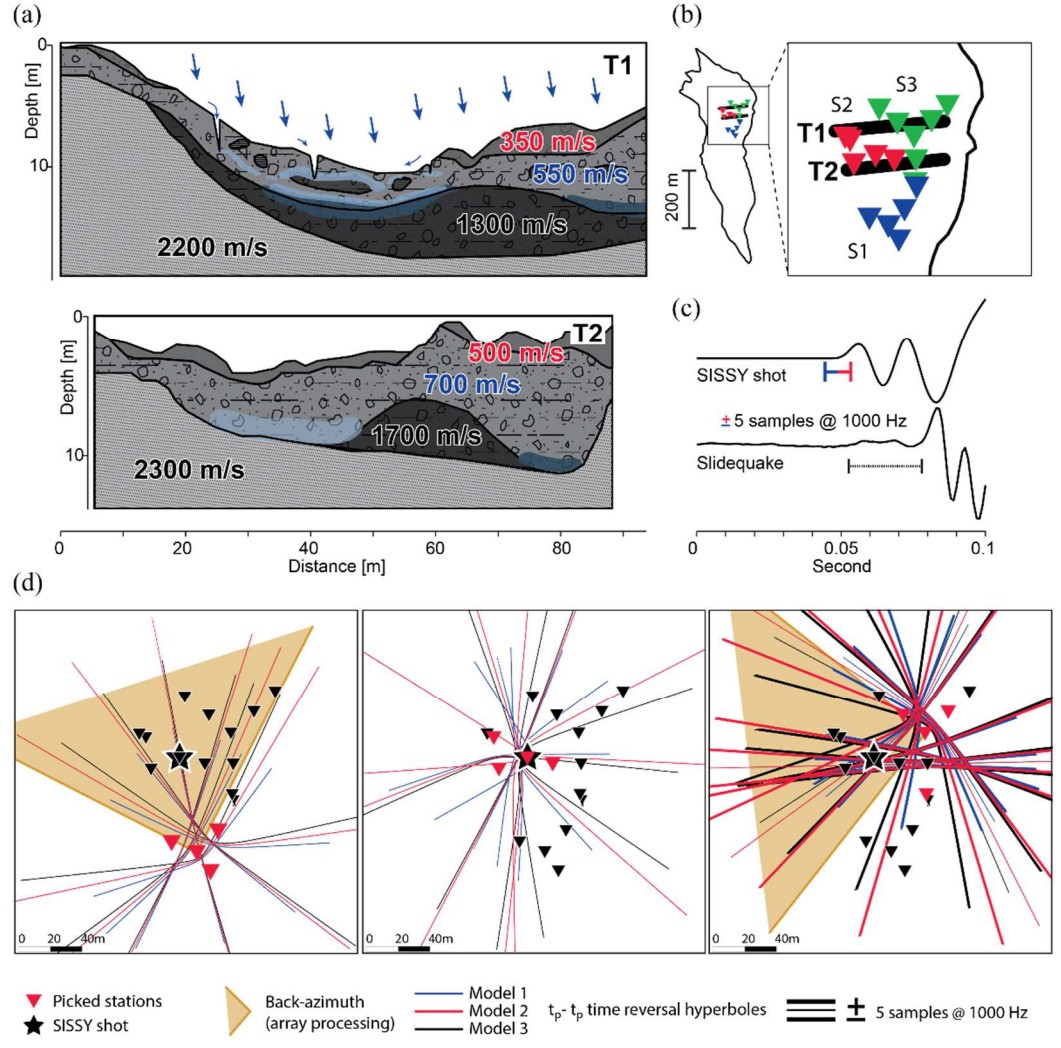

(a)

Depth [m]

T1

350 m/s
550 m/s
2200 m/s
1300 m/s

Depth [m]

T2

500 m/s
700 m/s
1700 m/s
2300 m/s

0    20    40    60    80

Distance [m]

(b)

200 m

S2    S3
T1
T2
S1

(c)

SISSY shot

± 5 samples @ 1000 Hz

Slidequake

0         0.05        0.1

Second

(d)

0  20  40m          0  20  40m          0  20  40m

▼ Picked stations     ▬ Back-azimuth        —— Model 1    $t_P$- $t_P$ time reversal hyperboles    ≡ ± 5 samples @ 1000 Hz
★ SISSY shot            (array processing)      —— Model 2
                                                              —— Model 3

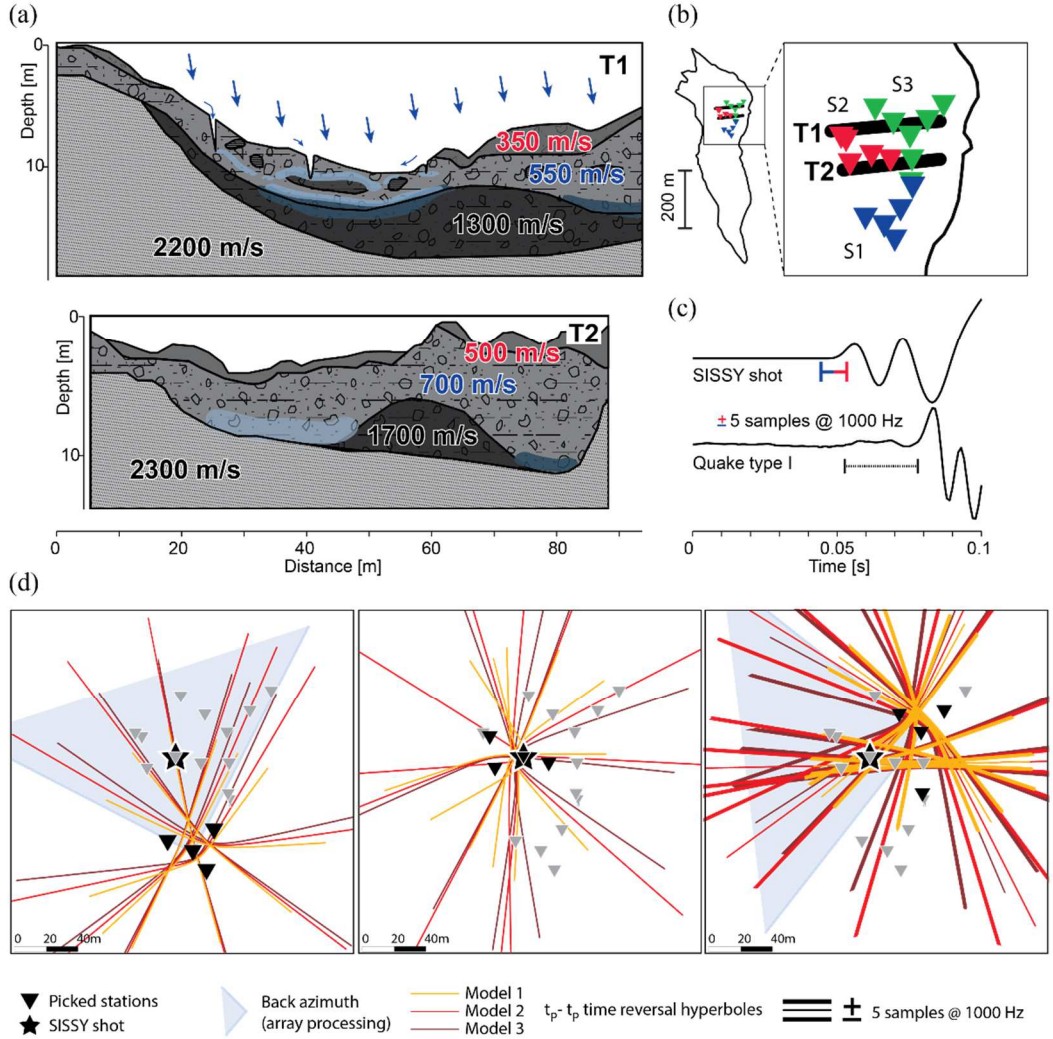

**Figure 10. Parameters impacting location uncertainties at clayey landslides. (a) Complex seismic velocity structures along two tomographic profiles T1 and T2 at Super-Sauze; modified from Tonnellier et al. (2013) and Gance et al. (2016). (b) Location of the tomographic profiles T1 and T2 within the seismic arrays S1, S2, S3. (c) High-quality first arrival of a SISSY calibration shot (top trace, SZ10, S2.2, t₀ 2010.06.04 11:56:22) and first arrival of a high-SNR quake type I event (bottom trace, SZ10, 2010.05.29 23:05:03). Note the higher uncertainties about the onset of the natural event. (d) Graphical location solutions for the SZ10 SISSY calibration shot at station S2.1, June 4, 2010, 11:56:22 derived from first arrivals at individual seismic array S1 (left panel), S2 (middle panel) and S3 inner ring (right panel). Picked stations are indicated by ~~red~~ black triangles, beam-processing results are symbolized by shaded ~~orange~~ light-blue quadrants, time-reversal hyperboles derived with three different velocity models (Table 2) are represented by ~~blue~~orange, red and ~~black~~brown ~~colored~~ lines. In the right panel, bold hyperboles image the effect of ± five samples uncertainties offset shifts in first arrivals. Discussion is found in Section 5.1.**

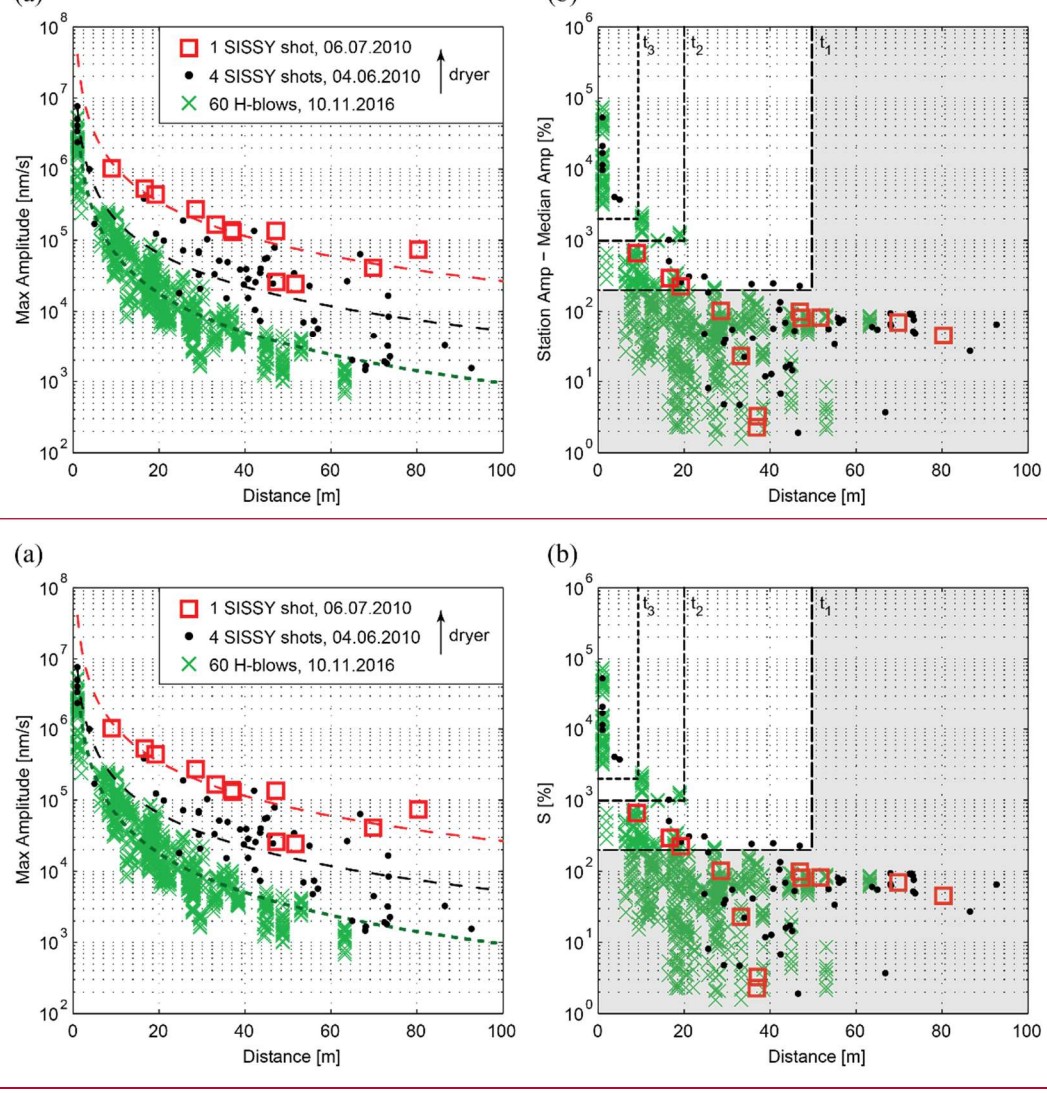

**Figure 11. (a) Maximum 0-to-peak amplitudes with distance to the source of SISSY calibration shots carried out at Super-Sauze June 4 (dots) and July 6 (squares), 2010, and hammer blows (crosses) carried out at Pechgraben, November 10, 2016.** ~~Dotted~~ **Dashed lines indicate log-log regression curves. Note the lower attenuation with dryer conditions. (b) Scatter about the median amplitude (S) of the calibration datasets presented in (a).** S~~catter about the median amplitude~~ **values of natural events higher than 200, 1000 and 2000 % are inferred to image receiver-source distances of about 50, 20 and 10 m ($t_1$ $t_2$ $t_3$) respectively.**

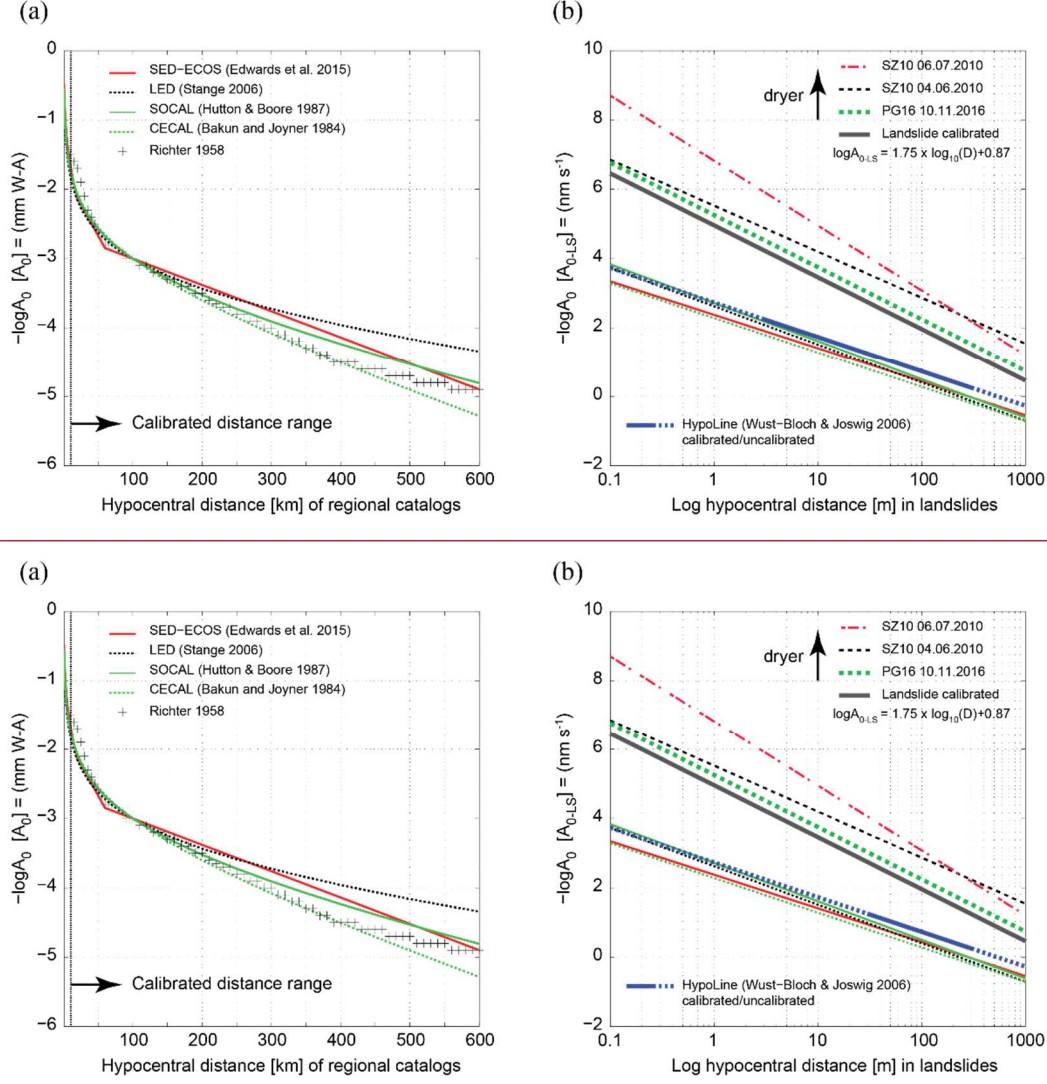

**Figure 12. (a) Distance attenuation functions (-log($A_0$)) of ~~regional~~ $M_L$ scales empirically calibrated for regional earthquakes with receiver-source distances between 10 and 600-1000 km. (b) Log-log zoom into the valid receiver-source distance range of landslide-induced microseismic signals. The HypoLine distance attenuation function, which was calibrated between 30 and 300 m in the Dead Sea Valley (Wust-Bloch and Joswig, 2006) is very similar to the projection of the regional $M_L$ scales. The distance attenuation regression curves derived from SISSY calibration shots and hammer blows data (see Figure 11) project in the upper area of the graphic, all with steeper slopes (imaging stronger attenuation) than the regional $M_L$ scales. The landslide calibrated distance attenuation function applies an average slope of 1.75 with an intercept of 0.87. Note that regional $M_L$ scales use displacement amplitudes in WA mm, whereas $M_{L-LS}$ scale is calibrated using velocity readings in nm s$^{-1}$. See discussion in Section 5.3.**

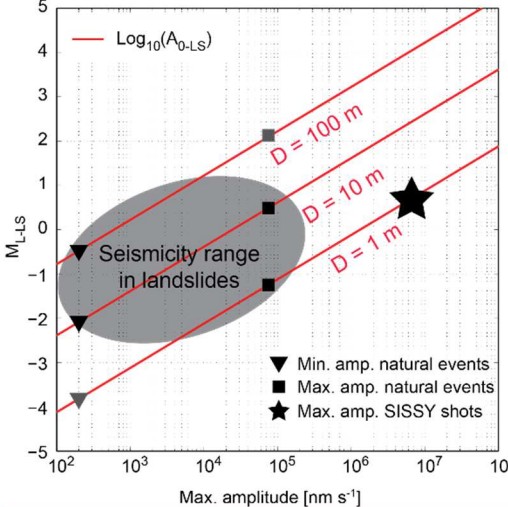

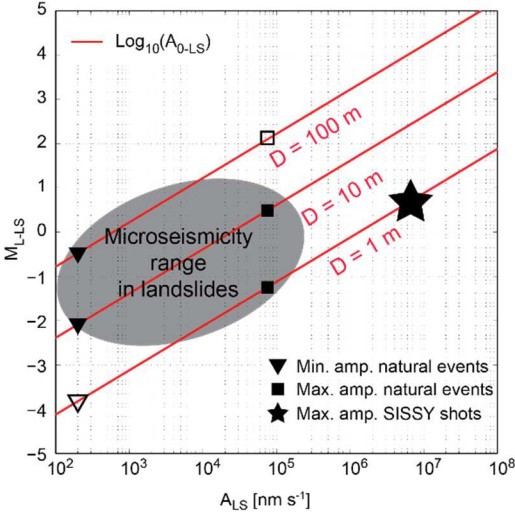

**Figure 13. M$_{L-LS}$ as a function of amplitude read in 1, 10 and 100 m receiver-source distances. The star indicates the average maximum amplitude reads of SISSY calibration shots in 1 m distance that corresponds to a M$_{L-LS}$ 0.58. Minimum and maximum signal amplitudes observed for landslide-induced signals are symbolized by triangles and squares respectively. ~~Grey~~ Empty symbols indicate lower probability valid distances of low and high amplitude values. A reasonable field of potential M$_{L-LS}$ of landslide-induced microseismic events is outlined by the shaded ellipse.**

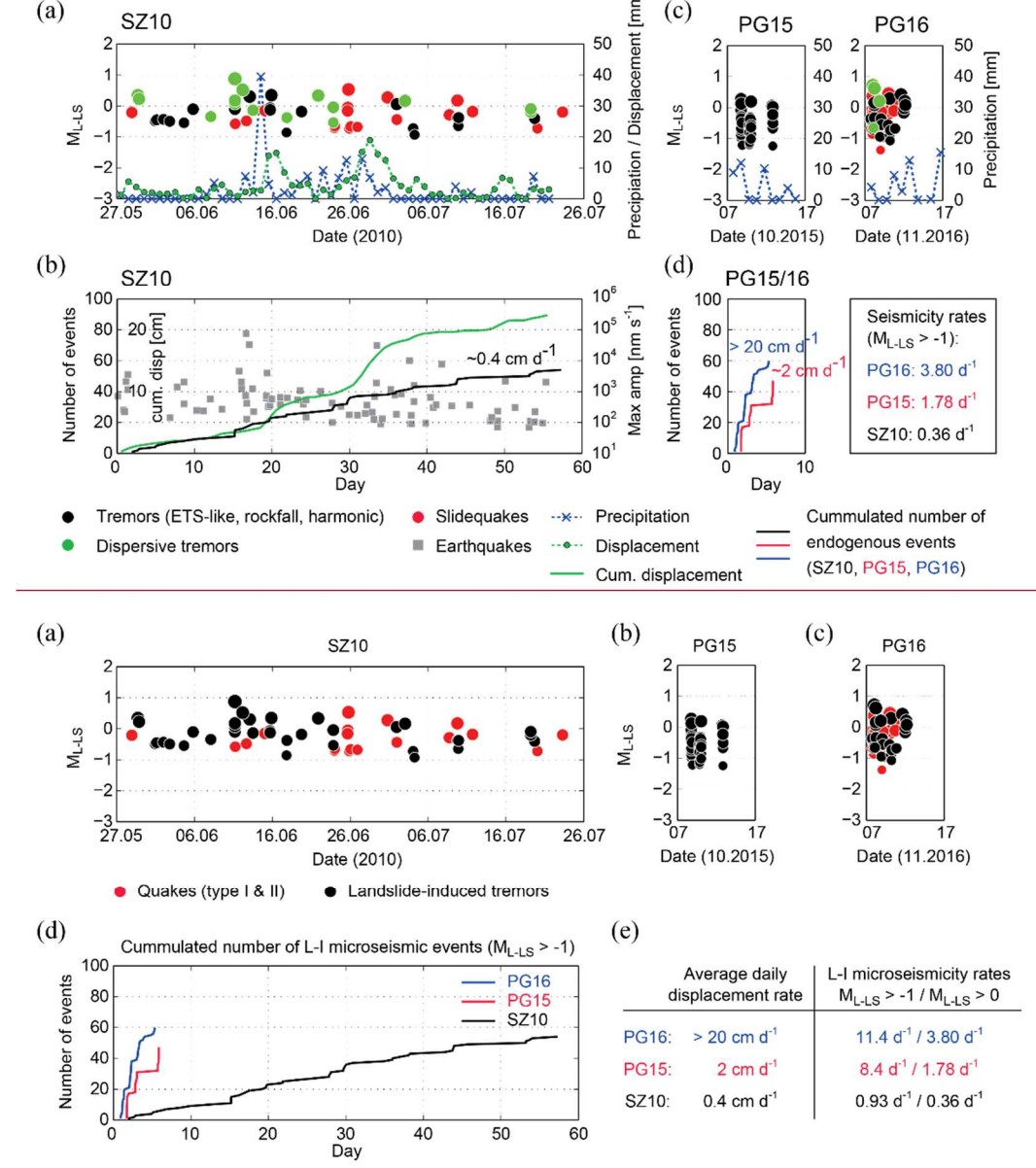

**Figure 14. (a-c)** Temporal distribution of $M_{L-LS}$ for near-~~source area~~ (< 50 m) landslide-induced microseismic events at SZ10 (a), PG15 (b) and PG16 (c). Red circles show quake events type I and II and black circles indicate landslide-induced tremors (ETS-like, rockfall, harmonic and dispersive with S > 200 %) ~~at SZ10~~. The time scale is constant in all plots. ~~with displacement and precipitation data.~~ **(b̶d)** Cumulative number of landslide-induced microseismic events ~~and cumulative displacement~~ at SZ10 with ~~the energy radiated by local and regional earthquakes (events median maximum 0-to-peak amplitudes in nm s⁻¹)~~ with S > 200 % curves for $M_{L-LS}$ > -1 events. **(c)** ~~Temporal distribution of $M_{L-LS}$ for near (< 50 m) landslide-induced events at PG15 and PG16 with precipitation data. (d) Cumulative number of landslide-induced seismic events at PG15 and PG16.~~ Landslide-induced microS̶seismicity rates for ~~($M_{L-LS}$ > -1)~~ and $M_{L-LS}$ > -0 show a clear increase with higher average daily displacement rates.

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
