# Peer review of "Characterizing the complexity of microseismic signals at slow-moving clay-rich debris slides: The Super-Sauze (Southeastern France) and Pechgraben (Upper Austria) case studies"

_Earth Surface Dynamics, 2017_

## Referee Comment (RC1) · E. Surinach (Referee) · 1 Jan 2018

1. Does the paper address relevant scientific questions within the scope of ESurf? YES.

2. Does the paper present novel concepts, ideas, tools, or data? YES.

3. Are substantial conclusions reached? Not, yet.

4. Are the scientific methods and assumptions valid and clearly outlined? Not, yet.

[Figure]

5. Are the results sufficient to support the interpretations and conclusions? Not,yet.

6. Is the description of experiments and calculations sufficiently complete and precise to allow their reproduction by fellow scientists (traceability of results)? Not,yet.

7. Do the authors give proper credit to related work and clearly indicate their own new/original contribution? Not,yet.

8. Does the title clearly reflect the contents of the paper? YES.

9. Does the abstract provide a concise and complete summary? YES.

10. Is the overall presentation well structured and clear? Not, yet.

11. Is the language fluent and precise? YES.

12. Are mathematical formulae, symbols, abbreviations, and units correctly defined and used? Not applicable.

13. Should any parts of the paper (text, formulae, figures, tables) be clarified, reduced, combined, or eliminated? YES.

14. Are the number and quality of references appropriate? YES.

15. Is the amount and quality of supplementary material appropriate? Not applicable.

More information.

-This is an interesting paper that deals with seismic signals associated to clay rich debris landslides. For the characterization of these signals the authors use well known seismological tools. The authors are very honest and prudent when presenting their results. It is a very hard work to deal with such a lot of signals; however, results are limited but not the information.

- The presentation of the paper is clear although the seismic terminology is used in a not appropriate manner. I recognise that it is difficult to use an appropriated terminology but I do not agree with the use of seismicity when the subject are not earthquakes.

We are dealing with events that produce seismic vibrations and seismic signals, but not earthquakes. Seismicity is related to earthquakes (e.g. Geological Survey or Enciclopedia Britannica definition). This observation is valid for the use of seismicity catalogue or seismicity rates.

- Landslide induced seismic signals or Seismic signals induced by landslides could be a good term.

- Regarding the localizations, in Seismology it is well known that one of the conditions for a good localization of events is to have a good and dense distribution of stations. As much as the readings you use best is the localization. The authors present a localization exercise with 4 readings, but they have more. In addition, to characterize the signals, the authors neglect the information of the horizontal components. A comment on all this will be illustrative.

- The authors in the interpretation of the different types of signals make some assertions related to their frequency content, attenuation, etc. (for example, small source, range of distances, apparent speed ...). Please, include a comment on the physics below or references.

-It is necessary an introduction on the tools used and what is the purpose of using them. Also the different roles and benefits of them or what the authors expect from their use. This would be useful for readers.

-It seems that there is redundancy in the applied methods. In addition, the explanation of the results in the text is very limited.

-In addition, the purpose of creating a catalogue can be explained in the introduction.

- As far as the FFT is concerned, this is not so trivial. Are you using the spectral amplitude or the PSD (Power Spectral Density)? The FFT of a function has a real part and an imaginary part (Phase). In addition, the units are not specified in the figure and this does not help to the interpretation.

- In general, the resolution of the figures is low. For example, in the spectrograms, a change of scale is necessary to observe in detail the behaviour of the signal. A comment on the resolution of the scale used in the analysis is needed. Or is it the same as shown in the figures?

-A Discussion section on the results is needed. A comment on the useful parameters and methods used with regard the expected. In the conclusions is indicated that information related to the frequency content and its time evolution was non relevant. This is due to the scale used? Please, an explanation on this is needed. Also relate your outputs to previous results.

-The most important contribution of the paper is the calibration of a magnitude scale (ML-SL) to the specific case of the clayey landslides for the locations of study. Also, the description of the conditions for a good study is valuable. The clarification that the magnitude scale for earthquakes is different to that used for landslides and also the differences in the ground motion parameters for their determination is a valuable contribution. A remark in this regard to the community was necessary for the proper use of seismological parameters.

ABSTRACT.

Pag. 1 L-13 The signals are generated, they are not triggered.

Pag. 1 L-15. Include a magnitude order for the duration of the longest signals (hours?).

Pag. 1 L-18 Replace ML to ML-ls in the local magnitude scale.

Introduction.

Pag. 2 L-17 Specify rock avalanches. Snow avalanches are not considered in the mentioned references.

Pag. 2 L-27. (3) sensor network geometry. Specify what type of network.

Pag. 2 L-24. The last part of the Introduction (from Pag. 2 L24) mostly corresponds to

results or discussion. Eliminate this part from here.

Pag. 2 L-28. Split the sentence: event location. Uncertainties. . ..

Pag. 2 L-31. Since the uncertainty. . . Please, explain better this sentence. What do you mean by seismicity rate? Note that you are not referring to earthquakes. And the word seismicity is specific for earthquakes.

Pag. 3 L-7. mm d-1 are not in SI units. I recommend to use SI units (m. s-1 ).

Pag. 3 L-10. Lindner et al., 2014 and 2016 are not included in references.

Pag. 3 L-12-14. Indicate the source of the displacement values or better explain how they are obtained.

Pag. 3 L-17. The data were collected.

Pag. 3 L-18. Please, include the references of Agécodagis (or Agéodagis ?) instruments.

Pag. 3 L-31. Replace the word seismic.

Line 25-26. This sentence is related to objectives does not correspond to data. Rephrase the sentence or move it to the introduction.

Pag. 4 L-5. Differentiate between sonogram and spectrogram in the context of this document. Both functions show the temporal evolution of the frequencies. Depending on the parameters of the display, the same information can be displayed.

Pag. 4 L-22. Complete this information explaining a little more the tools applied with respect to what you expect. This will help the reader.

Pag. 4 L-23. Indicate the aim to design a catalogue. This could be included in the introduction.

However, is design the correct word? Do you want to generate a catalogue with a good design?

[Figure]

Pag. 4 L-24. Replace seismic catalog by catalog of seismic signals induced by landslides or similar.

Pag. 5 L-7. Explain why the signals are expected to be severely attenuated and give references.

Pag. 5 L-17. Specify to what previous studies are you referring.

Pag. 5 L-26. landslides seismicity catalogs. Better replace by Lanslides induced sismic signals catalogs.

Pag. 5 L-32. The content of Fig. 2 should be explained in the text. In addition, somewhere in the text should include an explanation of the selection of the tools used in the Figure, their role and benefit. As far as the FFT is concerned, this is not so trivial. Are you using the spectral amplitude or the PSD (power spectral density)? The FFT of a function has a real part and an imaginary part (Phase). In addition, the units are not specified in the figure.

Pag 6 (4.4.2) and Fig 3. You affirm that in type II there are surface waves, I also observe in Type III similar shapes of better developed surface waves. This is normal because distance is longer. In the case of local microearthquake (Type IV) I observe perfectly the surface waves at 3 s from the first arrival. The energy immediately 0.2 s are S waves (although is a vertical component).

Pag. 6 L-14. Indicate the reasoning to assume that the distances are > 50 m or other in the different types of signals.

Pag. 6 L-15. SI units, 10-5 ms-1.

Pag. 6 L-18. What do you mean by later phases? Later arrivals? If they correspond to surface waves, the speed is lower, but if they are wave phases that travel in depth, the velocity must be greater although the distance affects the apparent speed. ... Explain this better or delete the parenthesis. The seismograms in figure 3 are presented following the stations code, but please, indicate which the time origin of the seismograms

is. The apparent velocity is in relation to the distance source-sensor and this is not indicated. Or is it unknown?

Pag. 6 L-22. Indicate the figure in detail (for example, Fig. 4c-5), when referring to the dominant frequencies.

Pag. 7 L-4. Please, indicate the epicentral distance. It seems to me that P and S waves and also superficial waves are observed clearly. See Fig 3d.

Pag. 7 L-5-15 this could also be part of the conclusions. Moreover, you obtained similar signals PI and PII at both sites (SZ10 and PG6). This is remarkable.

Pag. 7 L-7. Explain why a large attenuation is evidence of a nearby source.

Pag. 7 L-9- 10. Please explain why you infer these statements. Nearest source, small source ... This could be discussed in the introduction or elsewhere.

Pag. 7 L-12-16. Some explanation or references are needed to support the assertions. Pag. 7 L-23. The mentioned publications are not indicated in the references.

Pag. 7 L-25. The mentioned publications are not indicated in the references. In addition, in figure 5 in Biescas et al, 2003 (Surveys in Geophysics, 24, 447-464) is the first time that helicopters and moving vehicles are mentioned in this context. In relation to the Doppler effect, it is necessary to vary the speed of the mobile source with respect to the stationary observer and an acceleration to observe a change in the frequencies that vary in time in the spectrogram (gliding).

Pag. 7 L-27. The correct word is generated, it is not triggered.

Pag. 7 L-28. What is the reason for including this sentence in relation to the Q factor? Presented in this way does not contribute to anything. Remove it.

Pag. 8 L-7. Specify seismic network here and in all the cases.

Pag. 8 L-13. Replace 4.3.1 by 4.3.2.

ESurfD
[Figure]

Pag. 8 L-15, 16 and 22 Replace are by were.

Pag. 8 L-24. Similarly to previous sections accompany the subtitles with a dash e.g. - ETS-like signals (episodic tremor and slip); - Confirmed rockfall events. . ...

- ETS-like signals. How do you deduce that the signals correspond to episodic tremors or slip? Is it related to previous studies? You have to mention that in this subsection.

Pag. 8 L-25. seismic network.

Pag. 8 L-26. fig. 5b Specify better and give more details.

Pag. 8 L.31- Fig. 6b Specify better and give more details.

Pag. 9 L-6. Do you mean low topography or smooth topography? Perhaps a short description of the sites is necessary.

Pag. 9 L-10. This subsection is not very clear. No description of the signals is presented. The only result is that since they are very similar to ETS they can be eliminated if there is a field observation.. However, as the physical process of rocksfalls (impacts) and slip is different, if the signals are observed in more detail they can possibly be separated. Additionally, if you consider the 3 components you can observe differences between a slip and a rockfall. See fig. 3 in Vilajosana et al., 2008. The signals are different in the horizontal and vertical components.

Pag. 9 L-13. Fig. 5a and 5c Specify better and give more details.

Pag. 9 L-17. Fig. 5a -c? Specify better and give more details.

Pag. 9 L-20. Figures 5d and 6d show two signals from Walter et al., 2012. However, no reference is made to these figures in the text. A discussion or comparison of the signals with those of this contribution is needed.

Pag. 9 L-24. In Fig 7 there is quantity of information that is not explained in the text. More description of the images is needed. Maybe I am lost, but how do you know the

source situation?

Pag. 9 L-26. Why does the temporal evolution of dominant frequency content suggest a mobile source? Indicate the appropriate figures and better specify the information you want to communicate. Also indicate references.

Pag. 10 L-1. The information in this subsection is anecdotal and unnecessary. It does not contribute anything to the knowledge of the phenomenon. These signals are context-independent. And its characteristics depend on the distance from the sensor to the seismic source. Always, when the location of a station is inappropriate, there are external sources and noise. Fortunately, as mentioned in the text, in some cases they are in the high frequency range and a high pass filter eliminates it. As it is impossible to collect all kinds of signals, I recommend that the authors mention the possibility of registering external sources and eliminate this part with the corresponding figures and devote the space to analyse the signals produced by landslides.

Pag. 10 L-28. Is the coda important for the location? In addition, as indicated, the attenuation is very high, so the coda disturbs little.

Pag. 11. All the points related to geometry and distribution of the elements of the network with regard the good location of the source are well known in seismic location. This is not new. The only doubtful point is the low impact of the velocity model to locate the superficial source. Normally velocity model affects locations. Are significant these mentioned few meters with regard the distances involved? I propose the authors to construct travel time curves for the direct and reflected waves in the range of 150 m for the 3 velocity models to see the effect in the travel times and discus the result. Additionally, you are using only 4 readings. Why? Usually, more readings are necessary to obtain an over-determined system. Comments on all this are necessary.

Pag.11 L-31. Seismicity map is not an appropriate term. Seismicity map is related to the distribution of earthquake epicenters. You are dealing with other events. Perhaps a good term is "map of landslides related events location".

Pag.12 Estimation of source proximity trough waveform attenuation. This is an option, but Fig. 11a shows local site effects, which are also observed in the dispersion on Fig. 11b. It's not so easy for me to validate the distances of the source. Explanations or references are necessary.

Pag.12 L-14. seismicity rate is an inappropriate term. Perhaps Rate of landslide related events.

Pag. 12 L-16. 5.3 I agree with the authors that it is necessary to remind the origin of the magnitudes to avoid an inadequate use of this concept. Very interesting precision regarding the differences between magnitude scales.

Pag. 12 L-17. Specify: Earthquake local magnitude scale ML.

Pag. 12 L- 24. Fig 12 is the correct Figure.

Pag. 13 L-1. ..define Ao-LS and indicate its units.

Pag. 13 L-1. is logfit a MATLAB script? Indicate.

Pag. 13 L-7. In units SI is KJoule not KiloJoule.

Pag. 13 L-11. Better to express A= 5.10 -3 m/s (SI units).

Pag. 13 L-14. A is the absolute value (include <0 values).

Pag. 13 L-17. Homogenize terminology in Pag.12 L-19 and here.

Pag. 13 L-26. Consider my previous comments on the use of the term seismicity. Catalogs of clayey landslides events.

Pag. 14 L-1 and L-4. Consider my previous comments on the use of the term seismicity. Landslides-induced events or landslides-induced seismic events are more corrected terms than seismicity.

Pag. 14 L-5. Section 5.4.1 is too short. I recommend the authors expand the explanation to give value of the results. Figure 14 shows amount of information that needs

**ESurfD**
comments. Pag. 14 L-5. Consider my previous comments on the use of the term seismicity. landslides-induced seismic events rates. Although, I recognise is a long title. In addition, what do you mean by seismicity rates, activity of seismic events?.

A Discussion section on the results is needed. A comment on the useful parameters and methods used. As it is indicated in conclusions information related to the frequency content and its time evolution was non relevant. Please, an explanation on this is needed. Also relate your outputs to previous results.

Pag. 14 L-7. indicate Fig. 14 a and c.

Pag. 14 L-10. SI units (?).

Pag. 14 L-13. landslides-induced events.

Pag. 14 L-14. Conclusions and outlook .

Split the content and include it in the discussion section to give it more.

Pag. 15 L-5. indicate ML-LS.

Pag. 15 L-6. Where are the mentioned displacements?

Pag. 15 L-24. Consider my previous comments on the use of the term seismicity.

Figures and Figure captions.

Figure 1. Seismic arrays Si should be presented before. Not in e) A better explanation of the colours will improve the understanding of the figure Higher and lower dynamics of what? To help the reader, indicate what 3C seismometers are. S1.1, S3.0 and S2.1 in SZ (c) and S1.1, S3.1 and S2.1 in PG (d)

Figure 2. -Show first, in a), your time series. These are your data. Indicate that is the vertical component

-a) sonogram of the vertical component. Explain the sonogram in the text and not in the legend of the figure; the same for the spectrogram.

-e) Specify amplitude spectra or PSD instead of FFT and indicate the units.

-Include the colour scales in a) and b). There is certain incongruence between the information for the sonogram and the spectrogram. I do not understand what the utility of the spectrogram is.

-The information of the spectrogram and sonogram must be the same with an appropriate scale and a correct overlap.

-Moreover, what is the role of the different filtered bandpass signals if you do not mention them in the text?

- Also, because the frequency content of the earthquake is up to 40 Hz, showing the scale up to 250 Hz is meaningless and leads to loss of information. Similar comments for figures 4 and 6.

Figure 3. What is the disposition of the seismograms? Which criteria have you follow?. Replace rectangles by bands.

Figure 4. Number columns 1-5 in each row to help the description in the text, Could you explain the difference in the frequency content of the sonogram and the spectrogram? Is there a filter between the two calculations?.

Figure 5. Indicate that the number of the stations is on the right. As regards the amplitude, note or specify that it is the absolute value. Events a) and c).

Figure 6. Consider my comments for figures 2 and 4. Indicate the distance to the sensor of the confirmed rockfall to help readers. Are the signals reproducible and repetitive? A comment on this is necessary in the text.

Figure 7. Consider my comments for figures 2 4 and 6.

Figure 8. This figure shows some of the external sources, the series is not complete and the signals depend on the cases and distances. It does not contribute anything to the paper. Remove it.

Figure 9. Consider the comments in Figure 8. Remove it. If you present these signals, you should explain their characteristics fully in the text, not just showing the plots.

Figure 11. If I understand correctly, data in a) correspond to experiments showing different amplitudes for similar distances. (e.g. squares in 40-60s in a) possible due to local effects. How do you correct the site effect in this process? An explanation is needed in the text. Replace dotted lines in the figure caption by dashed lines.

Figure 12. a) Specify that it is for regional earthquakes

Figure 13. Better to place empty symbols because grey is difficult to see. Replace the word seismicity.

Figure 14. Figure very interesting that corresponds to the results. More explanations in the text are needed. Replace the word seismicity.

---

## Referee Comment (RC2) · A. Köhler (Referee) · 9 Jan 2018

General comments:

The authors do a great job analyzing and categorizing the seismic emissions of very complex source processes. It is a laborious task, but essential to be able to use seismic monitoring as a reliable tool to forecast the behavior of debris slide. The authors try to establish an objective and consistent processing scheme, which of course has its limitations for such a complex seismic record. The paper is well-written and presents

in most parts conclusive and comprehensive results, giving all the necessary details. However, some clarification are necessary and there are issues that should be discussed in more detail. My main concern is the temporally varying completeness of the seismic event record resulting from the manual processing scheme applied. This could have important implications on the interpretation of how well the seismic rates correlate with measured displacement of the slide.

Specific comments:

(1) Page 4, section "3 Methods": I would suggest to remove the sub-section 3.1 since only a single method is introduced.

(2) Only sonograms on night time measurements were screened to minimize false detections: What implications does this have on the fact that you use the seismic rate in the end and compare it to slide displacement? Or does it not matter because the displacement data has daily resolution (see also comment about Fig. 14 below)? Is there any process that could lead to an increase of seismic emissions from the landslide body during daytime which you would miss (e.g. diurnal temperature forcing)? Is there a changing noise level at nighttime that could affect the observed seismic emission rate (change of "visual" detection threshold)?

(3) Features presented on Page 5: Maybe it is worth a try to use an automatic clustering method in the feature space. That would open up the possibility to use some sort of STA/LTA network trigger and then perform post-classification afterwards. Showing some scatter plots would help to evaluate if the feature distribution from all events actually shows a clustering related to your event classes, or if the classes rather represent end-members of a continuous distribution. Even if implementing an automatic clustering method is beyond the scope of this paper, the potential of such an approach could be discussed.

(4) Array processing: How reliable is beamforming here given that source-receiver distances are rather short compared to the array apertures (non-planar wavefronts)?

(5) Page 8: 4.3.1 change to 4.3.2.

(6) Page 6: If Type IV are most likely events outside the landslide, wouldn't it be better to describe them in section 4.1. (local earthquakes)? In this case section 4.2. "quakes" could be renamed to something like "seismic landslide signals" or "slidequakes" (which has been used previously).

(7) Page 10, line 9: Since you wrote you just used nighttime records for manual data inspection and there are signals corresponding to "geophysicist walking": Is this consistent? (Of course geophysics might also work at night sometimes . . .)

(8) Page 11: Please explain what you mean with "the domain of existence of the hypolines was tested ..."

(9) Page 11: Line 23-27: See previous comments (consider non-planar wavefronts in addition to scattering and inhomogeneities)

(10) Page 12, Section 5.2: "normalized difference between the maximum amplitudes of the signals and the median value of all maximum amplitudes": This is a bit unclear. I would suggest to add an equation to define your variable "scatter about the median amplitude". Then, you could refer to this quantity simply with a letter. Also, what is the percentage of this variable being larger than 200% for each of the 4 event types? Here you just give percentage of all events (type I-IV (?) quakes and tremors). I would expect a clear difference for Type I and II compared to Type IV.

(11) I suggest to discuss some potential methods for automatizing the event detection. You mentioned template matching. What about cross-correlation of (array) envelopes instead of waveforms? See also comment above about clustering. It could be an option to introduce a separate discussion chapter which includes the issues of catalog completeness, automatization, and correlation of event rate with slide displacement.

(12) Table 2: Is this Vp or Vs?

(13) Fig 3: Why are Type II events absent for stations S1.1-S1.5 ?

(14) Figure 10d: This panel is a bit hard to read, especially because you use two colors (red and black) with two different meanings (Red for picked stations and model 2). Maybe I misunderstand, but it looks like the hyperbolas intersect always close to the center of the array used for beamforming, but the true location is outside the array. Is this coincidence and what could be the reason?

(15) Figure 11: It is not clear how you defined the three thresholds. I can follow the choice for T3, but T2 and T1 seem a bit arbitrary. Please explain in more detail.

(16) Figure 14: The comparison between seismic rates and displacements from this figure is very difficult, especially to prove your statement on Page 14, line 19-20, that seismicity rates show a clear increase with increasing displacement. Consider to increase the size of the sub-figure. Also, red symbols (slidequake) may be hidden by tremor symbols. As far as I understood, all events only occurred at nighttime. In this case you could indicate the data time periods you did not screen. Please also indicate where you see correlation between seismicity and displacement. I suppose "endogenous events" is the same as "land-side induced events" and includes both tremors and slidequakes?

---

## Author Comment (AC1) · 5 Feb 2018

**Final response to Reviewers**

Dear Reviewers and dear Editor,

My co-authors and myself would like to thank Emma Suriñach (Reviewer #1) and Andreas Köhler (Reviewer #2) for reviewing our manuscript and for providing us with very constructive comments. We've also much appreciated a non-anonymous review!

In this letter we have responded first to the general comments and critics made by the two reviewers. Then, we have addressed each comment individually in red and applied the suggested and related modifications in the revised manuscript using a track-change mode. New text sections included in the response letter are indicated in green, with reference to the page and line numbering in bold of the revised track-change manuscript.

We hope that our changes and clarifications, both in this letter and in the revised manuscript text and figures, fully address the remarks made by the two reviewers.

Again, we would like to thank you for your time and efforts in helping us improving this manuscript.

With my best regards,

Naomi Vouillamoz

**Response letter**

The main critic arising from the two reviewers, was the lack of a clear and structured discussion about methods and results. We agree and have now removed results interpretation parts previously disseminated in Sections 1, 4, 5 of the original manuscript to create a new, well-structured dedicated discussion section (see Section 6 in the revised manuscript):

**6 Discussion of microseismicity catalogs at clayey landslides**

6.1 Landslide-induced microseismic events detection and classification

6.2 Landslide-induced microseismic event location and interpretation

6.3 Landslide-induced microseismicity rates

The main concern of Reviewer #1 was that "*results are limited, but not the information*". We disagree with that statement: the information IS indeed limited! The seismic network operation is difficult. The terrain roughness makes it only partially accessible, rendering continuous seismic recordings challenging. The seismic network cannot be deployed with optimal station geometry because of fissures, vegetation, creeks, etc. existing on the slope. With slope movements (up to 50 cm per day), 1-C station get tilted; 3-C stations get tilted and rotated. These points impact the seismic record quality and completeness. The target landslide-induced microseismic events are of low magnitudes. Because of the heterogenous material and its high saturation, waveforms are strongly attenuated and scattered. As a consequence, landslide-induced microseismic events are not recorded optimally by all seismic stations, signals display low-SNR onsets and phases are difficult to pick.

This leads to the sad situation, that despite a very dense seismic network (station-station distance of 5-50 m), a source generated within the seismic network cannot be located by standard seismological approaches with less than about 50 m uncertainty. Whereas this kind of uncertainty is well adapted for microseismic studies at the kilometric scale, it is problematic for us, where 50 m scales with the seismic network and with the landslide itself.

In other words, such results do not allow to discriminate between a source within or outside the landslide and thus, the source cannot be interpreted unambiguously. This was the main reason that drove us to develop a simple approach based on waveform attenuation to better constrain receiver-source distance, to discriminate near-source area microseismic events from distant microseismic event and diminish bias in the estimation of landslide-induced microseismicity rates. We reworked the corresponding Sections 5.1, 5.2 and 6.3, so that the aim of the location exercise is made clearer to the reader.

An important comment by reviewer #2 concerns automatic approaches for event detection and classification. We totally agree that the future is about developing automated systems to detect and classify landslide-induced microseismicity (as mentioned already in the abstract). However, automatic detection algorithms work fine for well-known routine seismic signatures but fail for the unknown and unexpected low-SNR microseismic signals. The feature space that we describe in the Section 3 provides the most primitive subset of the evaluated high-dimensional feature space of landslide-induced microseismic events. As an example, Provost et al. 2016 used 71 seismic features to train their random forest classifier on a large training set at Super-Sauze landslide. The classifier developed by Provost et al. 2016 works fine for well-defined single microseismic events such as earthquakes, quakes, rockfall but fails to classify complex tremor signals. Other automatic classifiers to be tested on clayey landslides include for example unsupervised pattern recognition (e.g. Sick et al. 2015) or Hidden Markov Models (e.g. Hammer et al. 2012). However, the high variability of tremors radiations as observed on clayey landslides is still best identified by an analyst in the enhanced visualization provided by the sonograms. This paper aims at providing an initial signal library of microseismic-signals at clayey landslides including tremors for the future development of automatic systems and their benchmark in terms of success rate and scalability. In case of interest, the data can be requested to the authors, as already mentioned in the data and resource section. As asked by reviewer #2, we have now developed and better specified these issues in the discussion of Section 6.1 and in the conclusion.

Taking into account the remarks made about the complexity of Figure 14, we decided to remove the 'landslide dynamics' interpretation part of Figure 14. The reason for this is that the data provided originally in Figure 14 was not completely discussed and beyond the scope of this paper. Each seismic campaign described in this paper was part of a multidisciplinary field campaign, where other geophysical, hydrological, geomorphological and geotechnical data were acquired at high repetition rate (daily), simultaneously to the passive microseismic measurements. The comparison of all dataset, and the discussion about the gain of the various methods to constrain passive microseismic observations was the central subject of a PhD thesis, submitted in December 2017 (Sabrina Rothmund, Geodätische und fernerkundliche Beiträge zum Verständnis rutschungsinduzierter Seismizität an tonreichen Lockergesteinsrutschungen). The precise analysis of these data will be the topic of a new dedicated paper. In the present paper, we thus limit the discussion to landslide-induced microseismicity rates based on the average daily displacement rates observed during the three seismic campaigns. The Figure 14 was modified accordingly.

**Comment by E. Suriñach (Reviewer #1)**

**A)** This is an interesting paper that deals with seismic signals associated to clay rich debris landslides. For the characterization of these signals the authors use well known seismological tools. The authors are very honest and prudent when presenting their results. It is a very hard work to deal with such a lot of signals; however, results are limited but not the information.

We answered this comment in the response letter and reworked Section 5.1.

**B)** The presentation of the paper is clear although the seismic terminology is used in a not appropriate manner. I recognise that it is difficult to use an appropriated terminology but I do not agree with the use of seismicity when the subject are not earthquakes. We are dealing with events that produce seismic vibrations and seismic signals, but not earthquakes. Seismicity is related to earthquakes (e.g. Geological Survey or Enciclopedia Britannica definition). This observation is valid for the use of seismicity catalogue or seismicity rates.

- Landslide induced seismic signals or Seismic signals induced by landslides could be a good term.

Indeed, the conservative terminology refers to earthquakes. We therefore used *landslide-induced microseismic events* or *landslide-induced microseismicity* when referring to signals generated by the landslide dynamics in the paper.

**C)** Regarding the localizations, in Seismology it is well known that one of the conditions for a good localization of events is to have a good and dense distribution of stations. As much as the readings you use best is the localization. The authors present a localization exercise with 4 readings, but they have more. In addition, to characterize the signals, the authors neglect the information of the horizontal components. A comment on all this will be illustrative.

We agree. The aim was not clear. We added an introduction to the Chapter 5.1, so the reader understands better the aim of the location exercise, which was to simulate the location procedure of a 'real' landslide-induced microseismic event and test which parameters are the most influencing ones in the location solution. In our case, with an average station-to-station distance of 5-50 m, most landslide-induced microseismic events feature no more than four unambiguous phase information. The use of horizontal traces to picking of S-phase is not trivial: for near-source area landslide-induced microseismic events (< 50-100 m), S-waves and surface waves cannot be discriminated. The location procedure of a real landslide-induced seismic event is therefore not as straightforward as the location of a calibration shot where many high-SNR phases information are available.

The first part of the Section 5.1 reads starting **P12 L32**:

"Seismic velocities and source location quality can be estimated and verified by calibration shots or hammer blows. Calibration shots and hammer blows were carried out at SZ10 and PG16 and could be located with average accuracies of about ± 50 m, when using all available first arrivals and back azimuth information with a half-space velocity model. Our results concur with previous results by Tonnellier et al. (2013) at Super-Sauze landslide, where uncertainties of 40-60 m where estimated for calibration shots carried out within the seismic network. It is worth mentioning that this corresponds to the size of the seismic network and scales with the landslide itself. Thus, even if the seismic network is dense, locating landslide-induced microseismic sources in clayey landslides and discriminating between a source originated within or outside the landslide body is challenging: (1) The velocity structures show drastic variations in short distances (complex material mélange, topography), and also evolves with time (slope deformation, hydrological state). Velocity models are thus only approximated by tomographic analysis for a specific time (Fig. 10a-b). (2) Scattering and attenuation of the waveforms result in low-SNR onsets where phases are difficult (if not impossible) to identify. (3) The seismic network geometry relative to the source is in most natural cases not optimal. (4) With an average station spacing of 5-50 m, as it is the case in our study, most landslide-induced microseismic events show no more than four unambiguous phase information."

**D)** The authors in the interpretation of the different types of signals make some assertions related to their frequency content, attenuation, etc. (for example, small source, range of distances, apparent speed ...). Please, include a comment on the physics below or references.

We added the references Aki and Richards 2002 and Koerner et al. 1981; and answered to related specific remarks below individually.

**E)** It is necessary an introduction on the tools used and what is the purpose of using them. Also the different roles and benefits of them or what the authors expect from their use. This would be useful for readers.

We reworked the Section 3 accordingly and removed the Section 3.1 as also suggested by Reviewer #2.

**F)** It seems that there is redundancy in the applied methods. In addition, the explanation of the results in the text is very limited.

We agree that sonogram and spectrogram are both representations of the frequency content over time. We used sonogram specifically for the detection of the events, because they feature a dynamic, frequency dependent adaptation to the background noise and therefore facilitate the detection and discrimination of weak signal energies. We presented spectrograms because this tool is widely used in the community and also because the spectrograms provide a finer appreciation of the dominant frequencies by applying a non-logarithmic ordinate. We specified more clearly the intention of using sonogram in the text of Chapter 3, **P4 L20 to clarify**

"The enhanced visualization of sonograms has unmatched power to facilitate the detection and recognition of various type of weak signal energies in low-SNR (signal-to-noise ratio) conditions without a-priori knowledge (Joswig, 1990; Sick et al., 2012; Vouillamoz et al., 2016)."

**G)** In addition, the purpose of creating a catalogue can be explained in the introduction.

We added a sentence in the introduction **P3 L8**: "Microseismic observations were gathered in a comprehensive catalog. The final catalog of landslide-induced microseismic signals provides an initial microseismic signals library to train automatic detection and classification systems as well as an important basis for a multidisciplinary comparative analysis with other landslides observations such as displacement, cracks and fissures development, or hydrometeorological data to gain knowledge about landslide dynamics.."

**H)** As far as the FFT is concerned, this is not so trivial. Are you using the spectral amplitude or the PSD (Power Spectral Density)? The FFT of a function has a real part and an imaginary part (Phase). In addition, the units are not specified in the figure and this does not help to the interpretation.

We computed spectral amplitudes with units in nm Hz$^{-1}$, now indicated in the text and in the caption of Figure 2. We added a reference line at $10^2$ nm Hz$^{-1}$, which corresponds about to the background noise threshold, to help the reader in comparison of the different events.

**I)** In general, the resolution of the figures is low. For example, in the spectrograms, a change of scale is necessary to observe in detail the behaviour of the signal. A comment on the resolution of the scale used in the analysis is needed. Or is it the same as shown in the figures?

We will provide vector files in the final version of the paper. We applied the same layout for all figures presenting the event typology to facilitate the comparison between different event types. Of course, larger figures with higher resolution, as well as table with quantitative values were used during the analysis.

**J)** A Discussion section on the results is needed. A comment on the useful parameters and methods used with regard the expected. In the conclusions is indicated that information related to the frequency content and its time

evolution was non relevant. This is due to the scale used? Please, an explanation on this is needed. Also relate your outputs to previous results.

Yes, as said in the response letter, we gathered results discussion previously disseminated in Chapter 1, Chapter 4 and 5 in the Section 6, discussion.

**K)** The most important contribution of the paper is the calibration of a magnitude scale (ML-SL) to the specific case of the clayey landslides for the locations of study. Also, the description of the conditions for a good study is valuable. The clarification that the magnitude scale for earthquakes is different to that used for landslides and also the differences in the ground motion parameters for their determination is a valuable contribution. A remark in this regard to the community was necessary for the proper use of seismological parameters.

We truly appreciate that comment. Thank you!

**ABSTRACT.**

**Pag. 1 L-13** The signals are generated, they are not triggered.

Modified **P1 L13** as asked.

**Pag. 1 L-15.** Include a magnitude order for the duration of the longest signals (hours?).

Specified **P1 L15** as asked: (> 2 s – several min)"

**Pag. 1 L-18** Replace ML to ML-ls in the local magnitude scale.

Modified **P1 L20** as asked.

**Introduction.**

**Pag. 2 L-17** Specify rock avalanches. Snow avalanches are not considered in the mentioned references.

Modified **P2 L20** as asked.

**Pag. 2 L-27.** (3) sensor network geometry. Specify what type of network.

We specified seismic network geometry in the complete document. This occurrence was removed from here (see next comment)

**Pag. 2 L-24.** The last part of the Introduction (from Pag. 2 L24) mostly corresponds to results or discussion. Eliminate this part from here.

We removed that part from here and included it in the discussion (Section 6)

**Pag. 2 L-28.** Split the sentence: event location. Uncertainties: : :.

Correction referring to previous comment. The sentence was removed.

**Pag. 2 L-31.** Since the uncertainty: : : Please, explain better this sentence. What do you mean by seismicity rate? Note that you are not referring to earthquakes. And the word seismicity is specific for earthquakes.

We agree the sentence was not clear.

At the considered short epicentral distances (< 100 m), a mislocation of about 50-100 m can impact the evaluation of the magnitude by as much as 3 order of magnitude units (in theory, the event could be originated in 1 m, 10 m or 100 m distance and the magnitude scale has a logarithmic dependence to distance). High uncertainties in the magnitude estimations compromise the evaluation of landslide-induced microseismicity rates. By microseismicity rates, we consider the magnitude-temporal occurrence of events generated by the landslide. We included this in the discussion (Section 6)

**Pag. 3 L-7.** mm d-1 are not in SI units. I recommend to use SI units (m. s-1 ).

We changed to **P3 L17**: "with moving rates ranging between a few mm up to several tenths of cm per day"

We indicated landslide displacement rates in cm d$^{-1}$, since this is the range of velocities observed in the difference field campaigns. m s$^{-1}$ are then not so easy to read.

**Pag. 3 L-10.** Lindner et al., 2014 and 2016 are not included in references.

We added the references and apologize. It seems that we've had a problem of project synchronization in our reference software *citavi*.

**Pag. 3 L-12-14.** Indicate the source of the displacement values or better explain how they are obtained.

We added explanations to **P3 22**:

- **Super-Sauze 2010 (SZ10)**: May 28–July 24, 2010; 58 days; 18 sensors in 2 ha; average displacement of 0.4 cm d$^{-1}$, obtained by daily dGNSS (differential global navigation satellite system) measurements.
- **Pechgraben 2015 (PG15)**: October 7-15, 2015; 9 days; 12 sensors in 6 ha; average displacement of 2 cm d$^{-1}$, obtained by weekly dGNSS measurements.
- **Pechgraben 2016 (PG16)**: November 8-12, 2016; 5 days; 12 sensors in 1 ha; average displacement of more than 20 cm d$^{-1}$, estimated by triangulation, using grids of fixed nails both on the stable and on the active part of the slide and daily photo-monitoring.

**Pag. 3 L-17.** The data were collected.

Modified **P3 L31** as asked.

**Pag. 3 L-18.** Please, include the references of Agécodagis (or Agéodagis ?) instruments.

It is indeed Agécodagis instruments and Kephren data logger, we are sorry for the misspelling. Agécodagis is a French company (S.A.R.L), located 16, rue d'Auriac, 31310 Rieux Volvestre, France. We corrected the spelling **P3 L32–P4 L1** but did not add more references as the other instruments are also only referred by their corporate naming.

**Pag. 3 L-31.** Replace the word seismic.

The sentence was modified to **P4 L13**: "This aspect must be considered when evaluating the completeness of landslide-induced microseismic catalogs."

**Line 25-26.** This sentence is related to objectives does not correspond to data. Rephrase the sentence or move it to the introduction.

We removed the sentence. The text reads now **P4 L4**: "A comparison of the data collected by the different installation systems proved consistent: identical waveforms featuring similar amplitudes are observed for microseismic events recorded at the co-located stations S1.5, S2.6 and S3.6. No significant difference in terms of waveform scattering was found for signals recorded by stations installed in the more stable areas."

**Pag. 4 L-5.** Differentiate between sonogram and spectrogram in the context of this document. Both functions show the temporal evolution of the frequencies. Depending on the parameters of the display, the same information can be displayed.

We differentiated sonograms and spectrograms. The sonogram applies a logarithmic scaling of the frequencies on the ordinate axis as well as frequency-dependent noise adaptation as described in Figure 2 of Sick et al. 2012. We rewrote the sentence **P4 L20**. "The enhanced visualization of sonograms has unmatched power to facilitate the detection and recognition of various type of weak signal energies in low-SNR (signal-to-noise ratio) conditions without a-priori knowledge (Joswig, 1990; Sick et al., 2012; Vouillamoz et al., 2016)."

**Pag. 4 L-22.** Complete this information explaining a little more the tools applied with respect to what you expect. This will help the reader.

Responded in comment E.

**Pag. 4 L-23.** Indicate the aim to design a catalogue. This could be included in the introduction. However, is design the correct word? Do you want to generate a catalogue with a good design?

Yes, we wanted to generate a catalog of microseismic observations at active clay-rich debris slides with a good design. In our opinion, a well-designed catalog (or database) gathers in a systematic way important and discriminating parameters about each event. We included a sentence in the introduction **P3 L8** "Microseismic observations were gathered in a comprehensive catalog. The final catalog of landslide-induced microseismic signals provides an initial microseismic signals library to train automatic detection and classification systems as well as an important basis for a multidisciplinary comparative analysis with other landslides observations such as displacement, cracks and fissures development, or hydrometeorological data to gain knowledge about landslide dynamics." We also modified the sentence to **P5 L1**: "Much attention was paid to design a comprehensive database gathering all microseismic signals observed by passive microseismic monitoring on active debris slides."

**Pag. 4 L-24.** Replace seismic catalog by catalog of seismic signals induced by landslides or similar.

We systematically modified to landslide-induced microseismic events / microseismicity.

**Pag. 5 L-7.** Explain why the signals are expected to be severely attenuated and give references.

We specified and added references in the sentence **P5-L27: "**The signals of landslide-induced microseismic sources are expected to be severely attenuated, because of their source proximity and their propagation through heterogenous clay-rich soils of various water saturation (e.g. Aki and Richards, 2002; Koerner et al., 1981).

**Pag. 5 L-17.** Specify to what previous studies are you referring."

We specified the references (already mentioned in the introduction)

**Pag. 5 L-26.** landslides seismicity catalogs. Better replace by Lanslides induced sismic signals catalogs.

We modified seismicity systematically in the text.

**Pag. 5 L-32.** The content of Fig. 2 should be explained in the text. In addition, somewhere in the text should include an explanation of the selection of the tools used in the Figure, their role and benefit. As far as the FFT is concerned, this is not so trivial. Are you using the spectral amplitude or the PSD (power spectral density)? The FFT of a function has a real part and an imaginary part (Phase). In addition, the units are not specified in the figure.

We explained Figure 2 in the text as an introduction to Section 4:

"To help the reader in the comparison of the different microseismic signals, we apply the layout of Figure 2 for all representative events of the classification (where only vertical traces are presented):

      a. Displays the signal sonogram (Joswig, 1990) up to the Nyquist frequency with a logarithmic ordinate, which corresponds to 1.95-250 Hz for Pechgraben data and to 3.91-500 Hz for Super-Sauze data. Darker colors indicate higher relative energies.

      b. Shows the non-logarithmic spectrogram of the signal, with an ordinate up to 250 Hz. The time-window is taken as the signal length divided by 30 and an overlap of 90 % was applied. Red colors indicate higher energies. Both the MATLAB® spectrogram code and colormap were provided by Clément Hibert, of the EOST (Ecole et Observatoire des Sciences de la Terre), University of Strasbourg, France.

c. Displays the unfiltered seismogram with maximum absolute 0-to-peak amplitude indicated above the trace in nm s$^{-1}$.

d. Provides a selection of bandpassed waveforms in nm s$^{-1}$, filtered from bottom to top between 1-5, 5-20, 20-50, 50-100 and 100-200 Hz using a second order Butterworth filter. Maximum absolute 0-to-peak amplitudes are indicated in nm s$^{-1}$ above each respective trace.

e. Provides the amplitude spectrum in nm Hz$^{-1}$, computed by FFT for the time window indicated by the red bar in (d).

**Pag 6 (4.4.2) and Fig 3.** You affirm that in type II there are surface waves, I also observe in Type III similar shapes of better developed surface waves. This is normal because distance is longer. In the case of local microearthquake (Type IV) I observe perfectly the surface waves at 3 s from the first arrival. The energy immediately 0.2 s are S waves (although is a vertical component).

Yes.

**Pag. 6 L-14.** Indicate the reasoning to assume that the distances are > 50 m or other in the different types of signals.

We removed numbered distance estimations from here and included this in the discussion (Section 6).

**Pag. 6 L-15.** SI units, 10-5 ms-1.

We leave it to the editor to decide if all reference to velocity units in the paper must be changed to m s$^{-1}$. We consider that information indicated in nm s$^{-1}$ facilitates the lecture as well as the comparison of various signal amplitudes, as amplitudes of microseismic signals in environmental seismology usually scale in nm s$^{-1}$.

**Pag. 6 L-18.** What do you mean by later phases? Later arrivals? If they correspond to surface waves, the speed is lower, but if they are wave phases that travel in depth, the velocity must be greater although the distance affects the apparent speed. ... Explain this better or delete the parenthesis. We delated the parenthesis. The seismograms in figure 3 are presented following the stations code, but please, indicate which the time origin of the seismograms is. The apparent velocity is in relation to the distance source-sensor and this is not indicated. Or is it unknown?

The apparent velocity of a wave front can be computed in HypoLine, by using relative wave-packets peaks arrivals in the records of seismic tripartite arrays. The procedure is illustrated in Figure 5b of Vouillamoz et al. 2016. In the Figure 3, the seismograms are all plotted on a same time line. Fig 3a represents less than 1 s of signal.

**Pag. 6 L-22.** Indicate the figure in detail (for example, Fig. 4c-5), when referring to the dominant frequencies.

We specified in the text as following for all similar comments (see spectrogram, bandpassed waveforms and amplitude spectrum in *Fig.xx*)

**Pag. 7 L-4.** Please, indicate the epicentral distance. It seems to me that P and S waves and also superficial waves are observed clearly. See Fig 3d.

Yes, we agree that P, S and surface waves are observed. This indeed suggests a location origin outside of the recording seismic network, rendering the epicentral distance quite uncertain to approximate. We modified the sentence **P8 L23** accordingly**: "**P- and S-phases can be identified."

**Pag. 7 L-5-15** this could also be part of the conclusions. Moreover, you obtained similar signals PI and PII at both sites (SZ10 and PG6). This is remarkable.

Yes, it is remarkable! We mentioned this in the reworked conclusion **P20 L10**: "Despite the complexity of the waveforms, similar landslide-induced microseismic signals were detected at both landslides, thereby suggesting that comparable microseismic source processes are taking places at different landslides and that the method is therefore scalable and reproducible."

**Pag. 7 L-7.** Explain why a large attenuation is evidence of a nearby source.

We see it empirically. We removed that part in the discussion and reworked Section 5.2. where calibration data are discussed according to the comment 10 by Reviewer #2.

**Pag. 7 L-9- 10.** Please explain why you infer these statements. Nearest source, small source ... This could be discussed in the introduction or elsewhere.

We removed that part from here and better specified in the discussion (Section 6)

**Pag. 7 L-12-16.** Some explanation or references are needed to support the assertions.

We removed that part from here and better specified and referred in the discussion (Section 6)

**Pag. 7 L-23.** The mentioned publications are not indicated in the references.

Again, we apologize for the synchronization problem in the references. We added the references.

**Pag. 7 L-25.** The mentioned publications are not indicated in the references. In addition, in figure 5 in Biescas et al, 2003 (Surveys in Geophysics, 24, 447-464) is the first time that helicopters and moving vehicles are mentioned in this context. In relation to the Doppler effect, it is necessary to vary the speed of the mobile source with respect to the stationary observer and an acceleration to observe a change in the frequencies that vary in time in the spectrogram (gliding).

We added Biescas et al. 2003 in the references for gliding harmonics/helicopters.

**Pag. 7 L-27.** The correct word is generated, it is not triggered.

Modified as asked.

**Pag. 7 L-28.** What is the reason for including this sentence in relation to the Q factor? Presented in this way does not contribute to anything. Remove it.

We removed the end of the sentence.

**Pag. 8 L-7.** Specify seismic network here and in all the cases.

Modified as asked in the complete document.

**Pag. 8 L-13.** Replace 4.3.1 by 4.3.2.

We corrected the numbering.

**Pag. 8 L-15, 16 and 22** Replace are by were.

We did not modify to past because the sentences are affirmations.

**Pag. 8 L-24.** Similarly to previous sections accompany the subtitles with a dash e.g. - ETS-like signals (episodic tremor and slip); - Confirmed rockfall events: : :..

Modified the style to bullet as asked.

- ETS-like signals. How do you deduce that the signals correspond to episodic tremors or slip? Is it related to previous studies? You have to mention that in this subsection.

We referred to Gomberg et al. 1995 and 2011, which were the first to consider the analogy between ETS in fault zones and landslide-induced tremors at the strike-slip shear zones of the landslide in the previous section 4.3.1,

**P9 L6** reads now: "Various tremor-like signals were observed at clay-rich instabilities. Gomberg et al. (1995) and Gomberg et al. (2011) report episodes of tremor-like radiation and sinusoidal waveforms lasting tens of

minutes and coherent across the seismic network, which they infer as ETS (episodic tremor and slip) analog of strike-slip faults."

We also modified **P10 L14**: - "Seismic signals showing similarities to ETS signals at strike-slip faults were observed."

**Pag. 8 L-25.** seismic network.

Modified as asked in the complete document.

**Pag. 8 L-26.** fig. 5b Specify better and give more details.

We provided more details in the bullet ETS-like signals.

**Pag. 8 L.31**- Fig. 6b Specify better and give more details.

We specified better the reference to the figure. The subsection about confirmed rockfall was reworked. Interpretation were removed to the discussion part. This applies also to other tremor like signals.

**Pag. 9 L-6**. Do you mean low topography or smooth topography? Perhaps a short description of the sites is necessary.

We rephrased (smoothed topography) and moved that part to the discussion (Section 6)

**Pag. 9 L-10**. This subsection is not very clear. No description of the signals is presented.

The only result is that since they are very similar to ETS they can be eliminated if there is a field observation.. However, as the physical process of rocksfalls (impacts) and slip is different, if the signals are observed in more detail they can possibly be separated. Additionally, if you consider the 3 components you can observe differences between a slip and a rockfall. See fig. 3 in Vilajosana et al., 2008. The signals are different in the horizontal and vertical components.

As mentioned the response letter, we do not trust at 100% our horizontal component data. Of course, we also see differences in our 3-C data, as it is observed in Figure 3 of Vilajosana et al. 2008, and for some high-quality events, the differentiation can be done. However, for most of the low-SNR signals, it is difficult.

This has been removed and better specified in the discussion (Section 6)

**Pag. 9 L-13**. Fig. 5a and 5c Specify better and give more details.

We specified and referred better to the figure numbering in the harmonic tremor subsection

**Pag. 9 L-17**. Fig. 5a -c? Specify better and give more details.

Yes, we refer to Figure 5a-c. We specified and corrected in the text.

**Pag. 9 L-20**. Figures 5d and 6d show two signals from Walter et al., 2012. However, no reference is made to these figures in the text. A discussion or comparison of the signals with those of this contribution is needed.

Figures 5d and 6d are now referred in the text. We observed signals similar to the ones of Walter et al. 2012. We decided to show signals from the Walter et al. 2012 study to present the variety that the signals can have depending on the source distance and the event size. The discussion of the signals from Walter et al. 2012 is provided in the previous section 4.3.1.

**Pag. 9 L-24**. In Fig 7 there is quantity of information that is not explained in the text. More description of the images is needed. Maybe I am lost, but how do you know the source situation?

We agree that there is a quantity of information provided in the Figure 7. We dedicated a poster to these signals alone at the EGU 2017 (Vouillamoz, Naomi; Rothmund, Sabrina; and Joswig, Manfred, 2017. Passive seismic monitoring of propagating seismic sources triggered by creeping landslide at Super-Sauze (Southeastern France) and Pechgraben (Upper Austria). The file is available in research gate:

https://www.researchgate.net/publication/316480558_Passive_seismic_monitoring_of_propagating_seismic_sources_triggered_by_creeping_landslide_at_Super-Sauze_Southeastern_France_and_Pechgraben_Upper_Austria

We rephrased, better specified references to figures in the subsection about dispersive tremors and added the source distance specification in Figure 7a.

**Pag. 9 L-26**. Why does the temporal evolution of dominant frequency content suggest a mobile source? Indicate the appropriate figures and better specify the information you want to communicate. Also indicate references.

We added Biescas et al. 2003 as a reference. In the dispersive events, you see a temporal shift of the main energy content in between different stations. This is not observed for quake events or for other tremor-like radiations (compare Figure 7 with Figures 3 and 5). However, such a shift is observed for example in snow avalanches signals (see for example Figure 11b in Biescas et al. 2003) or signals generated by a person walking across the seismic network (see Figure 8a). We modified the text as indicated in previous comment.

**Pag. 10 L-1**. The information in this subsection is anecdotal and unnecessary. It does not contribute anything to the knowledge of the phenomenon. These signals are context-independent. And its characteristics depend on the distance from the sensor to the seismic source. Always, when the location of a station is inappropriate, there are external sources and noise. Fortunately, as mentioned in the text, in some cases they are in the high frequency range and a high pass filter eliminates it. As it is impossible to collect all kinds of signals, I recommend that the authors mention the possibility of registering external sources and eliminate this part with the corresponding figures and devote the space to analyse the signals produced by landslides.

We disagree to that comment. Of course, it is not possible to collect all kind of signals. However, we want to present the most common ones. External sources of noises constitute the major part of the observed signals. This is also mentioned by other studies (e.g. Gomberg et al. 2011; Provost et al. 2016). In many instances a high pass filter is not sufficient to eliminate the high frequency noises. Indeed, most of these signals also show energies below 50 Hz and could then be misinterpreted (see band-passed waveforms in Figure 8). We therefore left that subsection and the related figures in the text.

In addition, we added a sentence **P10 L11**: "Anthropogenic noises can share similarities in waveform amplitudes and in spectral content with landslide-induced tremor signals. It is therefore important to gain knowledge about the characteristics of such events for the manual and automatic detection of landslide-induced tremor signals."

**Pag. 10 L-28**. Is the coda important for the location? No! In addition, as indicated, the attenuation is very high, so the coda disturbs little.

Answered in the response letter and comment C.

**Pag. 11**. All the points related to geometry and distribution of the elements of the network with regard the good location of the source are well known in seismic location. This is not new. The only doubtful point is the low impact of the velocity model to locate the superficial source. Normally velocity model affects locations. Are significant these mentioned few meters with regard the distances involved? I propose the authors to construct travel time curves for the direct and reflected waves in the range of 150 m for the 3 velocity models to see the effect in the travel times and discus the result. Additionally, you are using only 4 readings. Why? Usually, more readings are necessary to obtain an over-determined system. Comments on all this are necessary.

Answered in the response letter and comment C.

**Pag.11 L-31**. Seismicity map is not an appropriate term. Seismicity map is related to the distribution of earthquake epicenters. You are dealing with other events. Perhaps a good term is "map of landslides related events location".

We agree and modified to **P14 L14**: "Consequently, the risk of including biased data in maps of landslide-induced microseismic events is high."

**Pag.12** Estimation of source proximity trough waveform attenuation. This is an option, but Fig. 11a shows local site effects, which are also observed in the dispersion on Fig. 11b. It's not so easy for me to validate the distances of the source. Explanations or references are necessary.

We agree there are site-effects, of course. However, we do think that the estimation of the source distance through attenuation pattern is better as a standard localization for landslide-induced microseismic events originated near the seismic network (< 50 m) - for which uncertainties reach several 10's of meters in the best cases - even in the case of site-effects. Maximum absolute vertical traces 0-to-peak amplitudes of local and distant earthquakes never reach S values above 200% (see the definition of S value in the new section 5.2) as shown in the graphic below:

[Figure]

NB: station numbering corresponds to S1, S2 and S3 array. Station number 13 corresponds to station S3.3 at SZ10 with corrupted records during the 8-week field campaign.

**Pag.12 L-14.** seismicity rate is an inappropriate term. Perhaps Rate of landslide related events.

We agree and modified to **P15 L10**: "… and are therefore used in the analysis of landslide-induced microseismic events rates (see Section 6.3)."

**Pag. 12 L-16. 5.3** I agree with the authors that it is necessary to remind the origin of the magnitudes to avoid an inadequate use of this concept. Very interesting precision regarding the differences between magnitude scales.

Thank you!

**Pag. 12 L-17.** Specify: Earthquake local magnitude scale ML.

Modified as asked.

**Pag. 12 L- 24**. Fig 12 is the correct Figure.

We corrected the figure number reference.

**Pag. 13 L-1.** ..define Ao-LS and indicate its units.

$A_{0\text{-}LS}$ is the distance attenuation function in landslides. We indicated it in the text.

**Pag. 13 L-1.** is logfit a MATLAB script? Indicate.

Yes, it is a MATLAB script, we indicated in the text**.**

**Pag. 13 L-7**. In units SI is KJoule not KiloJoule.

We corrected it**.**

**Pag. 13 L-11**. Better to express A= 5.10 -3 m/s (SI units).

We left it in nm s$^{-1}$ as this is the unit requested in the $M_{L\text{-}LS}$ function.

**Pag. 13 L-14**. A is the absolute value (include <0 values).

We specified consistently in the text, maximum absolute vertical trace 0-to-peak amplitude

**Pag. 13 L-17**. Homogenize terminology in Pag.12 L-19 and here.

We homogenized the terminology.

**Pag. 13 L-26**. Consider my previous comments on the use of the term seismicity.

Catalogs of clayey landslides events.

We modified to 6. Discussion of microseismicity catalogs at clayey landslides

**Pag. 14 L-1 and L-4**. Consider my previous comments on the use of the term seismicity. Landslides-induced events or landslides-induced seismic events are more corrected terms than seismicity.

Answered in previous comments.

**Pag. 14 L-5. Section 5.4.1** is too short. I recommend the authors expand the explanation to give value of the results. Figure 14 shows amount of information that needs comments. Pag. 14 L-5. Consider my previous comments on the use of the term seismicity. landslides-induced seismic events rates. Although, I recognise is a long title. In addition, what do you mean by seismicity rates, activity of seismic events?.

A Discussion section on the results is needed. A comment on the useful parameters and methods used. As it is indicated in conclusions information related to the frequency content and its time evolution was non relevant. Please, an explanation on this is needed. Also relate your outputs to previous results.

We agree. We reworked and completed the discussion as asked and added the Section 6

**Pag. 14 L-7**. indicate Fig. 14 a and c.

Modified as asked**.** Figure 14 was modified, see comment on **Figure 14**.

**Pag. 14 L-10**. SI units (?).

Modified to cm d$^{-1}$

**Pag. 14 L-13**. landslides-induced events.

Modified to landslide-induced microseismic events.

Pag. 14 L-14. Conclusions and outlook .

Split the content and include it in the discussion section to give it more.

We removed the discussion part from the conclusion and put it in the discussion, we focused the conclusion (Section 7) on the outputs and the future directions of this studies that includes the correlation of seismic data with high repetition rate displacement data (dGNSS / GB-InSAR), daily orthophotomosaics and photo-monitoring, hydrometeorological data…

**Pag. 15 L-5**. indicate ML-LS.

We specified $M_{L\text{-}LS}$.

**Pag. 15 L-6**. Where are the mentioned displacements?

We use average daily displacement rates, indicated in the Figure 14e and specified now in the text of Section 2 for each campaign.

**Pag. 15 L-24**. Consider my previous comments on the use of the term seismicity.

We modified consistently the use of the term seismicity.

**Figures and Figure captions.**

**Figure 1.** Seismic arrays Si should be presented before. Not in e) We referred arrays in Figure 1c-d. A better explanation of the colours will improve the understanding of the figure. We indicated that colors refer to individual tripartite array. Higher and lower dynamics of what? We removed specification to lower and higher dynamics as this is not discussed anymore in the text. To help the reader, indicate what 3C seismometers are. S1.1, S3.0 and S2.1 in SZ (c) and S1.1, S3.1 and S2.1 in PG (d) We specified in the Figure caption the three component seismometers.

**Figure 2.** -Show first, in a), your time series. These are your data. Indicate that is the vertical component

We did not modify the Figure layout sequence because our methodology starts with sonograms.

-a) sonogram of the vertical component. Explain the sonogram in the text and not in the legend of the figure; the same for the spectrogram.

Answered previously.

-e) Specify amplitude spectra or PSD instead of FFT and indicate the units.

Answered previously.

-Include the colour scales in a) and b). There is certain incongruence between the information for the sonogram and the spectrogram. I do not understand what the utility of the spectrogram is.

The color bars are relative, we specified the color scale in the text, in the introduction of Section 4:

   a. Displays the signal sonogram (Joswig, 1990) up to the Nyquist frequency with a logarithmic ordinate, which corresponds to 1.95-250 Hz for Pechgraben data and to 3.91-500 Hz for Super-Sauze data. Darker colors indicate higher relative energies.

   b. Shows the non-logarithmic spectrogram of the signal, with an ordinate up to 250 Hz. The time-window is taken as the signal length divided by 30 and an overlap of 90 % was applied. Red colors indicate higher energies. Both the MATLAB® spectrogram code and colormap were provided by Clément Hibert, of the EOST (Ecole et Observatoire des Sciences de la Terre), University of Strasbourg, France.

-The information of the spectrogram and sonogram must be the same with an appropriate scale and a correct overlap. Yes.

-Moreover, what is the role of the different filtered bandpass signals if you do not mention them in the text?

In our opinion, the bandpassed waveforms help to quickly evaluate the signals. The higher band-pass waveforms provide information about landslide-induced microseismic event proximity.

For example, in Figure 4, it is well observed that the energy content in higher bandpass decreases with increasing distances (or event types).

- Also, because the frequency content of the earthquake is up to 40 Hz, showing the scale up to 250 Hz is meaningless and leads to loss of information. Similar comments for figures 4 and 6.

We wanted to provide the reader with easy to compare figures. This is the reason why we applied the same scaling on each diagram and whenever possible, used similar time window for event type of similar categories.

**Figure 3.** What is the disposition of the seismograms? Which criteria have you follow?. Replace rectangles by bands.

We used the station numbering, starting with the tripartite array number 1, in order to see the signal over the complete seismic network. Empty traces indicate missing or corrupted data. We specified this in the Figure caption. We modified the rectangles to red dashed lines.

**Figure 4.** Number columns 1-5 in each row to help the description in the text, Could you explain the difference in the frequency content of the sonogram and the spectrogram? Is there a filter between the two calculations?.

We referred the figure numbering more specifically in the text. Sonogram and spectrogram are now better explained, see comment E and section 3.

**Figure 5.** Indicate that the number of the stations is on the right. As regards the amplitude, note or specify that it is the absolute value. Events a) and c). We specified absolute values consistently in the text.

We also modified the stations indication position to the left and the axis label for consistency with figure 3.

**Figure 6.** Consider my comments for figures 2 and 4. Answered in previous comment. Indicate the distance to the sensor of the confirmed rockfall to help readers. We indicated the receiver-source distance (29 m) in the caption. Are the signals reproducible and repetitive? A comment on this is necessary in the text.

We specified in the discussion Section 6. **P18 L2: "**Despite many landslide-induced microseismic events were observed to occur in sequences, thus suggesting a potential common source process, a cross-correlation analysis performed in the time domain within 1-30 Hz returned no evidence of similar events among the considered sequences."

And in the conclusion **P20 L11**: "Despite the complexity of the waveforms, comparable landslide-induced microseismic signals were detected at both landslides, thereby suggesting that similar microseismic source processes are taking places at different landslides and that the method is therefore scalable and reproducible.."

**Figure 7.** Consider my comments for figures 2 4 and 6. Answered in previous comment. We added information about the probable receiver-source distance in Fig. 7a.

**Figure 8.** This figure shows some of the external sources, the series is not complete and the signals depend on the cases and distances. It does not contribute anything to the paper. Remove it.

Answered in comment **Pag. 10 L-1**.

**Figure 9.** Consider the comments in Figure 8. Remove it. If you present these signals, you should explain their characteristics fully in the text, not just showing the plots. Answered in comment **Pag. 10 L-1**.

**Figure 11.** If I understand correctly, data in a) correspond to experiments showing different amplitudes for similar distances. Yes, we carried out one SISSY at seismic stations S2.1 S2.2 S2.3 S2.6/S1.5 and one SISSY close to station S3.0 at SZ10 and several hammer blows at each station as well as offset blows at specific places in the seismic network at PG16.

(e.g. squares in 40-60s in a) possible due to local effects. At SZ10, the stations at 40-60 m distances of the SISSY shot on 06.07.20 in red squares are most probably tilted, the signals are of low quality on these stations. How do you correct the site effect in this process? An explanation is needed in the text. We did not correct for site effects since amplitudes observed for regional earthquakes or teleseismic earthquakes are very similar (see for example Figure 3e, bottom panel, for a local micro-quake and discussion and additional Figure in the answer

of comment **Pag.12.**). The discussion about waveform attenuation has been completed (See comment 10 by reviewer #2. In addition, we added in the conclusion **P20 L20:** "Since S values of local and regional earthquake stay systematically below 200 %, we did not correct for potential site-effects and $M_{L-LS}$ were computed for near-source area events (< about 50 m), applying a distance attenuation function calibrated for clayey landslides (Section 5.3)."

Replace dotted lines in the figure caption by dashed lines.

Dashed lines modified as asked in the figure caption.

**Figure 12.** a) Specify that it is for regional earthquakes

We rephrased to: "(a) Distance attenuation functions (-log(A0)) of $M_L$ scales empirically calibrated for regional earthquakes with receiver-source distances between 10 and 600-1000 km."

**Figure 13.** Better to place empty symbols because grey is difficult to see. Replace the word seismicity.

We replaced the grey symbols to empty symbols and replaced the word seismicity to microseismicity.

**Figure 14.** Figure very interesting that corresponds to the results. More explanations in the text are needed. Replace the word seismicity.

We replaced the word seismicity.

The Figure 14 was modified. We removed precipitation and displacement data for clarity since we do not discuss this in the text anymore. The motivation for this change is that our paper discusses the events classification. The final catalog provides the basis for a further multidisciplinary study, where the seismic data are correlated to other geophysical information for a better interpretation of landslide dynamics. This analysis was the central topic of a PhD thesis, submitted (S. Rothmund, 2017, see response letter) and will provide the material subject of a new dedicated paper.

**Comment by A. Köhler**

**General comments:**

The authors do a great job analyzing and categorizing the seismic emissions of very complex source processes. It is a laborious task, but essential to be able to use seismic monitoring as a reliable tool to forecast the behavior of debris slide. The authors try to establish an objective and consistent processing scheme, which of course has its limitations for such a complex seismic record. The paper is well-written and presents in most parts conclusive and comprehensive results, giving all the necessary details.

However, some clarification are necessary and there are issues that should be discussed in more detail. My main concern is the temporally varying completeness of the seismic event record resulting from the manual processing scheme applied. This could have important implications on the interpretation of how well the seismic rates correlate with measured displacement of the slide.

We agree some clarifications were necessary and have responded and discussed more specifically the temporally varying completeness of our microseismic database; see comment (2) below.

**Specific comments:**

**(1)** Page 4, section "3 Methods": I would suggest to remove the sub-section 3.1 since only a single method is introduced.

We removed the sub-section, and reworked and completed the chapter 3 as also asked by reviewer # 1.

**(2)** Only sonograms on night time measurements were screened to minimize false detections: What implications does this have on the fact that you use the seismic rate in the end and compare it to slide displacement? Or does

it not matter because the displacement data has daily resolution (see also comment about Fig. 14 below)? Is there any process that could lead to an increase of seismic emissions from the landslide body during daytime which you would miss (e.g. diurnal temperature forcing)? Is there a changing noise level at nighttime that could affect the observed seismic emission rate (change of "visual" detection threshold)?

We apologize for the misunderstanding. We did screen day-time measurements, however, paying attention when screening data contaminated by the noises produced by the geophysicist activity on the slope. Due to the higher noise level during the day, the catalog of landslide-induced microseismic events may be uncomplete during the day. However, this does not impact our interpretations, as displacement rates are measured at best on a daily (24h) basis. We don't have a sufficiently robust statistic dataset to derive conclusion about for example diurnal or seasonal temperature forcing. The noise level is of course lower for night time measurements; however, the dynamic frequency-dependent noise filter of the sonogram provides a powerful tool to detect events in very bad SNR conditions.

We rephrased the sentence **P5 L1:** "Much attention was paid to design a comprehensive database gathering all microseismic signals observed by passive microseismic monitoring on active debris slides. Continuous sonograms of the three seismic datasets (SZ10, PG15, PG16) were visually screened in SonoView. To avoid false noise detection, special attention was paid when screening day-time measurements contaminated by anthropogenic noise caused by geophysicists or geotechnical work carried out on the slope. Only signals recorded coherently by three sensors at least were considered as a detection."

**(3)** Features presented on Page 5: Maybe it is worth a try to use an automatic clustering method in the feature space. That would open up the possibility to use some sort of STA/LTA network trigger and then perform post-classification afterwards. Showing some scatter plots would help to evaluate if the feature distribution from all events actually shows a clustering related to your event classes, or if the classes rather represent end-members of a continuous distribution. Even if implementing an automatic clustering method is beyond the scope of this paper, the potential of such an approach could be discussed.

The five criteria enumerated in **P5 L25** are a minimal subset of the evaluated high-dimensional feature space, which is best represented in the graphical display of sonograms. We wrote Section 6.1:

**6.1 Landslide-induced microseismic events detection and classification**

Automatic detection algorithms work fine for well-known routine seismic signatures but fail for unknown and unexpected low-SNR microseismic events. In order to gain knowledge about the existing types of landslide-induced microseismic event signatures, we therefore used an enhanced visualization alternative, where continuous seismic data were screened for visual pattern recognition in the form of sonograms (Joswig, 2008; Sick et al., 2012; Vouillamoz et al., 2016). Using a minimal number of seismic features, detected events could be gathered into three main groups (Section 4): earthquakes, quakes and tremors signals. The shallow installation of seismic stations in the landslide body resulted in high level of noise contamination of the data and the distinction between landslide-induced microseismic events and other environmental (or anthropological) sources was not straightforward. Due to the near-source area of the targeted microseismicity, individual source signal seismic signature can show tremendous variations between records from different stations, depending on the respective receiver-source distance (e.g. Figs. 3a-b, 5a-c and 7). Despite many landslide-induced microseismic events were observed to occur in sequences, thus suggesting a potential common source process, a cross-correlation analysis performed in the time domain within 1-30 Hz returned no evidence of similar events among the considered

sequences. This stresses out the complexity of the signals radiated by near-source area microseismic processes and hence the variability of the related seismic feature space. Individual microseismic sources can also occur simultaneously on a complex debris-slide, thereby leading to time-overlapping tremor signals with hybrid characteristics, where individual source radiations cannot be unambiguously separated. Several instances of quakes doublets (type II and III), similar to short-duration ETS-like signals were observed at both landslides. At Pechgraben, frequent near quakes (type I and II) featuring short duration harmonics were observed. Thus, it can be concluded that an unequivocal manual or automated classification of landslide-induced microseismic signals is possible for well-defined landslide-induced single microseismic events. This is supported by previous results of Provost et al. (2017) at Super-Sauze landslide, where quake and rockfall microseismic events could be successfully detected and classified using a Random Forest supervised algorithm trained with 71 seismic features on a large training set. However, inputs from the analyst are still a requisite in the analysis of complex, hybrid tremor signals recorded at active clayey landslides in order to obtain comprehensive and robust landslide-induced microseismic events libraries for the training of automated detection systems and classifiers.

**(4)** Array processing: How reliable is beamforming here given that source-receiver distances are rather short compared to the array apertures (non-planar wavefronts)?

Based on SISSY shots and hammer blows calibration datasets, we see that when the source is near ($< 50$-$100$ m) to the tripartite seismic array used for the beam calculation, the back azimuth can still be constrained to about one quadrant. However, if the source is within the tripartite array, no solution is found! The applied sampling rate does not affect these results: no differences were found for data sampled at 500 or 1000 Hz. Based on other personal experiences, a sampling rate at 250 Hz returns also the same order of values. Therefore, a beam provides a directional location constraint of one about one quadrant, which makes tripartite array a useful tool in environmental seismology.

**(5)** Page 8: 4.3.1 change to 4.3.2.

We corrected the numbering.

**(6)** Page 6: If Type IV are most likely events outside the landslide, wouldn't it be better to describe them in section 4.1. (local earthquakes)? In this case section 4.2. "quakes" could be renamed to something like "seismic landslide signals" or "slidequakes" (which has been used previously).

We agree with the comment and modified the term micro-earthquake to micro-quake for clarity and consistency. The goal was to provide the reader with a continuum of 'quake events', from very near landslide-induced microseismic events recorded only at a few nearby stations and local micro-quakes recorded consistently by the complete seismic network. At recording receiver-source distances of more than about 100 m, it is difficult to discriminate if the source is related to the landslide activity or not.

We reworked the section 4.2.2 and introduced it as following **P7 L22**:

"Based on waveform attenuation pattern, dominant frequency content and duration of the signals, we propose four types of quake events which represent a continuum between very near-source area quake events recorded only at a few nearby stations to local micro-quake events recorded consistently across the complete seismic network (Table 1; Fig. 3 and Fig. 4)."

**(7)** Page 10, line 9: Since you wrote you just used nighttime records for manual data inspection and there are signals corresponding to "geophysicist walking": Is this consistent? (Of course geophysics might also work at night sometimes : : :) we actually worked at night, in winter the evening comes early ☺

We responded to this issue in comment 2.

**(8)** Page 11: Please explain what you mean with "the domain of existence of the hypolines was tested ..."

This sentence was indeed unclear. We observed the variability of the mathematical solution represented graphically by the hyperbolae. We rephrased **P13 L30** "At ± 20 samples (± 0.02 s), no more mathematical existent solution is found."

**(9)** Page 11: Line 23-27: See previous comments (consider non-planar wavefronts in addition to scattering and inhomogeneities).

We added receiver-source distance information in the text and reverted the order of the bullets for clarity **P14 L5**:

- Complex velocity structures and resulting waveforms scattering impedes array-processing and back azimuth information can be significantly biased. The calibration datasets at Super-Sauze and Pechgraben derive uncertainties in the order of one quadrant (± 45°) for well constrained beams (using high correlation values of four and more coherent waveform spikes), for source located at 50-100 m outside of the seismic mini-array.

- Sources originated within the seismic network return incoherent array-processing and back azimuth data.

**(10)** Page 12, Section 5.2: "normalized difference between the maximum amplitudes of the signals and the median value of all maximum amplitudes": This is a bit unclear. I would suggest to add an equation to define your variable "scatter about the median amplitude". Then, you could refer to this quantity simply with a letter. Also, what is the percentage of this variable being larger than 200% for each of the 4 event types? Here you just give percentage of all events (type I-IV (?) quakes and tremors). I would expect a clear difference for Type I and II compared to Type IV.

We totally agree to this comment and thank you. We introduced the value S with a formula, and then used it consistently in the discussion for the interpretation of different landslide-induced microseismic event source type.

The section 5.2 was rewritten and S values were added in Table 1:

**5.2 Waveform attenuation pattern to estimate source proximity**

Because of the high uncertainties returned by standard seismological approaches to event location, the drastic attenuation of waveforms observed within the landslide body was used to constrain the source proximity of near-source area landslide-induced microseismic events to be used in the calculation of events local magnitude. Distance attenuation data of SISSY calibration shots and hammer blows at Super-Sauze and Pechgraben show that signals are strongly attenuated within the first 50 m. The water content of the landslide material influences the waveform attenuation: signals are less attenuated when dryer conditions prevail (Fig. 11a). This observation is consistent with laboratory experiments (e.g. Koerner et al., 1981). To quantify the waveform attenuation pattern of an event, we use the scatter about the median amplitude, S, which we compute for each trace that recorded the signal (Eq. (1)):

$$S = \frac{A_{sta} - Med(A_{sta})}{Med(A_{sta})} \times 100 \%$$

(1)

where $A_{sta}$ is the station maximum absolute vertical trace amplitude of the signal in nm s$^{-1}$ and Med($A_{sta}$) is the median value of all $A_{sta}$ where the signal was recorded. S values computed for the calibration dataset of Figure 11a show a drastic diminution with increasing receiver-source distances (Fig. 11b). Based on these observation,

we use maximum S values of landslide-induced microseismic events to approximate receiver-source distances. We infer S values higher than 200 % to correspond to receiver-source distance of less than about 50 m. This is consistent with the observation that local and regional earthquake never return S values above 200 %. At smaller distances, we selected thresholds (in an arbitrary, but very conservative way) of 1,000 % and 2,000 % to correspond respectively to receiver-source distances of about 20 m and 10 m from the recording station. The source distance of natural events for which S values remain below 200 % is considered as uncertain. Among the inferred landslide-induced microseismic events (quakes and tremors), 28 % of events at SZ10, 42 % at PG15 and 39 % at PG16 feature at least one station with a scatter about the median amplitude value above 200 %. With estimated source-receiver distance of less than about 50 m, these events can be reasonably assumed as originated within the landslide body or at its edges and are therefore used in the analysis of landslide-induced microseismicity rates (see Section 6.3).

**(11)** I suggest to discuss some potential methods for automatizing the event detection. You mentioned template matching. What about cross-correlation of (array) envelopes instead of waveforms? See also comment above about clustering. It could be an option to introduce a separate discussion chapter which includes the issues of catalog completeness, automatization, and correlation of event rate with slide displacement.

We agree with that comment and discussed it in the review letter. In our opinion, the automated detection is not the issue. Nowadays, automatic detection algorithms succeed in detecting all kind of signals. The issue is the automated classifier for complex hybrid signals. We developed and specified in Section 6.1.

**(12)** Table 2: Is this Vp or Vs?

We specified $v_P$.

**(13)** Fig 3: Why are Type II events absent for stations S1.1-S1.5 ?

The stations were tilted or the data was corrupted. We specified in the Figure caption.

**(14)** Figure 10d: This panel is a bit hard to read, especially because you use two colors (red and black) with two different meanings (Red for picked stations and model 2). Maybe I misunderstand, but it looks like the hyperbolas intersect always close to the center of the array used for beamforming, but the true location is outside the array. Is this coincidence and what could be the reason?

It is not coincidence; when working with tripartite arrays in place of a network, the observation of parallelized hyperbolae, crossing at low angle, provides the direction of the beam. The zone of intersection then moves away from the array when forcing the solution in the depth profile.

We agree that the panel was difficult to read with the default colors of the HypoLine software. We thus changed the colors: picked stations are labelled in black, and non-picked stations in grey; velocity models are displayed by orange to brown colors and beam in light blue.

**(15)** Figure 11: It is not clear how you defined the three thresholds. I can follow the choice for T3, but T2 and T1 seem a bit arbitrary. Please explain in more detail.

It is unfortunately arbitrary. However, we choose T2 and T1 in a very conservative way, in order to avoid a bias in the high magnitude events (better discard events than having one too much). In that sense, we consider that the landslide-induced daily microseismicity rates are reliable.

**(16)** Figure 14: The comparison between seismic rates and displacements from this figure is very difficult, especially to prove your statement on Page 14, line 19-20, that seismicity rates show a clear increase with increasing displacement. Consider to increase the size of the sub-figure. Also, red symbols (slidequake) may be

hidden by tremor symbols. As far as I understood, all events only occurred at nighttime. In this case you could indicate the data time periods you did not screen. Please also indicate where you see correlation between seismicity and displacement. I suppose "endogenous events" is the same as "land-side induced events" and includes both tremors and slidequakes?

We agree this was too much in one Figure. We decided to remove the more detailed interpretation as previously discussed in the response letter and in comments Figure 14 by Reviewer #1.

---

## Referee Report (RR1)

Comments on the new version of the paper

**Characterizing the complexity of seismic signals at slow-moving clayrich debris slides: The Super-Sauze (Southeastern France) and Pechgraben (Upper Austria) case studies by Naomi Vouillamoz, et al.,**

In this new version the authors have considerably improved the manuscript. The present version is much better organized and the aims have been clarified.

It is a shame that in the reorganisation of figure 14, the authors have omitted information on precipitation. I hope this will be explained in the future together with an explanation of the situation. Now, it is clear that the aim of the paper is only to classify and to catalogue the seismic signals.

**As regards this, I suggest a change in the title of the paper.**

The incorporation of the S value in section 5.2, as suggested by Andreas Khöler, clarifies the results. Good suggestion.

I suggest that you should include a reference or a discussion on seismic array configuration optimization. This is well known in earthquake studies, but perhaps it is worth emphasizing this here.

Further comments

Pag 1 Line 25. I am not sure that this reference must be included here.

Pag 2 Line 4. It is confusing. Perhaps: acoustic emission 10 -1,000 kHz (AE)

Pag 2 Line 15 . Please check these two references.

Pag 2 line 25. Perhaps explaining what the tripartite arrays are and method (or indicating a reference) could be informative to the reader.

Pag 3 line 2. Replace tele-earthquakes by teleseisms (I understand the context, but I think that the term tele-earthquake is not correct)

Pag 4 line 4. I would stress the difference between the sites when beginning with Pechgraben. e.g. As regards PG….

Replace PG15 seismic arrays by seismic arrays in the PG15 campaign.

Pag 5 line 8. I would include a subsection entitled "classification" . The features you indicate are not simple waveform features: **Apparent velocity of trackable wave packets** is not a single waveform. You calculate the app. velocity with more than 1 waveform. Rewrite the sentence.

Pag 5 line 13. Indicate the software or method used to obtain the app. velocities.

Pag 5 line 16.   Replace **Unique versus multiple events** by Number of events, Clustering of events or similar because this is not a feature name.

Pag 5 line 21. I do not follow this sentence.  If the distance is short, then there is little attenuation. Attenuation is caused by geometrical spreading (distance) and by intrinsic attenuation. Although you explain the situation below, this specific sentence leads to a misunderstanding.  Rewrite the sentence.

Pag 5 lines 26-28.  This is not part of classification.  Don't think it is strictly method?

Pag 6 line 4.  Replace teleseismic  by Teleseisms

Pag 6 line 9. Specify that in Fig. 2 the layout is for an earthquake. You are in a section devoted to **microseismic signals typology at clayey landslides.** Perhaps a short introduction is needed to avoid confusion.

Pag 6 line 10. Split the sentence into two parts.  …. ordinate.  For Pechgraben  NqF is 1.95…. Note that you are referring to the plots of an earthquake recorded in SZ10.

Pag 6 line 14. At least indicate that the colours are in dB, if you do not show the colour scale.

Pag 6 line 21. …. Hz, defined as b1 to b5.

Pag 7 line 21. I understand that seismograms are not displayed by distance with the result that no apparent velocities can be obtained from the plots of figure 3. It could be helpful to indicate this in the figure caption.

Pag 7 line 25. Replace Fig. 4b and 4e, lower panel by Fig. 4b and 4e(lower panel). The same in the other cases below.

Pag 7 line 26. I would replace "consist of " by "appear as ". In fact, you have not modelled the wave field.

Pag. 7.  It seems to me that the differences between Type III and IV are very small. Is the value of the duration indicated in Table 1 for Type III (moderate distance) correct? As showed, is the apparent velocity the only difference? I am not aware if this is sufficiently stressed in the text.

Pag 9 line 20. Mention Table 1 somewhere in the ETS-like signal and in the Confirmed rockfall events paragraphs.

Pag 10 line 25. I would number the recording stations to identify the seismograms in fig 7 and 8 so as to help the reader.

Pag 11 line 1-10. Same as above comment for figures 8 and 9.

Pag 11 line 28. This link is not working at the moment

Pag 11 source location. I suggest that you should include a reference or a discussion on seismic array configuration optimization in this section (or in 6.2). This is well known in earthquake studies, but perhaps it is worth emphasizing this here.

Pag 12 lines 15-16 and 24. Perhaps I am lost, but don't you think that there is a contradiction?

Pag 13 line 15. For me this is not a valid argument. Note that site effects are frequency dependent. The frequency content of teleseisms and distant earthquakes is lower than that of the events that you are considering. Fig 2 indicates f<10Hz. Moreover, note the frequencies indicated in Table 1. Perhaps applying Kanamura method H/V would be useful.

Pag 14 line 8. I am sorry, but I do not follow you. Why do you not represent the values of the amplitudes in the plots in Fig. 12b? Perhaps an explanation is needed.

Pag 14 line 11. This link is not working at the moment

Pag 14 line 16. Is the value $5e10^6$ correct?

Pag 14 line 22. Complete the legend of Fig. 12b indicating this.

Pag 15 line 13. Replace this by near, local and regional earthquakes and teleseisms.

Pg. 16 line 32. Schlindwein et al., 1995 is not indicated in the references.

Pag 19 line 18. Note that the name of my department has changed. It is Department of Earth and Ocean Dynamics. Faculty of Earth Sciences. University of Barcelona (UB)

Figures, Figure captions and Tables

- Is the value of the duration for Type III (moderate distance) indicated in Table 1 correct?

- Figure caption 1. Please define PG15, PG16 and SZ10

- Figure 2. Please replace the title. Seismic features of an earthquake in different representations… You are using different representations to show the characteristics

- Fig 6a. Indicate bp1 to bp5 similar as in Fig 4a

- Fig 8. b) You must add "at short distances" after human footsteps. This must be specified.

- Fig 9.  a) You must add "at short distances" after human footsteps. This must be specified.

- Fig 10. I would indicate the solution by an open circle.

- Fig 12. In b) the extrapolated functions must be indicated. In line 5, is projection the correct word?

- Figure 15. I noticed that you have unified the slide quakes and dispersive tremors in landslide- induced tremors. This is consistent with the classification of your catalogue. Accordingly, I would replace Quakes by landslide –induced microearthquakes to be consistent.

- References

Aki and Richards 2002, is not mentioned in the text.

Joswig 2008. Please complete or correct (capital LETTERS) the reference:  First Break (June)

---

## Author Response (AR3)

**Final response letter to the Editor and the Reviewers**

Dear Editor and dear Reviewers,

My co-authors and myself would like to thank you very much for your time, efforts and contribution in helping us improving the quality of this paper. We are very thankful to Fabian Walter (Editor), Emma Suriñach (Reviewer #1) and Andreas Köhler (Reviewer #2) for reviewing our manuscript; we've also much appreciated a non-anonymous review!

In this letter we have responded the comments made on the revised manuscript. We've addressed each comment made by the two Reviewers individually in red and applied the suggested and related modifications on the revised manuscript version of March 20, 2018, using a track-change mode. Reference to page and line numbering is given in bold and correspond to the final version of the revised manuscript in track change mode.

We hope that our changes and clarifications, both in this letter and in the final revision manuscript text and figures, fully address the remarks made by the two reviewers.

Again, we would like to thank you for your time and efforts in helping us improving this manuscript.

With my best regards,

Naomi Vouillamoz

**Associate Editor Decision: Publish subject to minor revisions (review by editor) (01 May 2018) by Fabian Walter**

Comments to the Author:

Dear authors,

thank you for your revisions, which substantially improved the quality of your submission. Based on the additional round of external reviews, your manuscript can be published after minor revisions. Please read the reviews carefully and respond to them in the next iteration. They are mostly minor aimed at improving the readability and clarity of your work. However, I fully agree with Reviewer 1 who specifically asks for scatter plots of continuous parameters. This will be a concise way to illustrate and support some of your major points. Please pay particular attention to this suggestion.

Best,

Fabian Walter.

**Report #1 from Referee #2 Andreas Köhler**

Suggestions for revision or reasons for rejection (will be published if the paper is accepted for final publication)

I think the manuscript has improve considerably and most of my previously comments have been addressed. I have just a few questions and suggestions:(1) I suggested to show scatter plots to visualize the distribution of event features and to evaluate how well defined the event classes are (clustering vs. end-members of a continuous distribution). I see from the new Fig. 14 that this is of course only possible for continuous variables like apparent velocity (3), duration (5), and frequency (7). In my opinion it would still be helpful to plot those features against each other (i.e, (3) vs. (5), (3)

vs. (7), etc.) using the measurements of all events. The variable "S" should also be included. However, maybe I misunderstood and those quantities are not stored, but only evaluated visually and qualitatively during the interactive processing?

As much as we agree that such plots would provide a great and elegant way to derive conclusions about our classification, we cannot generate such graphics at the moment and do not think they would necessarily help to better define the classes. Our arguments for this are manifold:

1) Those quantities are indeed not stored in a way we can produce continuous plots. Signal durations were attributed as a duration range for a given event (< 2 s; 2-10 s; > 10 s); dominant frequencies were defined also as a range. The related quantitative values that we have stored are the maximum station amplitudes of the event measured in the series of pre-defined bandpass waveform ((1-5, 5-20, 20-50, 50-100, 100-200 Hz) see the figure 1 below). Apparent velocity could not be derived for all events and in some instances also defined as a range, due to the general bad waveform quality and the corresponding uncertainties.

2) These attributes, especially the frequency content, are strongly dependent to the distance and the size of the source, meaning that one should expect important variations of some attributes for events of the same kind but recorded in different distances, as well as for records from one single event at different stations (i.e. different receiver-source distances).

3) The statistical significance in terms of high-quality end-members events of our dataset is unfortunately low, additional measurements and more complex waveform and spectral attribute shall be used to pursue and refine the classification.

**We would like to stress that the method applied which consist in visual sonogram screening is de facto a PATTERN RECOGNITION method.** The human eye still performs better than any currently existing algorithm. One will not miss a face, whatever its size, resolution or color in a bad quality picture, the best algorithms still do! As mentioned in part 6, further work for automated event detection and classification systems will require more complex waveform and spectral features as the ones used in this paper.

We added references in the method parts: Joswig, 1990,1995,1996 regarding sonograms and Vouillamoz 2015, Sick, 2016 regarding visual pattern recognition by sonogram screening.

The Figure 1 below displays data of the SZ10 catalog. For each event of the SZ10 catalog, the trace featuring the maximum signal amplitude was used. Then, maximum absolute 0-to-peak amplitudes were computed for the signal within the five defined bandpass filters (1-5, 5-20, 20-50, 50-100, 100-200 Hz) ($A_{max-bp}$) and scaled (as a percentage) to the maximum of the $A_{max-bp}$ of that trace. Except for the distant earthquakes which show a coherent pattern of higher amplitudes in the 1-5 Hz bandpass, the distinction of the other classes is not straightforward, showing many patterns overlaps and illustrating the complexity of finding an unequivocal classification based on simple seismic features.

[Figure]

*Figure 1.*

(2) Array processing: I agree that classical beamforming (FK analysis) with non-planar wave fronts will probably still provide an idea about the source direction. Resolving the quadrant sounds reasonable. However, since the authors also use the apparent velocity: How much biased will this measurement be for sources very close to the array? Does it affect the classification?

No, we don't think this affects the classification.

We computed apparent velocity for those events displaying well defined and trackable wave packets, hence implying a source originated outside the recording network. Apparent velocity was not used for event location, we applied a pre-defined velocity model (following Tonnellier et al. 2013) and apparent velocity could not be computed for events originated within the network (no coherent solution was found using jackknifing as explained for example in Joswig, 2008 or Vouillamoz et al. 2016, Figure 5). As mentioned above, for some events, we ended up defining a range of apparent velocity. But clearly, a nearby source always features slow apparent velocity (< 2 km s$^{-1}$), hence the classification is not affected.

(3) Figure 14: I think I misunderstood the comparison of displacement rates and seismic emission rates previously. It seems that there is just a single average displacement rate in cm per day for each data set and not a displacement for each day Yes, as mentioned in the Data section (Why could the displacement not be measured with daily resolution? Moneywise and lack of human resources). So the correlation between seismic and displacement rate is based on three data points (three data

sets). Yes. In other words, also seismic rates are averaged rates over all days. Yes. Please make sure that this is clearly described in the text or clarify if I am wrong.

We specified **average** daily rates of landslide-induced microseismic events in Section 6.3. (now section 6.4)

(4) Earth surface dynamics allows to publish supplementary material / data. Have you thought about publishing the complete event catalog as supplement (not raw seismic waveforms, but plots like Fig.6 for each event and/or detection time, duration, frequency, apparent velocity, and "S" for each events)?

Yes, we have thought about it. However, as we still wish to publish further analysis of these data, we prefer to keep a track records of the dataset. As specified in the data and resources section, the data are of course available upon request to the authors (in SEG-2 or MSEED data format). All events traces are furthermore available in ASCII (*.dat) data format (1 file for one event and one station). All the plots (waveforms, spectrogram, bandpass filtered waveforms, amplitude spectrums…) are also available for each event and event trace.

**Report #2 from Referee #1 Emma Suriñach**

**Comments on the new version of the paper**

**Characterizing the complexity of seismic signals at slow-moving clayrich debris slides: The Super-Sauze (Southeastern France) and Pechgraben (Upper Austria) case studies by Naomi Vouillamoz, et al.,**

In this new version the authors have considerably improved the manuscript. The present version is much better organized and the aims have been clarified.

It is a shame that in the reorganisation of figure 14, the authors have omitted information on precipitation. I hope this will be explained in the future together with an explanation of the situation. As mentioned in the previous iteration, this is the subject of the PhD thesis of Sabrina Rothmund (submitted) and will be the subject of a new publication. Now, it is clear that the aim of the paper is only to classify and to catalogue the seismic signals.

**As regards this, I suggest a change in the title of the paper.**

We do not wish to change the title of the paper. We do characterize the signals. The paper also includes a 'location' and 'magnitude' section.

The incorporation of the S value in section 5.2, as suggested by Andreas Khöler, clarifies the results. Good suggestion.

I suggest that you should include a reference or a discussion on seismic array configuration optimization. This is well known in earthquake studies, but perhaps it is worth emphasizing this here.

We included a new section 6.1 "Passive seismic monitoring at clayey landslides" to discuss the seismic network.

Further comments

Pag 1 Line 25. I am not sure that this reference must be included here.

We moved the reference of P1 L25 at the end of the sentence.

Pag 2 Line 4. It is confusing. Perhaps: acoustic emission 10 -1,000 kHz (AE)

We modified P2 L3 to: "Seismic investigations of natural and artificial slope instabilities started in the 1960's with acoustic emission (10-1,000 kHz)"; as the acronym is not used further in the paper.

Pag 2 Line 15. Please check these two references.

The two references of Brückl and Mertl, 2006; Mertl and Brückl, 2007 are correct.

Pag 2 line 25. Perhaps explaining what the tripartite arrays are and method (or indicating a reference) could be informative to the reader.

We included a sentence. The text reads now P2 L25-30:

"This study aims at proposing a classification of microseismic signal types as recorded by tripartite microseismic arrays deployed at slow-moving clay-rich debris slides ("clayey landslides"). Tripartite microseismic arrays are suited for the determination of the back azimuth and apparent velocity of an incoming signal, hence providing key information about the signal source location (e.g. Joswig, 2008; Sick et al., 2012; Vouillamoz et al., 2016). The classification of microseismic signals is based on waveform and spectral attributes of the signals and uses microseismic observations reported by similar case studies as a benchmark."

Pag 3 line 2. Replace tele-earthquakes by teleseisms (I understand the context, but I think that the term tele-earthquake is not correct).

It is P4 now L5. Modified as asked.

Pag 4 line 4. I would stress the difference between the sites when beginning with Pechgraben. e.g. As regards PG….

We modified P4 L7-9 to: "At Pechgraben, due to the relatively large aperture (30-50 m) of the seismic arrays in the PG15 campaign, many near-source area microseismic events were recorded by less than three sensors. Consequently, a denser seismic network configuration was designed for the PG16 campaign."

Replace PG15 seismic arrays by seismic arrays in the PG15 campaign.

Responded by previous comment.

Pag 5 line 8. I would include a subsection entitled "classification" . The features you indicate are not simple waveform features: Apparent velocity of trackable wave packets is not a single waveform. You calculate the app. velocity with more than 1 waveform. Rewrite the sentence.

We included the section 3.1 Classification and modified P5 L18-19 to: "considering the following features:"

Pag 5 line 13. Indicate the software or method used to obtain the app. velocities.

The software is indicated P5 L4: HypoLine;

We added a reference to Vouillamoz et al. 2016, where the approach to compute the apparent velocity in HypoLine is described.

The sentence P5 L3-6 reads now:

"Each detection was first evaluated individually and interactively in HypoLine, where phases information were picked, and time offsets between array-correlated wave packets used to derive apparent velocity and back azimuth information following the approach described in Figure 5 of Vouillamoz et al., 2016."

Pag 5 line 16. Replace Unique versus multiple events by Number of events, Clustering of events or similar because this is not a feature name.

We modified to "clustering of events", also in Figure 14.

Pag 5 line 21. I do not follow this sentence. If the distance is short, then there is little attenuation. Attenuation is caused by geometrical spreading (distance) and by intrinsic attenuation. Although you explain the situation below, this specific sentence leads to a misunderstanding. Rewrite the sentence.

We agree, the paragraph was not clear and misleading. Attenuation is of course more important for larger distance. In our case, working at distances of less than a few hundred meters, we observed events originated a few hundred meters away from the networks showing waveforms being attenuated homogeneously resulting in all the waveform showing more or less the same amplitudes. On the contrary, a source originated within the seismic network displays a large range of amplitudes (several orders of units). We rewrote the paragraph P5 L29 P6 L6:

"The signals of landslide-induced microseismic sources are expected to be severely attenuated, mainly because of their propagation through heterogenous clay-rich soils of various water saturation (e.g. Koerner et al., 1981). Calibration shots and hammer blows carried out at Super-Sauze and Pechgraben showed that sources occurring within the seismic network feature prominent waveform attenuation across the seismic network, whereas sources originated a few hundred meters outside the seismic network feature waveforms being homogeneously attenuated, resulting in similar signal amplitudes across the seismic network. Therefore, only microseismic events featuring prominent and consistent attenuation of the signal maximum amplitudes across the seismic network are considered as a nearby source, potentially induced by the landslide dynamics."

Pag 5 lines 26-28. This is not part of classification. Don't think it is strictly method?

We moved the sentence to P5 L14-16: "Since the short receiver-source distances of the considered signals do not allow a clear separation of body waves and surface waves, amplitude information was taken as the maximum absolute 0-to-peak amplitude of the signal unfiltered vertical seismogram."

Pag 6 line 4. Replace teleseismic by Teleseisms

Modified P6 L18 as asked.

Pag 6 line 9. Specify that in Fig. 2 the layout is for an earthquake. You are in a section devoted to microseismic signals typology at clayey landslides. Perhaps a short introduction is needed to avoid confusion.

we specified P6 L23-24: "we apply the layout of Figure 2, which illustrates an earthquake signal…"

Pag 6 line 10. Split the sentence into two parts. …. ordinate. For Pechgraben NqF is 1.95…. Note that you are referring to the plots of an earthquake recorded in SZ10.

We modified to P6 L24-25:

a.       Shows the signal sonogram (Joswig, 1990) up to the Nyquist frequency with a logarithmic ordinate, which corresponds to 1.95-250 Hz for Pechgraben data and to 3.91-500 Hz for Super-Sauze data.

Pag 6 line 14. At least indicate that the colours are in dB, if you do not show the colour scale.

Modified P6 L29 as asked.

Pag 6 line 21. …. Hz, defined as b1 to b5.

Modified P7 L2 to: "defined as bp1 to bp5"

Pag 7 line 21. I understand that seismograms are not displayed by distance with the result that no apparent velocities can be obtained from the plots of figure 3. It could be helpful to indicate this in the figure caption.

Even if the plots of Figure 3 were displayed as a function of source distance, one could not derive a decent apparent velocity based on the figure resolution!

Pag 7 line 25. Replace Fig. 4b and 4e, lower panel by Fig. 4b and 4e(lower panel). The same in the other cases below.

We modified this consistently in the document.

Pag 7 line 26. I would replace "consist of " by "appear as ". In fact, you have not modelled the wave field.

Modified P8 L8 as asked.

Pag. 7. It seems to me that the differences between Type III and IV are very small. Is the value of the duration indicated in Table 1 for Type III (moderate distance) correct? As showed, is the apparent velocity the only difference? I am not aware if this is sufficiently stressed in the text.

Text P17 L21-25: "Type III and type IV events feature S values which are below 200 % and must represent a continuous transition of quake events recorded at larger receiver-source distances. The higher apparent velocities of wave packets of type IV events and the consistent signal amplitudes of well distinguishable successive phases across the seismic network suggest a source origin outside of the landslide body in the host rock.

Pag 9 line 20. Mention Table 1 somewhere in the ETS-like signal and in the Confirmed rockfall events paragraphs.

We mentioned Table 1 at P9 L21.

Pag 10 line 25. I would number the recording stations to identify the seismograms in fig 7 and 8 so as to help the reader.

As much we are willing to help the reader, indicating a complete station numbering in Figure 8 results in a pretty bad resolution…

Pag 11 line 1-10. Same as above comment for figures 8 and 9.

Stations are indicated in the caption of Figure 9!

Pag 11 line 28. This link is not working at the moment!

It was working on September 13. The internet page was unfortunately removed.

We modified P12 L10-11 to: "developed by the LIAG, Leibniz-Institut für Angewandte Geophysik, Germany".

Pag 11 source location. I suggest that you should include a reference or a discussion on seismic array configuration optimization in this section (or in 6.2). This is well known in earthquake studies, but perhaps it is worth emphasizing this here.

Answered previously in the non-numbered comments. We included a section 6.1 Passive seismic monitoring at clayey landslides to discuss passive seismic monitoring at active landslide.

Pag 12 lines 15-16 and 24. Perhaps I am lost, but don't you think that there is a contradiction?

A source located within the seismic network is resolved without the need of array processing. Phase information are processed in network mode, providing the hypolines (see Joswig 2008; Vouillamoz et al. 2016 for details).

Pag 13 line 15. For me this is not a valid argument. Note that site effects are frequency dependent. Yes. The frequency content of teleseisms and distant earthquakes is lower than that of the events that you are considering. Yes. Fig 2 indicates f<10Hz. Moreover, note the frequencies indicated in Table 1. Table 1 is intended to give general characteristics of local and more distance earthquakes. For local earthquakes, we still have energies up to 20 Hz. Perhaps applying Kanamura method H/V would be useful. Of course, in an additional publication maybe?

Pag 14 line 8. I am sorry, but I do not follow you. Why do you not represent the values of the amplitudes in the plots in Fig. 12b? Perhaps an explanation is needed.

Local magnitude is still computed by many seismological institutions around the world by applying a Wood-Anderson filter to transform seismic records measured by a modern seismometer in nm s$^{-1}$

into "Wood-Anderson" records in mm as used in the original definition of the local magnitude by Richter.

It is however more convenient to work directly with the original data, as measured, in nm s$^{-1}$ and get rid of all the filtering needed to convert and measure the peak-to-peak Wood-Anderson amplitude in mm (which is in addition period/frequency dependent). Other studies used such approaches, for instance: **Gaucher, 2015**; Earthquake detection probability within a seismically quiet area: application to the Bruchsal geothermal field, Geophysical Prospecting, doi: 10.1111/1365-2478.12270; or Edwards et al. 2015; Seismic monitoring and analysis of deep geothermal projects in St Gallen and Basel, Switzerland, Geophysical Journal International, doi: 10.1093/gji/ggv059.

Pag 14 line 11. This link is not working at the moment

This page has also been removed. We modified P14 L26 to: SISSY product information sheet.

Pag 14 line 16. Is the value 5e106 correct?

Yes, 5e10$^6$ is the correct value! It could be discussed of course, but it was the best compromise we found between the different calibration datasets we have.

Pag 14 line 22. Complete the legend of Fig. 12b indicating this.

Indicated in the caption of Figure 12 as asked.

Pag 15 line 13. Replace this by near, local and regional earthquakes and teleseisms.

Modified P 16 L12 as asked.

Pg. 16 line 32. Schlindwein et al., 1995 is not indicated in the references.

We added the reference.

Pag 19 line 18. Note that the name of my department has changed. It is Department of Earth and Ocean Dynamics. Faculty of Earth Sciences. University of Barcelona (UB)

We corrected the name of your department.

Figures, Figure captions and Tables

- Is the value of the duration for Type III (moderate distance) indicated in Table 1 correct?

Yes! It is about 2 seconds.

- Figure caption 1. Please define PG15, PG16 and SZ10.

We specified.

- Figure 2. Please replace the title. Seismic features of an earthquake in different representations… You are using different representations to show the characteristics

Modified as asked.

- Fig 6a. Indicate bp1 to bp5 similar as in Fig 4a

We modified the figure 6 and 7 as asked

- Fig 8. b) You must add "at short distances" after human footsteps. This must be specified.

We agree and modified as asked.

- Fig 9. a) You must add "at short distances" after human footsteps. This must be specified.

Modified as asked.

- Fig 10. I would indicate the solution by an open circle.

There are several solutions, hence the uncertainties…

- Fig 12. In b) the extrapolated functions must be indicated. In line 5, is projection the correct word?

This is now indicated it in the plot of Fig 12b.

- Figure 15. I noticed that you have unified the slide quakes and dispersive tremors in landslide-induced tremors. This is consistent with the classification of your catalogue. Accordingly, I would replace Quakes by landslide –induced microearthquakes to be consistent.

Modified as asked.

- References

Aki and Richards 2002, is not mentioned in the text.

We removed the reference.

Joswig 2008. Please complete or correct (capital LETTERS) the reference: First Break (June).

We corrected the capital letters.